# Humanized avian embryo models replicate an immune tumor environment for rapid immunotherapy studies

Marjorie Lacourrège [1✉], Loraine Jarrosson[1], Clélia Costechareyre[1], Mathis Marcel[1], Audrey Prunet [2], Mayrone Mongellaz[1], Camille Futelot[1], Cyrus Rouhani[1], Huong Giang Nguyen [3], Thomas Aparicio [3,4], Lionel Le Bourhis[3], Céline Delloye-Bourgeois [2,5,6✉], Romain Teinturier [1,6✉] & Valérie Castellani [2,6✉]

## Abstract

**Re-arming the immune system to fight cancer holds significant promise. However, models enabling the rapid and reliable evaluation of immunotherapies on patient tumors are still lacking. Here, we report a humanized version of the avian model for cell and patient-derived xenograft models, which involves the creation of miniature replicas of patient and cell line-derived tumors in avian embryo tissues. Leveraging this unique technology, we developed two approaches: co-micro-implantation of human PBMCs and cancer cells or implantation of cancer cells followed by the intravenous injection of human PBMCs. By combining whole-embryo light sheet microscopy, analysis of immune cell molecular markers and evaluation of anti-PD-1 therapy in solid tumor models, we demonstrate that the humanized AVI technology recapitulates an immune microenvironment, enables the rapid screening of immune-based therapeutic strategies and supports patient response stratification for biomarker discovery.**

**Keywords** Avian Embryo Model; Humanized Preclinical Model; Immuno-oncology; Immunotherapy; Solid Cancer
**Subject Categories** Cancer; Immunology; Methods & Resources

## Introduction

The mutational burden of cancer cells creates major differences from their physiological cells of origin, making them recognizable to the immune system. However, cancer cells activate immuno-suppressive mechanisms to evade immune surveillance (Chen and Mellman, 2017). Therapeutic targeting of these immune check-points, such as T-lymphocyte-associated protein 4 (CTLA-4) or programmed cell death 1 (PD1), has emerged as one of the most promising paradigms for the treatment of cancer, including solid tumors such as metastatic melanoma and non-small cell lung cancer (Rui et al, 2023; Sharma et al, 2023). The clinical success of immunotherapies with efficient and durable anti-tumor effects provides evidence that affected individuals have a reservoir of immune cells that can potentially be remobilized to attack malignant cells.

However, current immunotherapies remain ineffective in many patients (Haslam et al, 2025). The Society of Immunotherapy for Cancer (SITC) distinguishes patients with primary resistance, secondary post-treatment resistance and third acquired post-treatment resistance (Sharma et al, 2023). Failure of immunotherapy is thought to result from both intrinsic features of cancer cells and the tumor microenvironment. Developing novel approaches to unlock the system in resistant patients is a major challenge. It requires a better understanding of the complex communication between immune-cancer cells, of the biological variables from the molecular to the tissue level, and even including patient history, that may modulate their outcome. Extensive efforts are being made to restore the functions of immune cell effectors with various strategies including combination therapies, enhancing T lympho-cyte responses with co-stimulation, targeting novel immune checkpoints, stromal cell components and peripheral immune cells (Gould Rothberg et al, 2022).

The degree of tumor immune infiltration by T and NK cells, dendritic cells, M1 type macrophages and monocytes is a first-level predictor of response to immunotherapy. In addition, certain molecular features of tumors are used to predict the outcome of immunotherapeutic treatments. For immunotherapies targeting the PD-1/PD-L1 axis, levels of the PD-1 ligand (PD-L1) in tumors, deficiencies in mismatch repair (dMMR), and high microsatellite instability (MSI-high) are the main markers currently used to determine patient eligibility (Landre and Guetz, 2025). Likewise in CRC, in about 15% of cases, cancer cells have a dysfunctional DNA mismatch repair (dMMR) which can lead to microsatellite

[1]Oncofactory ERBC, Bioparc 60 Avenue Rockefeller, Lyon 69008, France. [2]Université Claude Bernard Lyon 1, MeLiS, INSERM 1314, CNRS 5284, Lyon, France. [3]Intestinal Immunity in Inflammation and Cancer (3IC), Institut de Recherche Saint-Louis INSERM U1342, Paris, France. [4]Gastroenterology Department, Saint Louis Hospital, APHP, Paris, France. [5]Present address: Université Claude Bernard Lyon 1, Cancer Research Center of Lyon, Inserm 1052, CNRS 5286, Lyon, France. [6]These authors contributed equally: Céline Delloye-Bourgeois, Romain Teinturier, Valérie Castellani. ✉E-mail: mlacourrege@erbc-group.com; Celine.DELLOYE@lyon.unicancer.fr; rteinturier@erbc-group.com; valerie.castellani@univ-lyon1.fr

instability (MSI) during DNA replication. Tumors with an MSI-high profile are heavily infiltrated by immune cells making patients eligible for anti-PD-1 treatment. However, the predictive value of these parameters remains partial (Monette et al, 2024).

Current immunotherapy research relies on a variety of preclinical models. Historically, immunocompetent syngeneic mouse models first allowed the discovery of currently approved immune checkpoint blockers such as anti-PD-1 and anti-CTLA-4. To overcome the major drawback of the non-human (mouse) immune system, sophisticated mouse models that recapitulate the human immune system have been developed based on the injection of peripheral blood mononuclear cells (PBMCs) or CD34+ hematopoietic stem cells (de La Rochere et al, 2018). The combination of these models with patient-derived xenografts adds a layer of complexity, which is difficult to reconcile with the need for rapid investigation of patient tumor response and inter-patient heterogeneity. Therefore, there is an urgent need for alternatives to evaluate the efficacy of candidate immunotherapies in models while incorporating patient tumor heterogeneity.

In previous work, we reported the development of avian patient-derived xenograft (PDX) models which involve the micro-implantation of tumor cells into the tissues of the avian embryo and the assessment of therapeutic efficacy within a few days using light sheet microscopy (Jarrosson et al, 2021; 2023; Zala et al, 2024). We demonstrated that this paradigm enables cancer cells to form tumors in an environment representative of tissue and whole-organism contexts. The avian embryo environment supports cell survival, even when the number of transplanted cells is low. This enables the cancer cells to communicate with the host and express their tumorigenic and metastatic potential (Jarrosson et al, 2021; 2023; Zala et al, 2024; Villalard et al, 2024). In models of follicular lymphoma and metastatic melanoma, we have shown that these miniaturized replicas not only efficiently revealed the efficacy of standard therapies but also faithfully recapitulated heterogeneous patient tumor responses.

Building on the advantages of the avian PDX technology, we describe in the present study the conception of a humanized version of this model featuring a reconstituted human immune microenvironment that recapitulates tumor infiltration processes and cancer-immune cell communication. We also report the study of cancer cell responses to anti-PD-1 in models of melanoma, breast and colorectal cancer (CRC), demonstrating that the humanized-AVI-PDX models allow rapid screening of immunotherapies and stratification of patient responses.

## Results

### Establishing humanized avian embryo models through allogeneic hu-PBMCs engraftment within a short-time frame

In our previous work, we demonstrated that engrafting human cancer cells into the ventral somitic compartment prefiguring the axial skeleton, provides a supportive microenvironment for tumor intake and growth (Jarrosson et al, 2021). To extend the use of such xenograft model in the field of immuno-oncology, we asked whether we could recapitulate a human immune component in the avian embryo. Thus, we labeled allogeneic human PBMCs (hu-PBMCs)

with vital fluorescent Orange Cell Tracker (OCT) prior to their transplantation within the ventral somite of Embryonic Day 2 (E2) avian embryos (Fig. 1A). Forty-eight hours later (E4 stage), avian embryos were harvested to image fluorescent hu-PBMCs using light sheet confocal microscopy. Interestingly, we observed clusters of PBMCs within the somite 48 h post-transplantation, with some cells spreading in adjacent tissues (Fig. 1B). This shows that the avian embryo provides an environment supporting hu-PBMCs survival. We thus further investigated the outcome of hu-PBMC engraftment.

Allogeneic hu-PBMCs exhibit donor-dependent immunogenicity that might affect the survival of the animal recipient and the success of transplantation, as shown in humanized mouse models with induction of lethal graft-versus-host disease (GvDH) (de La Rochere et al, 2018). To take into consideration this parameter, we implanted hu-PBMC samples isolated from whole blood of five different healthy donors and analyzed the survival of chick embryos 48 h post-transplantation in several independent experiments (Fig. 1C). As expected, the average survival rate varied between donors, from 69% to 91%. Nevertheless, it reflected high tolerability in all cases, since engrafted control embryos show around 80% average survival rate (Jarrosson et al, 2021; 2023). In previous work, we defined several morphological checkpoints (brain, eye, limbs, heart) to evaluate potential developmental defects in manipulated embryos (Jarrosson et al, 2021; 2023). We found that hu-PBMCs transplantation neither affected these features nor the global growth of embryos, as assessed by the measurement of their body surface area (BSA).

Next, to further study engrafted hu-PBMCs, we micro-dissected the somitic domain containing hu-PBMCs clusters and dissociated the tissues, then analyzed their viability by flow cytometry. For all hu-PBMCS donors, we observed a CD45+ cell viability exceeding 79% (Fig. 1D). Thus over 48 h post-implantation, the avian embryo supports a high survival rate of engrafted CD45+ cells. To explore this further, we analyzed the distribution of the main immune cell sub-populations by flow cytometry at this time point (Fig. 1E, F). Among viable human CD45+ cells, we monitored CD3 + CD4+ and CD3 + CD8 + T lymphocytes, B lymphocytes, NK cells, as well as the myeloid cell fraction. First, we pooled data obtained for these 5 healthy donors and compared the fractions represented by the different cell-types within the whole population before and after implantation in avian embryos. Interestingly, we found that the diversity of immune cell populations was maintained in the transplanted samples. When compared to pre-transplanted hu-PBMCs, we found that the CD8+ cell representativity was unchanged, while that of the other populations showed significant variations, decreasing for NK, CD4+ and myeloid cells, and increasing for B cells. We also assessed the inter-individual variability, by analyzing the data per donor (Fig. 1G). While we found that all cell populations were present in all donors, their respective frequency significantly fluctuated between donors. Altogether, beyond these inter-individual differences, our study shows that all tested immune cell populations were present in consistent proportions within a 48 h post-implantation. A set of these donors was used in the following experiments.

### Development of an avian model to study intra-tumoral immune cell infiltration upon I.V injection of hu-PBMCs

Next, we thought to set-up a process combining implantation of cancer cells in the ventral somite and intravenous injection of hu-

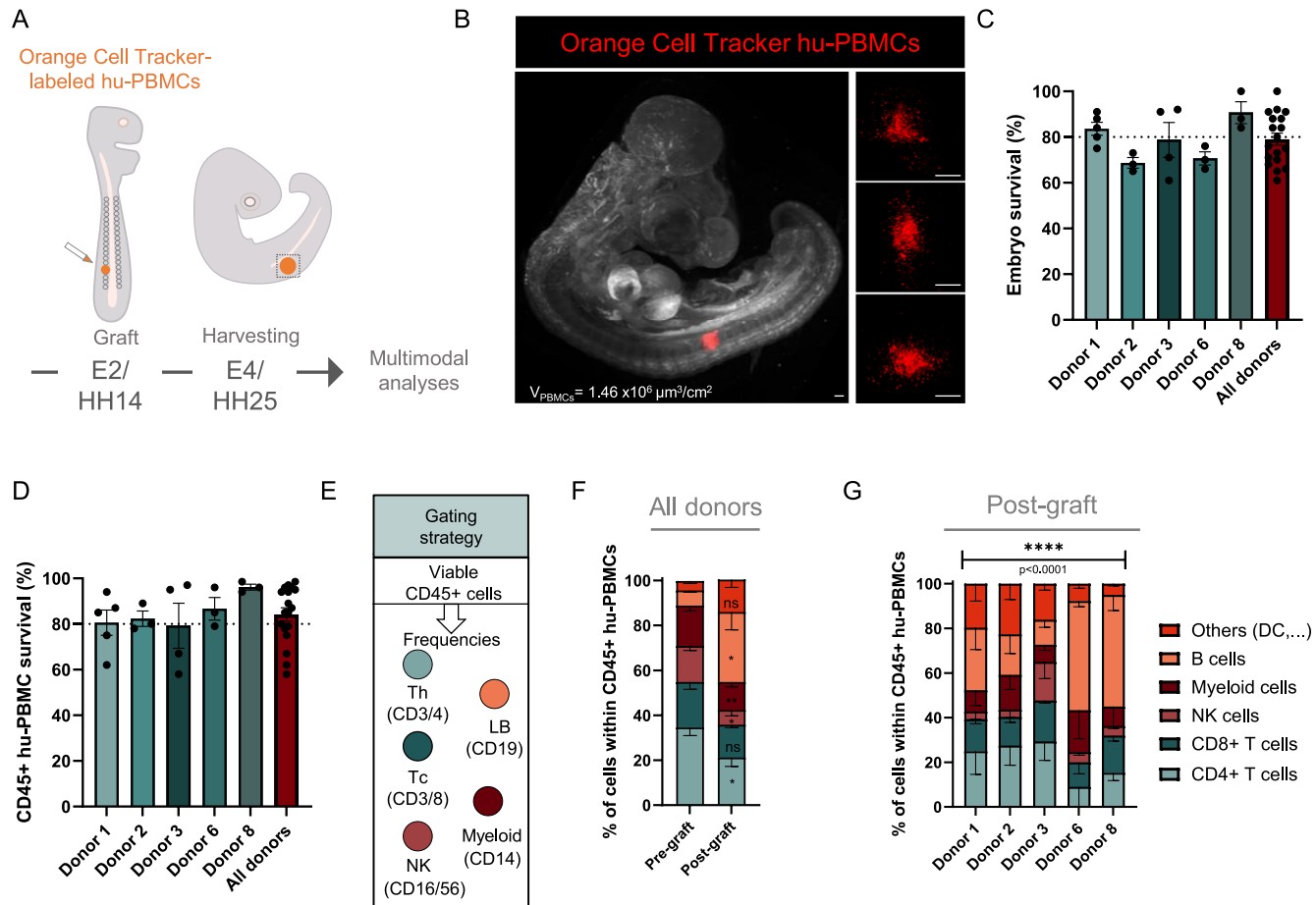

**Figure 1. Humanization of avian embryos with hu-PBMCs allows high survival rate of host and grafted immune cells.**

(**A**) Schematic drawing of the humanization process. Hu-PBMCs are micro-injected in developing somites at E2 stage (HH14). Humanized embryos are harvested 48 h post-graft, at E4 stage (HH25), and processed for a panel of analyses. (**B**) 3D view of HH25 chick embryo engrafted with hu-PBMCs labeled with Orange Cell Tracker and capture of the fluorescent signal using Imaris software (red). Volume of PBMC normalized to the BSA of the embryo is indicated on the picture ($V_{PBMC}$). Scale bar: 250 μm. (**C**) Histograms of survival rate of chick embryos engrafted with hu-PBMCs from 5 healthy donors, per donor and for pooled donors. Each dot represents an independent experiment. $N = 5$ experiments for donor 1 ($n = 25$ to 41 engrafted embryos); $N = 3$ donor 2 ($n = 33$ to 38); $N = 4$ donor 3 ($n = 20$ to 43); $N = 3$ donor 6 ($n = 20$ to 24); $N = 3$ donor 8 ($n = 16$ to 17). Data are represented as mean ± SEM. (**D**) Histograms of survival rate of CD45+ hu-PBMCs post engraftment in avian embryos, for 5 healthy donors and for pooled donors. Each dot represents an independent experiment. $N = 5$ experiments for donor 1 ($n = 25$ to 41 engrafted embryos); $N = 3$ donor 2 ($n = 33$ to 38); $N = 4$ donor 3 ($n = 20$ to 43); $N = 3$ donor 6 ($n = 20$ to 24); $N = 3$ donor 8 ($n = 16$ to 17). Data are represented as mean ± SEM. (**E**) Schematic diagram of immunophenotyping gating strategy: hu-PBMC sub-populations are defined within viable cells and CD45+ cells. Sub-populations are identified based on cell surface protein expression (CD3-CD8 for cytotoxic cells, CD3-CD4 for helper T cells, CD16 and CD56 for NK cells, CD19 for B cells, CD14 for myeloid cells). (**F, G**) Analysis of hu-PBMC sub-populations present in E4 micro-dissected embryonic tissues and comparison with those of corresponding pre-grafted hu-PBMC samples, for pooled donors (**F**) or for individual donors (**G**). The histograms present the percentage of sub-populations within viable CD45+ cells from pooled embryos. $N = 3$ per donor; Donor 1 ($n = 4$ to 6 pooled embryos); donor 2 ($n = 6$); donor 3 ($n = 6$); donor 6 ($n = 7$ to 8); donor 8 ($n = 16$ to 17). Data are represented as mean ± SEM. Paired T test (**F**) or Chi-square test (**G**) was performed. ****$P < 0.0001$, **$P < 0.01$, *$P < 0.05$, ns: not significant. Exact $P$-value indicated on graph (**G**). In (**F**), $P = 0.0102$ for CD4 +; $P = 0.1363$ for CD8 +; $P = 0.0343$ for NK; $P = 0.0075$ for myeloid; $P = 0.0291$ for B cells; $P = 0.0865$ for others. Source data are available online for this figure.

PBMCs. Such a model would allow recapitulating tumor-infiltration by effective immune cells circulating in the blood, which is a key process both in the context of the disease and for effectiveness of therapeutic immune-modulators.

As a first step and as done for implanted hu-PBMCs, we assessed avian embryo viability upon intravenous injection (I.V) of hu-PBMCs from 3 healthy donors (donors 2, 4 and 5) over 48 h (Fig. 2A). We found that the embryos displayed a mean survival rate exceeding 85%, indicating satisfying tolerability (Fig. 2B). Second, we harvested blood from hu-PBMC-injected embryos to analyze the frequencies of viable circulating human CD45+

immune cell populations by flow cytometry. We analyzed the data combining all three donors and found that the diversity of human immune cell populations in avian blood samples was preserved, when compared with pre-grafted hu-PBMCs. Nevertheless, when analyzing the donors individually, we found significant inter-individual differences of cell population frequencies, as observed in the paradigm of hu-PBMCs transplantation in the avian tissue (Fig. 2C,D).

In the second step we engrafted human cancer cells into the ventral somite of avian embryos at E2. We leveraged triple-negative breast cancer cells, which we previously identified as capable of

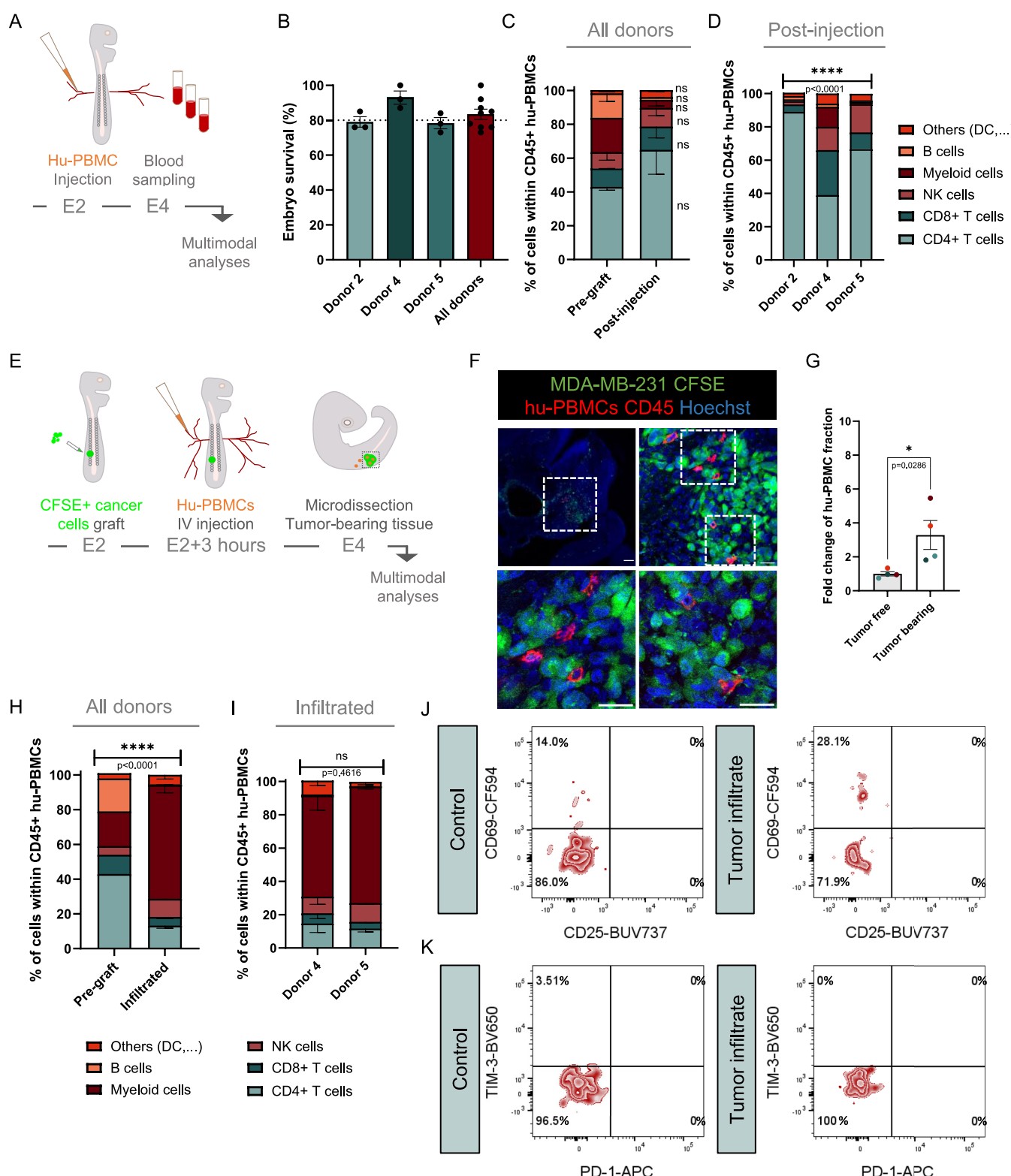

establishing tumors within the ventral somite (Jarrosson et al, 2021). MDA-MB-231 cells were labeled with vital fluorescent tracker CFSE before the graft. Then, 3 h post-implantation, hu-PBMCs were independently injected into the bloodstream of batches of grafted embryos and non-grafted controls. 40 h later,

embryos were harvested for subsequent analyses (Fig. 2E). Firstly, by CD45 immunolabeling on cryosections of avian embryos, we observed the presence of human CD45+ cells within the cancer cell environment, demonstrating that within 40 h, circulating cells were able to infiltrate the tumors (Fig. 2F). Secondly, we assessed

**Figure 2. Tumors formed in the avian embryo are infiltrated by hu-PBMCs injected in the general blood circulation.**

(A) Schematic drawing of the process. Hu-PBMCs were injected intra-venously in E2 embryos. 40 h post-injection, embryos were harvested at E4 stage (HH25) and their blood were sampled for immunophenotyping. (B) Analysis of survival rate of embryos post-injection for individual and pooled hu-PBMC donors. Each dot represents an independent experiment. $N = 3$ per donor; donor 2 ($n = 25$ to 38 injected embryos); donor 4 ($n = 10$ to 30); donor 5 ($n = 23$). Data are represented as mean ± SEM. (C, D) Analysis of hu-PBMCs in HH25 avian embryo blood, comparing cell populations representation prior and post-injection for pooled donors (C) and per donor (D). The histograms present the percentage of sub-populations within viable CD45+ cells from pooled embryos. $N = 1$ per donor; donor 2 ($n = 29$ pooled embryos); donor 4 ($n = 12$); donor 5 ($n = 15$). In (C), data are represented as mean ± SEM. Paired T test, ns: not significant. $P = 0.2756$ for CD4 +; $P = 0.7084$ for CD8 +; $P = 0.8130$ for NK; $P = 0.0544$ for myeloid; $P = 0.1229$ for B cells; $P = 0.2612$ for others. In (D), data represented the exact value. Chi-square test was performed between the 3 donors, exact $P$-value indicated on graph. ****$P < 0.0001$. (E) Schematic drawing of the process, showing the sequence of I.V injection including engraftment of MDA-MB-231 CFSE labeling cells at E2 stage followed by hu-PBMC micro injection 4 h later. In some experiments, hu-PBMCs were stained with Orange Cell Tracker before intravenous injection. Engrafted embryos were harvested at E4 stage (HH25) for a panel of analyses. (F) Microphotographs of E4 transverse embryonic cryosections showing immunolabeled CD45+ hu-PBMCs (red) from donor 6 infiltrated in CFSE+ MDA-MB-231 tumor mass (green). Chick and human cell nuclei were labeled with Hoechst (blue). The areas surrounded by white squares were magnified. Scale bar: 20 μm. (G) Analysis of intravenously injected hu-PBMCs from 4 healthy donors that infiltrated the somitic territory of tumor cells-grafted and non-grafted embryos. The histogram shows the increased rate of infiltrated immune cells in presence of tumoral cells, compared to equivalent tumor-free somitic tissue. Individual donors are color-coded (dark blue: donor 2, red: donor 4, orange: donor 5, light blue: donor 6). $N = 1$ per donor; donor 2 ($n = 18$ embryos micro-dissected for empty somite and $n = 14$ embryos dissected for tumor site); donor 4 ($n = 18$ and $n = 10$); donor 5 ($n = 11$ and $n = 7$); donor 6 ($n = 4$ and $n = 5$). Data are represented as mean ± SEM. Mann–Whitney test, exact $P$-values indicated on graph. P* $< 0.05$. (H, I) Histograms showing the immunophenotyping of hu-PBMCs infiltrated in MDA-MB-231 tumors for two donors, compared to the pre-grafted sample, for pooled donors (H) and per donor (I). The histograms present the percentage of sub-populations within viable CD45+ cells from pooled embryos. $N = 1$ experiment per donor for prior injection and $N = 3$ experiments per donor for infiltration; Donor 4 ($n = 11$ pooled embryo); donor 5 ($n = 15$ to 18). Data are represented as mean ± SEM for infiltrated conditions in (H) and (I). Chi-square test, exact $P$-values indicated on graphs. ****$P < 0.0001$, ns: not significant. (J) Representative FACS profiles of CD69 and CD25 expressions within the CD3+ CD8+ population of hu-PBMCs infiltrated in MDA-MB-231 tumor site or in equivalent control tissues. Data are presented for 37 pooled embryos intra-venously injected with hu-PBMCs from donor 5. (K) Representative FACS profiles of TIM-3 and PD-1 expressions within the CD3+ CD8+ population of hu-PBMCs infiltrated in MDA-MB-231 tumor site or in equivalent control tissues. Data are presented for 37 pooled embryos intra-venously injected with hu-PBMCs from donor 5. Source data are available online for this figure.

whether the presence of immune cells within the tumor in the ventral somite reflected an infiltration process promoted by cancer cells. To address this question, we analyzed the distribution pattern of hu-PBMCs from 4 donors injected in engrafted embryos. We micro-dissected the tumor and its counterpart contralateral tumor-free somite region to analyze viable human CD45+ cells (Fig. 2G). Strikingly, we found that the average frequency of tumor-resident human CD45+ cells was four times higher in tumor-bearing tissues than in their tumor-free counterparts. For individual donors, the enrichment ranged from 1.8- to 5.5-fold, reflecting donor-dependent variability in efficiency. Thus, tumor cells engrafted in the avian embryonic tissues can attract circulating hu-PBMCs. Thirdly, we analyzed the infiltrated immune component of micro-dissected tumors by flow cytometry (Fig. 2H,I). We pooled the data for two donors and compared the frequencies of infiltrated cell-types with those of the pre-grafted fractions (Fig. 2H). We found significant differences. B cells poorly infiltrated the tumor. In contrast, myeloid cells were largely predominant, as reported for a humanized murine model of engrafted MDA-MB-231 (Rios-Doria et al, 2020). Effective immune cells such as NK, CD8+ and CD4 + T cells were also significantly represented in the immune infiltrate. These cell-type frequencies were similar between donors as no significant difference was found (Fig. 2I). Fourthly, we studied the activation of these infiltrated immune cells. We monitored two activation markers, CD69 and CD25, at the surface of CD8+ and CD4+ T cell populations. Post-activation of T cells, the CD69 antigen is rapidly induced whereas CD25 is a surface marker expressed at a later stage (Caruso et al, 1997). We also monitored two markers of exhaustion, the T-cell immunoglobulin and mucin containing protein-3 (TIM-3) and PD-1 ((Cibrián and Sánchez-Madrid, 2017; Sancho et al, 2005; Yi et al, 2010; Roberts et al, 2021). Tissues with tumors and contralateral controls were micro-dissected out from a batch of 37 embryos and processed for flow cytometry analysis of these different markers. When compared to

control tissues, we found an increase of CD69 marker within both CD4+ and CD8 + T cell population in the tumor infiltrate. The other markers, CD25, PD-1 and TIM-3 markers were expressed at very low levels, in both experimental groups (Figs. 2J,K and EV1A). To examine if T cells exert cytotoxic activity against tumor cells in our infiltration model, we grafted MDA-MB-231 cells in avian embryos and quantified the tumor volumes in embryos that received or not intravenous injection of hu-PBMCs. We found no significant difference between the two conditions, suggesting no T cell cytotoxic activity in this short time frame (Fig. EV1B).

## Development of humanized models with controlled tumor microenvironment

Next, we thought to develop a model bypassing the infiltration process of circulating immune cells that faithfully allows the control and manipulation of the immune cell component within the tumor. We setup a process consisting in the direct co-transplantation of cancer cells and hu-PBMCs (Fig. 3A). These experiments were achieved with 2 donors (donors 6 and 10). To individually visualize both cell contingents, MDA-MB-231 cells and hu-PBMCs were labeled with CFSE and Orange Cell Tracker, respectively. Cells were then mixed at a ratio of 1:1 and the resulting cell suspension was implanted within the ventral somite of batches of embryos for 48 h. To study the influences of cancer cells on hu-PBMCs, we also engrafted hu-PBMCs alone. As expected from our previous findings, we obtained an overall survival of the grafted embryos over 70%. However, given the enriched proportion of immune cells in this restricted territory compared to the infiltration model, we assessed whether this had any deleterious impact on host cells. Cryosections of three immune-cancer cell grafted embryos were prepared and immuno-stained with antibody against cleaved Caspase 3. We quantified the number of immunolabeled host cells in the territory hosting the tumor and in the contralateral non-

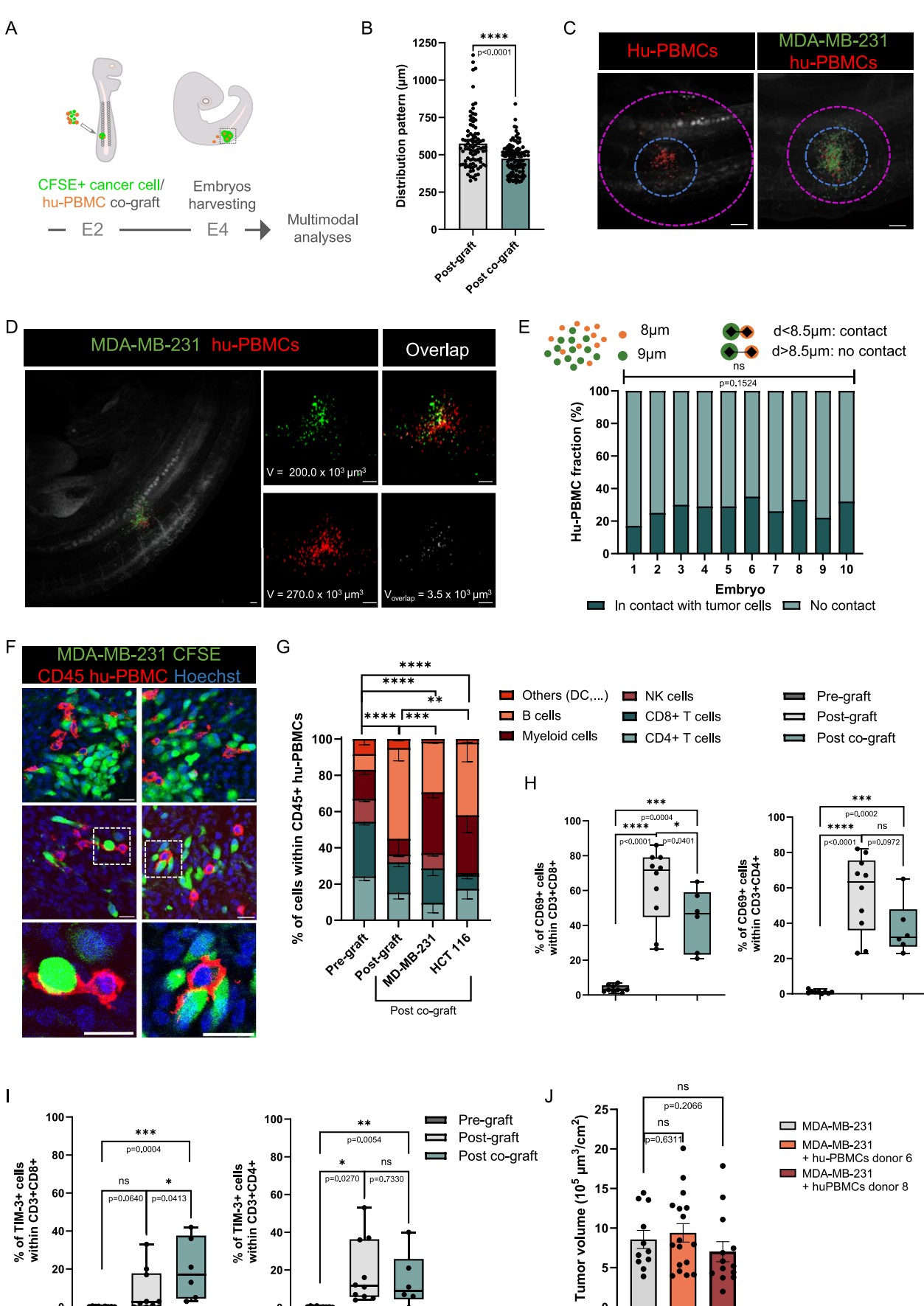

**Figure 3. Cancer cells/hu-PBMCs co-engraftment process establishes a tumor microenvironment.**

(A) Schematic drawing of the humanization process: CFSE-labeled MDA-MB-231 cells were mixed with hu-PBMCs at defined ratio. In some experiments, PBMCs were labeled with Orange Cell Tracker. Mix of cells was micro-injected in the developing somites of chick embryos at E2 (HH14). Engrafted embryos were harvested at E4 (HH25) for a panel of analyses. (B) Histogram depicting the position of the 10 Orange Cell Tracker-labeled hu-PBMCs that were the most distal to the grafting site, per grafted embryo. Hu-PBMCs from donor 10 were co-engrafted with MDA-MB-231 in avian embryo at a ratio 1:1 ($n = 11$ analyzed embryos) or engrafted alone ($n = 9$), $N = 1$. Data are represented as mean ± SEM. Mann–Whitney test, exact $P$-value indicated on graph. ****$P < 0.0001$. (C) Representative light sheet microscopy photographs illustrating the quantitative analysis of distribution pattern. Blue circles represent the grafting site, pink circles represent the area covered by hu-PBMCs. Scale bar: 200 µm. (D) Light sheet microscopy photographs of whole embryos illustrating the co-engraftment process, 48 h post-grafting. The green signal depicts MDA-MB-231 tumor cells, the red signal the hu-PBMCs and the white signal the co-localized fraction. Volumes were calculated using Imaris software (V). Scale bars: 100 µm. (E) Analysis of the respective location of cancer cells and hu-PBMCs. Diameters of 20 Orange Cell Tracker PBMCs (donor 10) and 20 CFSE+ MDA-MB-231 tumor cells were measured per embryo ($n = 10$) with Imaris software to determine an average cell size of each cell types. Schematic showing the calculation of distances between cells. A measure of 8.5 µm, representing the average distance between the centers of mass of two adjacent immune-cancer cells, was used as the threshold for qualifying contact. The histogram presents the fractions of adjacent and non-adjacent immune-cancer cells calculated with Imaris software for 10 engrafted embryos, $N = 1$. Chi-square test, Exact $P$-value indicated on the graph, ns: not significant. (F) Microphotographs of immunolabeled cryosections of E4 chick embryo grafted at E2. MDA-MB-231 cells were detected with CFSE (green) and hu-PBMCs from donor 6 with anti-human CD45 antibody (red). Chick and human cell nuclei are labeled with Hoechst (blue). The images show CD45+ cells in contact with cancer cells, the magnification showing immune cell processes surrounding cancer cells. Scale bars: 20 µm. (G) Histograms showing the hu-PBMC subpopulation fractions for donor 8, between conditions of pre-grafting, grafting alone or grafting in combination with MDA-MB-231 or HCT 116 cells. Embryos were harvested 48 h post grafting. $N = 3$ per experimental group; post graft ($n = 9$ to 17 pooled embryos); co-engrafted with MDA-MB-231 ($n = 9$ to 11); co-engrafted with HCT 116 ($n = 12$ to 18). The histogram of post-graft is also presented in Fig. 1G. Data are represented as mean ± SEM. Chi-square test. ****$P < 0.0001$, ***$P < 0.001$ (exact $P = 0.0002$), **$P < 0.01$ (exact $P = 0.001$). (H, I) Histograms showing the fraction of CD69 (H) and TIM-3 (I) positives cells within both CD3+ CD8+ and CD3+ CD4+ populations, for 3 different donors (1, 6 and 8) and 3 experimental groups representing conditions of pre-grafting, grafting alone and combined with MDA-MB-231 cells. Box-plots represent the median, interquartile range (25th–75th percentiles), with whiskers indicating the minimum and maximum values. Each dot represents pooled embryos that were engrafted with the same donor. $N = 3$ for all donors in pre-graft and post-graft. In post-graft group, $N = 1$ for donor 1 ($n = 6$ to 7 pooled embryos); $N = 4$ for donor 6 ($N = 8$ to 13); $N = 3$ for donor 8 ($n = 9$ to 16). For post co-graft group, $N = 1$ for donor 1 ($n = 51$); $N = 2$ donor 6 ($n = 17$ to 33); $N = 3$ donor 8 ($n = 24$ to 27). Experiments with fewer than 20 CD3+ CD8+ or 20 CD3+ CD4+ cells were excluded from analysis. Unpaired T test was performed between all pre-graft and post-graft groups except for CD69 in CD3+ CD4+ population. Mann–Whitney test was performed in all other comparisons, exact $P$-values indicated on graphs. ****$P < 0.0001$***$P < 0.001$, **$P < 0.01$, *$P < 0.05$, ns: not significant. (J) Histograms showing the quantification of the tumor volumes of MDA-MB-231 engrafted alone or co-engrafted with hu-PBMCs from donor 6 or 8. For each embryo, the volume was normalized to the BSA. Dots represent individual embryos. $N = 1$ experiment per group ($n = 11$ embryos in engrafted group; $n = 17$ in co-engraftment with donor 6, and $n = 13$ in co-engraftment with donor 8). Data are represented as mean ± SEM. Unpaired T-test (donor 6) or Mann–Whitney test (donor 8). Exact $P$-values indicated on graph. ns: not significant. Source data are available online for this figure.

grafted equivalent territory. We found no significant difference, thus indicating a low allogeneic reaction against the host (Fig. EV1C,D).

Then, we imaged the single or dual fluorescent signal within grafted embryos by light sheet microscopy. We developed a method to evaluate the distribution pattern of hu-PBMCs in presence and absence of cancer cells. It consisted in measuring first the distance covered by the 10 hu-PBMCs showing the highest spreading from the injection site and second determining the average distance covered by all hu-PBMCs. We compared the measures obtained for single and co-implantation conditions and found that the presence of cancer cells significantly reduced the spreading of hu-PBMCs (Figs. 3B,C and EV1E).

In co-implanted embryos, the global pattern of red and green signals was largely overlapping suggesting possible interactions between immune and cancer cells (Fig. 3D). To further assess this possibility, we evaluated the physical proximity between cells. We calculated the average diameter of cancer and immune cell populations in series of light sheet microscopy images. We then measured the distance between mass centers of both cell types, considering that measures below this distance depict contact between cells (Fig. 3E). We found an average of 28% of cells showing such physical proximity among the entire immune and cancer cell population. Statistical analysis showed that this proportion remains constant between embryos. We then imaged this intermingled immune/cancer cell microenvironment at higher resolution, by immunolabeling CD45 marker in embryonic sections and confocal microscopy. We frequently observed hu-PBMCs with complex morphologies surrounding or coming in contact with tumoral cells (Fig. 3F). Altogether this imaging analysis suggested

that active communications might occur within the microenvironment, that trap hu-PBMCs within the tumor and bring them in close contact to cancer cells.

In the next step, to more closely study these immune-tumor cell interactions, we analyzed the activation of T cells. As cancer cells, we chose two cell lines, HCT 116 and MDA-MB-231, respectively representing colorectal and triple-negative breast cancers, already used in mouse models for immuno-oncology studies (Capasso et al, 2019; Wang et al, 2018; Shang et al, 2022). We selected an effector:target ratio of 4:10 that represents an intermediate infiltration level for CRC and TNBC patient tumors (Leon-Ferre et al, 2024). We transplanted hu-PBMCs from donor 8 in the ventral somite either alone, or combined with colorectal HCT 116 or breast cancer MDA-MB-231 cells. In order to assess the impact of cancer cells, we studied by flow cytometry the frequencies of immune cell populations in tissues micro-dissected from embryos engrafted with hu-PBMCs, alone, and combined with TNBC or CRC cancer cells (Fig. 3G). For both cancer types, we found that the presence of cancer cells significantly impacted on the immune cell populations frequencies. Changes were reflected in reduction of the B cell fraction and expansion of the myeloid fraction. The T cell population was moderately modified. We also assessed if TNBC and CRC cancer cells had comparable impact. We found that changes of immune cell type frequencies were more pronounced for MDA-MB-231 than for HCT 116 cells. As an example, the NK population shrank to 1% average frequency in presence of HCT 116 cells, while it represented 8% of the whole immune infiltrate in presence of MDA-MB-231 cells. We repeated these experiments with 2 other donors (donor 1 and donor 6) and globally found comparable outcomes, suggesting that the

differences are driven by cancer cell properties rather than by immune cells (Fig. EV1F,G).

Next, using our model of co-implantation of hu-PBMCs/MDA-MB-231, we investigated T cell activity for 3 different hu-PBMC donors (donors 1, 6 and 8). CD4+ and CD8+ cells were immunolabeled to detect the early activation marker, CD69, and the marker of exhaustion, TIM-3 (Fig. 3H,I). We compared T cell activation in 3 experimental conditions: pre-grafted hu-PBMCs, hu-PBMCs in single implantation, PBMCs/MDA-MB-231 co-implantation. In analysis compiling data for the 3 donors, we observed as expected an activation of both CD4+ and CD8+ populations 48 h post-implantation of hu-PBMCs, with a drastic increase in the CD69+ cell count, compared to the pre-grafted condition. Interestingly, we observed a mirrored decrease in CD69+ cell numbers and an increase in TIM-3+ cell numbers within the CD8+ and CD4+ T cell populations in presence of cancer cells, compared to PBMCs-only implantation. This suggests a rapid suppression of T cell activation by tumoral cells. Analysis of individual donors showed inter-donor variability. For example, for 2 of 3 donors, TIM-3+ cell number increased in the grafted condition, but dropped for the remaining donor (Fig. EV1H,I). Altogether, the analyses suggested that our humanized avian model (humanized AVI-CellDX) allows the reconstitution of a tumor microenvironment, recapitulating an inhibition of T cell functions by cancer cells. Finally, we assessed if the co-implantation of hu-PBMCs has a functional impact on the cancer cells. Hu-PBMCs from 2 healthy donors, 6 and 8, were co-grafted with MDA-MB-231 cells and the tumor volumes were compared to those in embryos grafted with MDA-MB-231 cells only. We found that the presence of hu-PBMCs had no significant effect on the tumor volume, although for one donor, a trend toward a decrease was observed (Fig. 3J).

## Pembrolizumab anti-PD-1 antibody proves efficacy in the humanized-AVI-CellDX models of solid cancers

To study the relevance of our preclinical model, we investigated whether it could reveal the efficacy of a standard immunotherapy, such as the anti-PD-1 antibody, pembrolizumab, that targets the PD-1/PD-L1 axis. We chose as cancer indications CRC and TNBC. For CRC, profiles of genomic features like micro-satellite stability (MSS) or instability-high (MSI-H) status and deficient mismatch repair (dMMR) are used to determine patient eligibility to anti-PD-1 treatment (Gorzo et al, 2022; Cervantes et al, 2024). The HCT 116 cell line having a MSI status is representative of responder CRCs. We also found that these cells can express high PD-L1 levels (Fig. EV2A). For TNBC, anti-PD-1 efficacy was proven dependent on PD-L1 levels (Villacampa et al, 2022). We selected as representative TNBC cancer cell lines: MDA-MB-231 and MDA-MB-436, which we found express high and low PD-L1 levels, respectively (Fig. EV2A, Dong et al, 2019).

First, we applied a protocol established in our previous work to determine the optimal dose of administration in the avian embryo model. We performed intravenous injection of doses of pembrolizumab ranging from pure solution to 5-fold dilution (890 mg/kg, 178 mg/kg, 35.6 mg/kg) in batches of avian embryos. Control embryos were injected with pembrolizumab excipient, NaCl (0.9%). As achieved for hu-PBMCs implantation and intravenous injection, we also examined morphological features to evaluate the outcome

on embryonic development. We observed a systematic high survival rate, above 80%, for all tested concentrations (Fig. 4A). The BSA was constant between experimental groups. Nevertheless, for 1 over 15 embryos, we noted the appearance of morphogenesis defects at the highest tested dose (890 mg/kg), affecting the eye and craniofacial structures (Fig. EV2B). Based on these results, we selected the 175 mg/kg as a working dose.

Next, we investigated whether intravenous administration of pembrolizumab enables efficient targeting of the T-cell surface PD-1 receptor. We used our established method for evaluating therapy efficacy, consisting in, first, engrafting cells in batches of embryos at E2, second, intravenously injecting the therapeutic agent and control excipient 24 h later, and, third harvesting the embryos for analysis 48 h post-implantation (Jarrosson et al, 2021; 2023; Zala et al, 2024) (Fig. 4B).

Firstly, in experiments conducted with 6 donors, we implanted hu-PBMCs alone in the ventral somite of batches of embryos, that received either pembrolizumab or control excipient. We analyzed T cells from micro-dissected ventral somite tissues containing the implanted hu-PBMCs by flow cytometry to determine whether pembrolizumab administration interfered with PD-1 immunodetection (Fig. 4C). By combining the data from the 6 donors, we found a significant 50% decrease of detected PD-1+ cells, when compared to the experimental group that received excipient injection. Analysis of individual donors showed some differences with a decrease in PD-1+ cell number ranging from 36% to 58%. This demonstrated that injected pembrolizumab properly targeted PD-1 expressed by T cell within implanted hu-PBMCs.

Secondly, in 2 independent experiments, we co-implanted allogeneic hu-PBMCs from 3 donors with CFSE-labeled MDA-MB-231 cells (Fig. 4D). As achieved for single PBMC implantation, we analyzed immunodetected PD-1+ cells and found a global significant decrease in the pembrolizumab condition, compared with control, consistent with a competition of therapeutic and immunodetecting antibodies. We next analyzed inter-donor and experiment variabilities. We observed that PD-1+ cell numbers varied between donors. Apart for one replicate from one donor, pembrolizumab administration consistently reduced PD-1+ cell numbers. We assessed if reduction of PD-1 levels could be reinforced by administration of higher pembrolizumab dose. We observed no benefit of increasing the dose from 178 to 890 mg/kg (Fig. EV2C).

Next, we investigated the outcome of anti-PD-1 administration on T cell activation. We used 3 donors, co-transplanted with MDA-MB-231 cells. Tumors were micro-dissected out 24 h post-treatment with pembrolizumab or excipient. We monitored CD69 and CD25 at the surface of CD8+ and CD4+ T cell population. Interestingly in the pembrolizumab condition, for all 3 donors, CD69+ cell numbers significantly increased in CD8+ T cell population but only slightly in CD4+ cell population, compared to the excipient (Figs. 4E,F and EV2D,E). We also found that pembrolizumab administration tended to increase CD25+ cell numbers in the CD8 + T cell population, consistent with T cell activation, while it was not observed for the CD4+ T cell population (Figs. 4F and EV2D–F).

We then quantified TIM-3 expression within the CD3+ CD69+ population, representing activated T cells (Saleh et al, 2019) (Figs. 4G,H and EV2G,H). We found for all 3 donors that TIM-3+ cell numbers were significantly decreased by pembrolizumab,

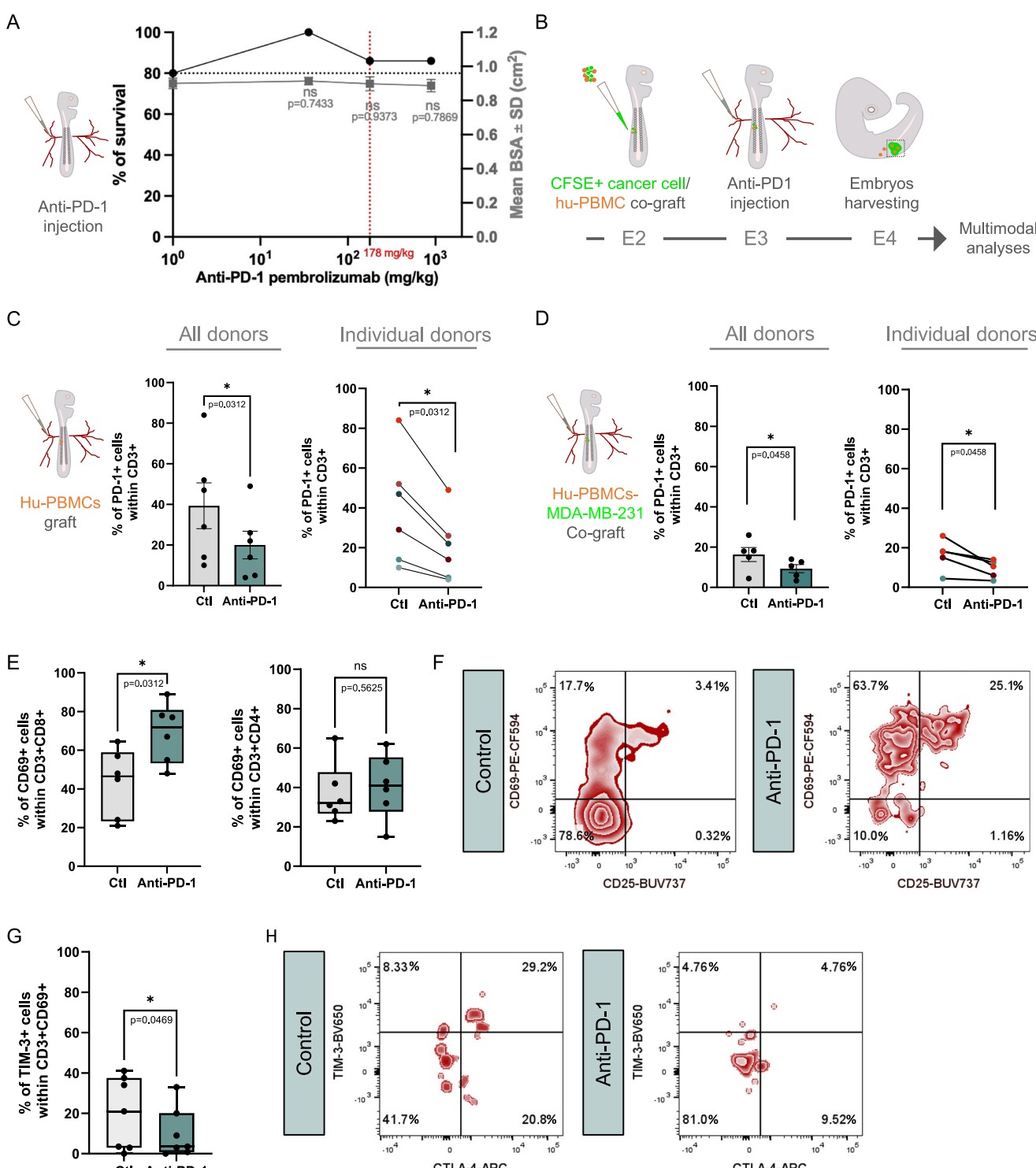

compared to control. Furthermore, in an additional experiment conducted on 1 donor, a similar profile was observed for another exhaustion marker, cytotoxic T-lymphocyte-associated protein 4 (CTLA4, Fig. 4H). Altogether, these data suggested that pembrolizumab is capable of restoring CD8+ T cells activity in the humanized avian embryo model.

We thus studied if these properties could sustain an anti-tumoral effect of pembrolizumab, in a similar short range of time. First, in models of MDA-MB-231 cell line co-transplanted with hu-PMBCs from donor 8, we assessed the efficacy of various doses of pembrolizumab, ranging from 35.6 to 890 mg/kg (Fig. 5A,B). Pembrolizumab and excipient were intravenously injected 24 h post

**Figure 4. Anti-PD-1 administration to humanized avian embryos affects the activity of hu-PBMCs.**

(A) Analysis of survival rate (left axis) and mean body surface area (BSA, right axis) of chick embryos injected with increasing doses of anti-PD-1 pembrolizumab. Each doses and control excipient (Ctl, NaCl 0.9%) were injected to 15 embryos, in $N = 1$. Data are represented as mean ± SD. Unpaired T-test compared to excipient, exact $P$-values indicated on graphs, ns: non-significant. (B) Schematic drawing of the humanization process with treatment injection. CFSE-labeled MDA-MB-231 or HCT 116 cells were mixed at effector:target ratio cells of 4:10 with hu-PBMCs. The mix was micro-injected in the developing somites at E2 (HH14). Pembrolizumab at 178 mg/kg or NaCl 0.9% was injected intra-venously 24 h post-grafting. Grafted embryos were harvested at E4 stage (HH25) for a panel of analyses. (C, D) Histograms depicting the fraction of PD-1 positive cells at the surface of CD3+ T cells, 24 h post-treatment with anti-PD-1 (pembrolizumab) or control (NaCl) of embryos grafted either with hu-PBMCs (C) or the mix tumor cells/hu-PBMCs (D). The experiments were conducted with 6 color-coded donors. For graph (C), data are represented as mean ± SEM or for individual donors (dark red: donor 1, red: donor 2, light blue: 4, orange: donor 6, blue: donor 8, dark blue: donor 9). $N = 1$ experiment per donor, donor 1 ($n = 18$ pooled embryos per experimental group); donor 2 ($n = 18$); donor 4 ($n = 9$); donor 6 ($n = 17$); donor 8 ($n = 3$ for control and $n = 9$ for pembrolizumab); donor 9 ($n = 8$). Wilcoxon test, exact $P$-value indicated on graph. *$P < 0.05$. For graph (D), data are represented as mean ± SEM or for individual donors (dark red: donor 1, orange: donor 6 and blue: donor 8). $N = 1$ for donor 1 ($n = 51$ pooled embryos in control group; $n = 54$ pooled embryos in pembrolizumab group); $N = 2$ for donor 6 ($n = 14$ to 17; $n = 16$ to 27); $N = 2$ for donor 8 ($n = 24$ to 27 $n = 19$ to 23). Paired T test, exact $P$-value indicated on graph. *$P < 0.05$. (E) Histogram showing the fraction of CD69 positive cells in CD3+CD8+ or CD3+CD4+ populations in embryos grafted with MDA-MB-231 cells and hu-PBMCs, treated with anti-PD-1 (pembrolizumab) or control (NaCl 0.9%). Box-plots represent the median, interquartile range (25th–75th percentiles), with whiskers indicating the minimum and maximum values. Each dot represents an independent experiment performed with 3 donors (donors 1, 6 and 8). $N = 1$ for donor 1 ($n = 51$ pooled embryos in control group; $n = 54$ pooled embryos in pembrolizumab group); $N = 2$ for donor 6 ($n = 17$ to 32 and $n = 27$ to 36); $N = 3$ for donor 8 ($n = 24$ to 27 and $n = 19$ to 23). Experiments with fewer than 20 CD3+CD8+ or 20 CD3+CD4+ cells were excluded from analysis. Wilcoxon test, exact $P$-values indicated on the graphs. *$P < 0.05$, ns: not significant. (F) Representative FACS profiles of CD69 and CD25 expressions within the CD3+CD8+ population of hu-PBMCs co-grafted with MDA-MB-231 cells in embryos treated either with control (NaCl 0.9%) or with anti-PD-1 (pembrolizumab). Data are presented for pooled embryos engrafted with donor 8 ($n = 24$ pooled embryos for control; $n = 19$ pooled embryos for pembrolizumab). (G) Histogram showing the fraction of TIM-3 positive cells within the CD3+CD69+ population of hu-PBMCs co-grafted with MDA-MB-231 cells in embryos treated either with anti-PD-1 (pembrolizumab) or control (NaCl 0.9%). Box-plots represent the median, interquartile range (25th–75th percentiles), with whiskers indicating the minimum and maximum values. Each dot represents pooled embryos for 3 donors (donors 1, 6 and 8). $N = 1$ for donor 1 ($n = 51$ pooled embryos in control group; $n = 54$ pooled embryos in pembrolizumab group); $N = 3$ for donor 6 ($n = 17$ to 32 and $n = 27$ to 36); $N = 3$ for donor 8 ($n = 24$ to 27 and $n = 19$ to 23). Experiments with fewer than 20 CD3+CD69+ cells were excluded from analysis. Wilcoxon test, exact $P$-value indicated on the graph. *$P < 0.05$. (H) Representative FACS profiles of TIM-3 and CTLA-4 expressions within the CD3+CD69+ population of hu-PBMCs grafted with MDA-MB-231 cells in embryos treated either with control (NaCl 0.9%) or with anti-PD-1 (pembrolizumab). Data are presented for pooled embryos engrafted with donor 6 ($n = 20$ pooled embryos for control; $n = 25$ pooled embryos for pembrolizumab). Source data are available online for this figure.

co-implantation. 24 h later, embryos were harvested. Embryos were imaged using light sheet microscopy and images were processed to measure tumor volumes as in Jarrosson et al (2021, 2023). When compared with the excipient-treated experimental group, the tumor volume was unchanged for the lowest 35.6 mg/kg dose. In contrast, we found that the two higher doses, 178 and 890 mg/kg, resulted in significant tumor reduction, by 26% and 25%, respectively. Thus, the 178 mg/kg dose which we found to reactivate the T cells was efficient to induce a functional outcome in MDA-MB-231 cells, consistent with their reported sensitivity to anti-PD-1. Next, we studied if this anti-tumoral response was mediated by hu-PBMCs, as expected. We analyzed the tumor volume in embryos engrafted with MDA-MB-231 cells only and found that a 178 mg/kg pembrolizumab injection had no effect on the tumor volume (Fig. EV3A). Then we thought to demonstrate that anti-PD-1 efficacy results from recognition of tumoral cells by T cells via MHC-I antigen presentation. MDA-MB-231 and hu-PMBCs from the same donor (donor 8) were engrafted in batches of embryos either directly or after incubation with anti-MHC-I antibody (50 µg/ml). Within each group, grafted embryos received either the excipient (control) or the pembrolizumab by intravenous injection. One day post-injection, the tumor volumes were analyzed. As observed in the previous experiment, pembrolizumab administration induced significant tumor volume reduction, compared to the excipient. Interestingly, this pembrolizumab effect was fully abolished by pre-incubation with anti-MHC-I, proving the involvement of T cells in the tumor volume reduction (Fig. 5C).

Next, using the same donor (donor 8) and dose, we assessed the efficacy of pembrolizumab injection in models of MDA-MB-436 and HCT 116. We observed that it induced a 35% reduction of tumor volume for HCT 116 cell line while having no effect for MDA-MB-436 cell line (Fig. 5D). We also tested if the outcome of

pembrolizumab injection on MDA-MB-436 cell line could be improved with another hu-PBMC donor, donor 1 and found similar lack of response of the cancer cells. Similarly, we confirmed the response of HCT 116 cell line to pembrolizumab using a second donor, donor 6 (Fig. EV3B). Thus, these results were consistent with the reported sensitivity and resistance of HCT 116 and MDA-MB-436 cell lines, respectively.

Then in MDA-MB-231 cell line model, we further assessed the inter-donor variability of hu-PBMCs, comparing pembrolizumab efficacy in co-transplantation experiments achieved with 5 different donors (Fig. 5E). Expectedly, we found that anti-PD-1 treatment efficacy varied between donors. Beyond donor-specific characteristics related to cell populations representativity and activity states, we also thought to examine whether variations could come from mismatch between haplotypes of tumor and immune cells. Focusing on HLA-A2, expressed by 50% of the Caucasian population, we found that the most competent donors (1, 8, 9) were HLA-A2 positive, matching with HLA-A2 positive MDA-MB-231 cells, while donors showing lower efficiency (6 and 2) were HLA-A2 negative (Fig. EV3C).

Next, we studied whether we could document the effect of pembrolizumab administration on cancer cell apoptosis. We focused on the AVI model with MDA-MB-231 cells co-grafted with hu-PBMCs from donor 6, and performed immunolabeling of cleaved Caspase 3 in cryosections of pembrolizumab-treated and control embryos ($n = 3$ embryos per condition). We counted the number of cleaved Caspase 3+ cells within CFSE+ cells. Consistent with the observed reduction in tumor volumes, we found that pembrolizumab administration induced a significant increase in cleaved Caspase 3 signal (Fig. 5F,G). Thus altogether, these findings showed that humanizing avian embryos with PBMCs allows to study anti-PD-1 efficacy on solid tumors.

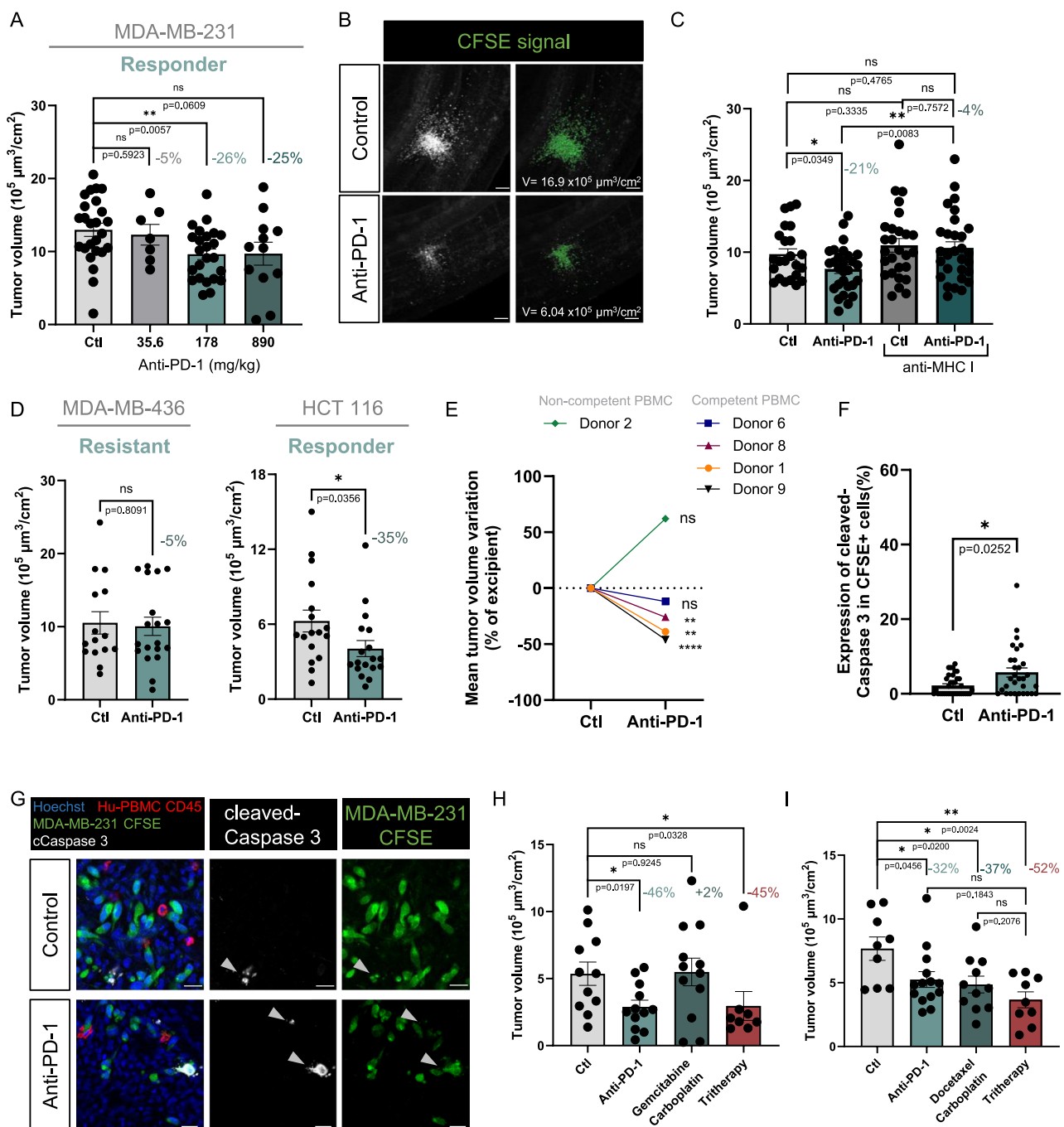

PD-1/PD-L1 therapies have transformed cancer treatment but face growing resistance, prompting a shift toward combination strategies to achieve stronger and longer-lasting responses (Yi et al, 2022; Mandal et al, 2025). We thus examined if we could engineer combination therapies in our model. We addressed this question with the MDA-MB-231 AVI model and donor 8, a combination that we found effective to reveal pembrolizumab-mediated anti-tumoral effect. In the clinic, anti-PD-1 has been combined with several chemotherapies, such as docetaxel/carboplatin and gemcitabine/carboplatin, to treat metastatic TNBC patients (Cortes et al, 2020; Sharma et al, 2024). The MDA-MB-231 cell line was reported

sensitive to docetaxel/carboplatin, while being rather poor responder to gemcitabine treatment (Chen et al, 2014; Di et al, 2018). We thus set-up two tri-therapeutic treatments, combining pembrolizumab with either docetaxel/carboplatin or with gemcitabine/carboplatin and compared their effect to those of anti-PD-1 monotherapy and of bi-chemotherapy.

The doses of administration were chosen from previous work for gemcitabine/carboplatin (Jarrosson et al, 2021) and based on a dose escalation assay performed for docetaxel (Fig. EV3D). We used the following doses: 1.4 mg/kg docetaxel, 92.3 mg/kg carboplatin, 0.68 mg/kg gemcitabine, and 178 mg/kg anti-PD-1. We

**Figure 5.   Anti-PD-1 administration shows anti-tumoral effect in humanized avian embryos grafted with cancer cell lines.**

(A) Histograms showing the quantification of the tumor volumes of MDA-MB-231 cells co-engrafted with hu-PBMCs from donor 8. Embryos were treated either with control (NaCl 0.9%) or with 3 increasing doses of pembrolizumab (35.6, 178, 890 mg/kg). Dots represent volumes normalized to each embryo's BSA. $N = 1$ ($n = 25$ embryos in control group, $n = 7$ in 35.6 mg/kg dose, $n = 25$ in 178 mg/kg dose and $n = 12$ in 890 mg/kg dose). Data are represented as mean ± SEM. The percentage represents the average volume reduction per experimental group. Unpaired T test for the two highest doses and Mann–Whitney test for 35.6 mg/kg dose. Exact P-values indicated on graph. **$P < 0.01$, ns: not significant. (B) Representative light sheet microscopy 3D images of tumors in E4 chick embryo co-engrafted with MDA-MB-231 cells and hu-PBMCs from donor 8, in pembrolizumab (178 mg/kg) and control conditions. The right panels illustrate the extraction of the fluorescent signal for measuring tumor volume (V) with Imaris software (in green). Scale bars: 150 μm. (C) Histograms showing the quantification of the tumor volumes of MDA-MB-231 cells co-engrafted with hu-PBMCs from donor 8, pre-incubated or not with anti-MHC I antibodies at 20 μg/mL. For each group, embryos were treated either with control (NaCl 0.9%) or with anti-PD-1 (pembrolizumab) at 178 mg/kg. Dots represent volumes normalized to each embryo's BSA. $N = 1$ experiment ($n = 23$ embryos in control group, $n = 28$ in anti-PD-1 group, $n = 27$ in control group with anti-MHC I and $n = 29$ in anti-PD-1 group with anti-MHC I). Data are represented as mean ± SEM. The percentage represents the average volume reduction between treated and control for each group. Statistical analyses were conducted with Mann–Whitney when comparing to control group with anti-MHC I, and Unpaired T test when comparing to control group or anti-PD-1 groups. Exact P-values indicated on graph. *$P < 0.05$, **$P < 0.01$, ns: not significant. (D) Histograms showing the quantification of the tumor volumes of MDA-MB-436 and HCT 116 cells co-engrafted with hu-PBMCs from donor 8. Embryos were treated either with control (NaCl 0.9%) or anti-PD-1 (pembrolizumab). Dots represent volumes normalized to each embryo's BSA. $N = 1$ experiment per cell line for MDA-MB-436 ($n = 15$ in control group and $n = 19$ in pembrolizumab group); for HCT 116 ($n = 17$ and $n = 18$). Data are represented as mean ± SEM. The percentage represents the average volume reduction. Statistical analyses were conducted with Unpaired T-test for MDA-MB-436, Mann–Whitney for HCT 116. Exact P-values indicated on graphs. *$P < 0.05$, ns: not significant. (E) Graph showing the variation of average volume reduction of MDA-MB-231 tumors co-grafted with hu-PBMCs from different donors in embryos treated with anti-PD-1 (pembrolizumab) or control (NaCl 0.9%). $N = 1$ for donor 1 ($n = 13$ embryos in control group and $n = 19$ embryos in pembrolizumab group); $N = 1$ for donor 2 ($n = 9$ and $n = 9$); $N = 1$ donor 6 ($n = 17$ and $n = 21$); $N = 1$ for donor 8 ($n = 25$ and $n = 24$); $N = 1$ for donor 9 ($n = 19$ and $n = 17$). Donors are color-coded (orange round: donor 1, green diamond: donor 2, blue square: donor 6, red upward triangle: donor 8, black downward triangle: donor 9). Statistical analysis was done with Mann–Whitney (donor 6) or Unpaired T test (donor 1, 2, 8 and 9). **$P < 0.01$ ($P = 0.0016$ for donor 1 and $P = 0.0057$ for donor 8), ****$P < 0.0001$ (donor 9), ns: not significant ($P = 0.916$ for donor 2 and $P = 0.3631$ for donor 6). (F) Histogram showing the fraction of CSFE+ Caspase 3+ cells in tumors of MDA-MB-231 cells co-grafted with hu-PBMCs from donor 6 in embryos treated with anti-PD-1 (pembrolizumab) and control. $n = 35$ sections for control from 3 embryos; $n = 30$ sections for pembrolizumab condition from 3 embryos. Data are represented as mean ± SEM. Mann–Whitney test, exact P-value indicated on the graph. *$P < 0.05$. (G) Microphotographs illustrating immunofluorescent labeling of chick embryo cryosections, 48 h after co-grafting of CFSE+ MDA-MB-231 cells and hu-PBMCs from donor 6 and treatment with anti-PD-1 (pembrolizumab) or control. Immune cells were labeled with anti-human CD45 antibody (red), apoptotic cells with anti-human cleaved Caspase 3 antibody (white). Avian and human nuclei were stained with Hoechst (blue). White arrow points CFSE+ Caspase 3+ cells. Scale bars: 20 μm. (H, I) Histograms showing the quantification of the tumor volumes of MDA-MB-231 cells co-grafted with hu-PBMCs from donor 8 and treated either with control, anti-PD-1 (pembrolizumab at 178 mg/kg), combination of 2 chemotherapies or combination of the three treatments (tritherapies). In (H), embryos were treated with gemcitabine-carboplatin (0.68 mg/kg and 92.3 mg/kg, respectively) or tritherapy (combination of pembrolizumab-gemcitabine-carboplatin). $N = 1$ ($n = 11$ embryos in control group, $n = 12$ in anti-PD-1, $n = 12$ in gemcitabine-carboplatin and $n = 8$ in tritherapy). In (I), embryos were treated with docetaxel-carboplatin (1.4 mg/kg and 92.3 mg/kg, respectively) or the tritherapy (combination of pembrolizumab-docetaxel-carboplatin). $N = 1$ ($n = 9$ embryos in control group, $n = 14$ in anti-PD-1, $n = 11$ in docetaxel-carboplatin and $n = 9$ in tritherapy). Dots represent volumes normalized to each embryo's BSA. Data are represented as mean ± SEM. The percentage represents the average volume reduction compared to control. Statistical analyses were conducted with Mann–Whitney (tritherapy group for histogram (H); anti-PD-1 group for (I)) or Unpaired T test (anti-PD-1 and chemotherapies groups for (H); tritherapy and chemotherapies groups for (I)). Exact P-values indicated on graphs. *$P < 0.05$, **$P < 0.01$, ns: not significant. Source data are available online for this figure.

found that tumor volumes were unaffected by gemcitabine/carboplatin treatment, likely because MDA-MB-231 cells responded poorly to gemcitabine. This suggests that combining it with carboplatin does not overcome this resistance. As observed in our previous experiments, pembrolizumab induced a significant reduction in tumor volumes. This efficacy was preserved in co-administration with gemcitabine/carboplatin, with no additional benefit from the combination therapy (Fig. 5H). For the second combination therapy, we found that both the individual administration of docetaxel/carboplatin and pembrolizumab resulted in a significant decrease in tumor volumes compared to the control group. Interestingly, in this case, co-administration tended to be even more effective than the individual treatments (Fig. 5I).

## Recapitulating patient response heterogeneity to anti-PD-1 with humanized-AVI-PDX model

We showed in our previous works on follicular lymphoma and metastatic melanoma that miniature replicas of patient tumors in avian embryos receiving standard therapies, manifested patient-dependent responses that matched those observed in the clinic (Jarrosson et al, 2023; Zala et al, 2024). Focusing on CRC, we thus investigated if the heterogeneity of response to anti-PD-1 could be recapitulated in the humanized-AVI-PDX model. We analyzed the effect of pembrolizumab on replicas of patient tumors having different molecular profiles: MSI, predicting sensitivity to anti-PD-1 and MSS, predicting resistance to anti-PD-1. Intriguingly, although considered of low predictive value and not included in the therapeutic decision tree, a MSS patient with high PD-L1 expression (85%) was reported with sensitivity to anti-PD-1 (Gomar et al, 2021). We thus included PD-L1 levels as an additional marker reporting inter-patient heterogeneity.

Six tumor samples (2 MSI and 4 MSS) were dissociated. A fraction of the suspension was harvested to evaluate PD-L1 levels by flow cytometry (Fig. 6D). For each sample, patient cells were labeled with CFSE and mixed with hu-PBMCs from donor 8, selected for its efficiency in our previous experiments with cell lines, at a ratio of effector:target of 4:10. The mix was implanted into the ventral somite of batches of embryos. As the number of cells varied between samples, we selected for each patient a cellular concentration allowing to engraft at least 20 embryos. Some embryos engrafted with the CRC-03 sample were collected for immunolabeling of CD45 in transverse sections, which confirmed the presence of hu-PBMCs in the tumors (Fig. 6A). We performed intravenous injection of pembrolizumab or excipient 24 h post-grafting. 24 h post-treatment, embryos were harvested for 3D imaging and tumor volume measurement (Fig. 6B–E). Interestingly, regardless of PD-L1 expression, we found that pembrolizumab induced a 37% and 48% tumor volume reduction for the 2 patient samples with MSI status (CRC-01 and CRC-02). Although

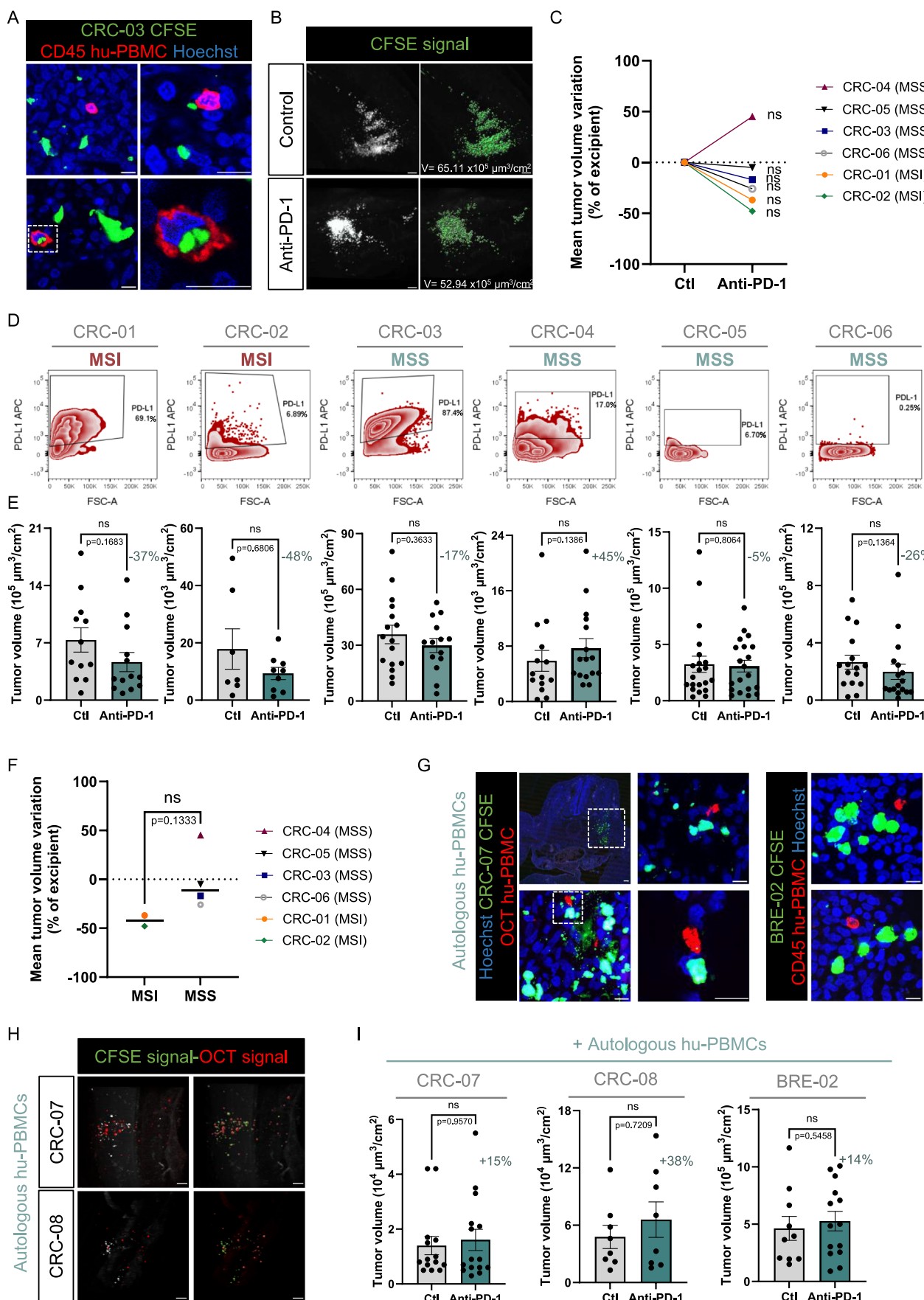

**Figure 6. Anti-PD-1 administration shows anti-tumoral effect in humanized patient-derived xenografted avian embryos.**

(A) Microphotographs illustrating immunofluorescent labeling of chick embryo cryosections, 48 h after co-grafting of CFSE+ CRC-03 tumoral cells (green) and hu-PBMCs from donor 8. Immune cells were labeled with anti-human CD45 antibody (red). Chick and human cell nuclei were labeled with Hoechst (blue). The area surrounded by a white square was magnified. Scale bars: 10 μm. (B) Representative light sheet microscopy photographs illustrating CFSE+ tumors imaged in E4 chick embryos co-grafted with CRC-03 patient sample and hu-PBMCs from donor 8, treated with anti-PD-1 (pembrolizumab) or control (NaCl 0.9%). The right panels show the fluorescent capture of the CFSE fluorescent signal for measure of tumor volume (V) with Imaris software (in green). Scale bars: 100 μm. (C) Histogram showing the variation of mean tumor volumes in embryos treated with anti-PD-1 (pembrolizumab) or control (NaCl 0.9%). Each dot represents a patient sample (orange round: CRC-01, green diamond: CRC-02, blue square: CRC-03, red upward triangle: CRC-04, black downward triangle: CRC-05, gray circle: CRC-06). Statistics show tumor volume reduction compared to control represented in Fig. 6E. ns: not significant. (D) Representative FACS profiles of PD-L1 expression in colorectal cancer patient samples (CRC-01, CRC-02, CRC-03, CRC-04, CRC-05, CRC-06) and their microsatellite stability status (MSI or MSS). (E) Histograms showing the quantification of tumor volumes in embryos co-grafted with CRC patient cells and hu-PBMCs from donor 8, treated with anti-PD-1 (pembrolizumab) or control (NaCl 0.9%). $N = 1$ for all sample, for CRC-01 ($n = 12$ embryos in control group and $n = 12$ embryos in pembrolizumab group); for CRC-02 ($n = 7$ and $n = 9$); for CRC-03 ($n = 16$ and $n = 14$); for CRC-04 ($n = 14$ and $n = 13$); for CRC-05 ($n = 21$ and $n = 20$); for CRC-06 ($n = 16$ and $n = 17$). Dots represent volumes normalized to each embryo's BSA. Data are represented as mean ± SEM. The percentage represents the average volume reduction compared to control. Statistical analyses were done using Unpaired T test (CRC-03) or Mann–Whitney (CRC-01, CRC-02, CRC-04, CRC-05 and CRC-06). Exact P-values indicated on graphs. ns: not significant. (F) Histogram showing the variation of mean tumor volumes in embryos treated with anti-PD-1 (pembrolizumab) or control per microsatellite stability status of CRC sample. Each dot represents a patient sample (orange round: CRC-01, green diamond: CRC-02, blue square: CRC-03, red upward triangle: CRC-04, black downward triangle: CRC-05, gray circle: CRC-06). Mann–Whitney test, exact P-value indicated on the graph. ns: not significant. (G) Microphotographs illustrating immunofluorescent labeling of chick embryo cryosections, 48 h after co-grafting of CFSE+ CRC-07 or BRE-02 tumoral cells (green) and their autologous hu-PBMCs (red). For CRC-07, hu-PBMCs were labeled with OCT before co-grafting in avian model. For BRE-02, cells were labeled with anti-human CD45 antibody. Chick and human cell nuclei were labeled with Hoechst (blue). The areas surrounded by white squares were magnified. Scale bars: 50 μm for x10 objective microphotograph, 10 μm for the 3 others. (H) Representative light sheet microscopy photographs illustrating CFSE+ tumors from colorectal patient samples (CRC-07 and CRC-08) co-engrafted with their Orange Cell Tracker+ autologous hu-PMBCs, in E4 chick embryos. The right panels show the fluorescent capture of the CFSE (in green) and OCT (in red) fluorescent signals with Imaris software. Scale bars: 100 μm. (I) Histograms showing the quantification of tumor volumes in embryos co-grafted with CRC patient cells or TNBC patient cells with their autologous hu-PBMCs, and treated with anti-PD-1 (pembrolizumab) or control (NaCl 0.9%). $N = 1$ for CRC-07 ($n = 14$ embryos in control group and $n = 15$ embryos in pembrolizumab group); $N = 1$ CRC-08 ($n = 8$ in both group); $N = 1$ BRE-02 ($n = 10$ and $n = 14$). Dots represent volumes normalized to each embryo's BSA. Data are represented as mean ± SEM. The percentages represent the average volume reduction of tumors compared to control. Statistical analyses were done using Unpaired T test (BRE-02) or Mann–Whitney (CRC-07 and CRC-08). Exact P-values indicated on graphs. ns: not significant. Source data are available online for this figure.

the difference was not statistically significant when compared with excipient condition, this tendency was fully consistent with clinical observations. For patient tumors with MSS status, we found a lack of response to pembrolizumab for all of them (Fig. 6C–E). PD-L1 was detected at high levels for one of them, intermediate levels for another one and very low levels for the 2 others (Fig. 6D). Thus, consistent with the clinic, the MSS/MSI status predominated over PD-L1 levels in predicting the sensitivity of patient tumors (Fig. 6F). Nevertheless, the potential response of MSS tumors with high PD-L1 levels to anti-PD-1 therapy is debated (Gorzo et al, 2022). We noted that for the MSS CRC sample having highest PD-L1 level (CRC-03, 87% PD-L1) the tumor volume tended to decrease following pembrolizumab treatment. To further examine this case, we thought to mix the tumor cells with another donor, also found competent in our previous experiments (donor 6) in order to determine if we could reveal a higher response. Interestingly, with this donor, we found a significant reduction of tumor volume in the pembrolizumab condition, compared to control (Fig. EV4A). Overall, these experiments allowed revealing heterogeneity in CRC patient tumor responses to anti-PD-1 in correlation with their MSS/MSI status (Fig. 6E,F). They also showed that genomic profiles and PD-L1 levels are not sufficient for properly stratifying the population of patients that would benefit from anti-PD-1 treatment.

Next, we thought to extend our investigation to other solid cancers, metastatic melanoma and TNBC, as our AVI-PDX models successfully reproduced the clinical response to chemotherapeutics and targeted therapies (Jarrosson et al, 2021, 2023). Patients affected by both of these cancer types are eligible to anti-PD-1 therapy, under specific eligibility criteria, like unresectable and advancement state for melanoma and CPS (combined positive

score) of PD-L1 levels above 10 for TNBC (Keytruda® FDA.gov, 2020).

For melanoma, cells were engrafted within the dorsolateral migration path of neural crest cells, which drives the cells to form tumors under the skin, as reported in our previous work (Jarrosson et al, 2023). For TNBC, cells were engrafted in the ventral somite. Patient tumor cells were mixed with human PBMCs prior to engraftment. Embryos received intravenous injection of pembrolizumab or excipient 24 h post-grafting, and were harvested for 3D imaging 24 h post-treatment. We measured PD-L1 levels and found low expression for these samples (Fig. EV4B–F). In embryos grafted with a metastatic melanoma sample, we observed that tumor cells established a subcutaneous pattern of tumoral masses surrounded by disseminated cells. We found no decrease of the tumoral volume in the pembrolizumab condition, when compared to control (Fig. EV4C). However, intriguingly, we noted in the pembrolizumab condition that the cancer cell dissemination pattern seemed reduced compared to the control condition. To quantify this observation, we measured the distance separating the 10 farther cells from the primary mass (Fig. EV4D,E). We found that the average distance was significantly decreased in the pembrolizumab group, compared with the control one, thus suggesting an anti-metastatic effect of anti-PD-1 treatment for this patient tumor. For the TNBC sample, we found no effect of pembrolizumab administration, the measured tumoral volume for this group being even increased compared to control group (Fig. EV4G). This could be explained by the measured PD-L1 levels which we found lower than 10%.

Lastly, we examined if we could set a humanized avian paradigm based on the co-transplantation of patient cells with their autologous hu-PBMCs. One TNBC/auto-PBMC and two

CRC-MSI/auto-PBMC sample pairs were collected and co-engrafted in batches of avian embryos. For one of the CRC and the TNBC samples, some embryos were harvested for confocal microscopy of the tumors on transverse sections which confirmed the presence of autologous immune cells together with tumor cells visualized with CFSE (Fig. 6G). In addition, for the 2 CRC samples, we imaged the cancer and immune cell contingents in whole embryos using light sheet microscopy (Figs. 6H and EV4H). For this purpose, autologous hu-PBMC cells were labeled with Orange Cell Tracker and tumor cells with CFSE prior to the co-grafting. The tumor cell density differed between the 2 cases, due to the initial sample size. Regardless of this difference, we found that, in both cases, tumor cells were surrounded by their autologous immune cells, as observed in the models engineered with hu-PBMCs from healthy donors (Fig. 6H). This validated the feasibility of enriching the tumors with autologous immune cells in the humanized AVI model. Next, we examined the outcome of pembrolizumab administration in these models. Batches of embryos were treated with pembrolizumab or excipient and harvested for tumor volume measurements. For all 3 cases, we found no significant tumor volume reduction in the pembrolizumab condition, compared to control excipient (Fig. 6I).

## Discussion

Our study reports the development of an avian model for cancer cell line-derived and patient-derived xenograft with a reconstituted human immune microenvironment that recapitulates tumor infiltration and establishment of cancer-immune cell communication. By combining experimental manipulations in the avian embryo, light sheet microscopy and molecular marker analyses, we provide the first proof-of-concept that this humanized AVI model enables assessing the efficacy of immunotherapeutic treatments on different solid tumors such as colorectal cancer, breast cancer and melanoma, in a procedure that takes only 3 days. By creating patient tumor replicas of colorectal cancer, we show that the humanized AVI model is capable of revealing heterogeneity in patient tumor responses according to genomic and molecular markers used in the clinic to select patients eligible to anti-PD1 immunotherapy. Finally, we provide evidence that humanized-AVI-Patient-Derived Xenograft-based immunotherapy studies can be extended to other solid cancers like melanoma and breast cancers, and can be conducted using models engineered with autologous immune cells.

### Humanizing the avian patient-derived xenograft model allows to recapitulate important features of the tumor microenvironment

In our previous work, we reported a process to engraft small amounts of patient tumor cells of many types, from pediatric to adult cancers, from solid and liquid ones, into selected tissues of avian embryos (Delloye-Bourgeois et al, 2017; Jarrosson et al, 2021; 2023; Zala et al, 2024). We demonstrated that it is a powerful paradigm enabling the fast and reproducible creation of miniaturized patient tumor replicas, allowing the study of drug candidate efficacy and mechanisms of action across heterogeneous patient populations (Delloye-Bourgeois et al, 2017; Jarrosson et al, 2021;

2023; Zala et al, 2024). In these models, the avian embryo's endogenous immune system is not yet functional during the experimental time window which ranges from E2 to E7 (Vainio and Imhof, 1995; Ribatti and Tamma, 2019). This condition minimizes a low allogenic response and supports efficient human cell survival and growth but limits immunomodulation-based studies with the avian adoptive immune system.

Cancer therapies targeting immune checkpoint inhibitors hold considerable promises but patients eligible to current approved immunotherapies still remain low (Haslam et al, 2025). Accelerating access to novel immunotherapeutic treatments for the patients require preclinical models recapitulating key features of their tumors, including their inter- and intra-tumor heterogeneity, and cancer-immune communications.

Our results show that within the avian embryonic tissues, engrafted human cancer and immune cells establish physical and molecular communications that can be manipulated.

First evidence comes from the analysis of the infiltration model, a process consisting in implanting tumor cells first and intravenously injecting hu-PBMCs second. When comparing hu-PBMCs distribution in cancer cells bearing-embryos and controls, we found that the density of hu-PBMCs in the tissue hosting the tumor was higher than in the equivalent control tumor-free tissue. The migration of immune cells to a tissue is a complex process involving specific signaling mechanisms that direct cell movement and perception of extracellular signals orienting the direction of the movement (SenGupta et al, 2021). These mechanisms also apply to the context of cancer and play a major role in setting tumor microenvironment characteristics (Ryan et al, 2024). Our findings show that human cancer cells forming tumors in the avian embryo tissues are capable of providing long range signaling attracting innate and adaptive human immune cells. Immune cell infiltration of tumors varies between cancers and across patients, and is considered as a factor of variability in the efficacy of immunotherapeutic treatments (Eljilany et al, 2024). Our model thus provides a fast track to study the mechanisms underlying immune infiltration and the communication with cancer cells.

Second evidence comes from our close analysis with confocal and light sheet microscopy of the spatial distribution of immune cells when engrafted in the avian embryo tissue either alone or together with tumor cells. We found that the presence of cancer cells significantly reduced the immune cell spreading, thus suggesting that cancer cells provide recognition signals trapping immune cells within the tumor and its close microenvironment. Further investigations assessing cytokine release may help decipher the underlying signals involved in immune cell chemotaxis within implanted tumors.

Third, our flow cytometry analysis of immune cell markers provides evidence for molecular crosstalk between immune and cancer cell populations. We observed that T lymphocytes responded to tumor cells by modulating activation and exhaustion markers, consistent with an ongoing immunosuppressive process.

Interestingly, we found that shunting the tumor infiltration process by directly co-implanting immune and cancer cells had minor impact on survival, diversity of immune cell populations and establishment of cellular and molecular communications.

This co-grafting paradigm also holds an important advantage in the control of the respective ratio of immune and cancer cell contingents, allowing to mimic different infiltration levels and

providing high flexibility for preclinical investigations. Furthermore, within the same time frame, while the infiltration model only revealed early T cell activation, we found an additional engagement towards exhaustion with the co-grafting paradigm. This difference is likely attributable to the extended time window of immune-cancer cell communication in the co-grafting procedure, compared to the infiltration one. Interestingly, this mixed phenotype of activation and exhaustion closely mirrors the functional state of T cells within the tumor microenvironment, thereby providing a relevant framework to study T cell modulation in response to immunotherapeutic interventions.

Beyond the immune component, the avian embryo models provide additional components to reconstitute a complex micro-environment within a tissue context. Vessels and neuronal structures are already present, irrigating and innervating the organs. Although the endogenous adaptive immune system is not yet functional, innate immune cells such as primitive macrophages circulate within the embryonic organism, where they are active in tissue maturation (Qureshi, 2003; Cuadros et al, 1992; Houssaint, 1987). The avian embryo also brings systemic molecular signaling (Bellairs and Osmond, 2014). Finally, the embryonic tissues, specifically the sclerotome compartment, in which tumors are driven in the models reported here, comprise a diversity of cell-types. Interestingly, mesenchymal cells that transition towards becoming either chondrocytes or cartilage, produce a high amount and variety of extracellular matrix components (Rifes and Thorsteinsdóttir, 2012). Nevertheless, despite this already high molecular, cellular and architectural complexity, differences between immature and mature tissues, as well as inter-species specificities, can impact on the cancer cells' behaviors and response to treatments.

## Circulating and infiltrated hu-PBMCs maintain diversity of cell populations and show inter-donor specificities

In mouse models humanized by intravenous injection of hu-PBMCs, about 50% of cells survive after 4 weeks, the vast majority of which are CD3+ T cells (de La Rochere et al, 2018). Notably, we found in the AVI model that immune cell populations of hu-PBMCs present in the peripheral blood and within the avian tissues were diverse. In particular, unlike in mouse models, B, Myeloid and NK cells were capable of surviving. These cells may benefit from chemokine supply by the avian environment. Consistently, many chemokines, such as CXCL12, among which some are important for B cell development and activity are strongly expressed in the avian embryonic organs (Nagy et al, 2020; Yahya et al, 2023; Nassari et al, 2017). Our paradigm thus opens the way for the study of immune responses beyond effector T cells in the tumor microenvironment. This is particularly promising for NK cells and macrophages, which play pivotal roles in tumor immune surveillance through mechanisms independent of antigen-specific recognition, NK by directly eliminating malignant cells in a major histocompatibility complex (MHC)-independent manner, and macrophages by shaping the tumor milieu through phagocytic activity and cytokine-mediated modulation, enabling rapid responses consistent with short time frame efficacy assessments in the AVI system (Shimasaki et al, 2020).

By analyzing hu-PBMC samples individually, we found inter-donor variabilities at different levels, from the representativity of immune cell populations in the tumor to T cell activation state and competency in enabling anti-PD-1-mediated anti-tumoral effect. Interestingly, when comparing the competency of different hu-PBMC donors we found that HLA-A2 matched tumor/hu-PBMC combinations were those showing the highest efficiency, suggesting enhanced recognition mechanisms between cells having common haplotype profiles. However, class I HLA possible combinations are multiple (Mangum and Caywood, 2022), which makes it highly challenging to predict competent donors. Consistently, donor 6, found with a lower competence to enable anti-PD-1 activity with MDA-MB-231 cells was proven effective when combined with a CRC patient sample.

## The humanized avian models reveal efficacy of anti-PD-1 administration

First using various cancer cell lines and hu-PBMCs, we demonstrated that anti-PD-1 administrated intravenously could rearm immune cells allowing them to attack cancer cells. By administering an anti-MHC-I antibody, we demonstrated that this anti-tumoral effect is mediated by T cell recognition of the tumor, providing initial mechanistic insights. However, further research is needed in the form of direct cytotoxicity assays, cytokine measurements and exploration of antigen presentation pathways to better characterize immune engagement and the mechanisms underlying the therapeutic response. Due to the miniaturized nature of avian models, the dosage of chemokines in the tumor microenvironment is likely to require specific technical developments.

Interestingly, in the humanized AVI-models, we could reproduce the sensitivity or resistance of these cell lines to anti-PD-1 reported with murine models and/or in vitro studies (Wang et al, 2014; Wang et al, 2018). We also found that the patient tumor responses to pembrolizumab treatment were consistent with those reported in the clinic. We characterized both sensitivity and resistance profiles, and also reproduced heterogeneity of patient tumor responses predicted by current prognosis markers. Likewise, CRC samples with a MSI profile but not with a MSS one, showed strong tendency of sensitivity to anti-PD-1, consistently with clinical responses (Landre and Guetz, 2025). However, interestingly, we also observed that the responses of some tumors were not predicted by the current markers. As examples, among CRC patient tumors, we found one PD-L1 negative MSS sample that tended to be sensitive. For a metastatic melanoma sample, we found that anti-PD-1 administration had no effect on the tumor volume but strongly reduced the dissemination profile of tumor cells.

These findings illustrate the capacity of the humanized AVI-PDX models to reveal the heterogeneity of patient tumor responses. Whether for this melanoma patient, the observed reduced migration would result in decreased metastases and to which extend this anti-PD-1 property could be generalized remains to be determined.

In the present study, most patient samples were primary tumor resections collected at diagnosis, with clinical information limited to molecular features, preventing direct comparison of therapeutic responses between patient and their AVI-models. Thus, an important next step will be to conduct co-clinical studies to determine if patient responses to anti-PD-1 correlate with those of their replicas in the humanized AVI-model as we demonstrated it in dedicated previous work for therapeutic standards of melanoma

and follicular lymphoma (Jarrosson et al, 2021; Zala et al, 2024). This would pave the way for the identification of new molecular features distinguishing anti-PD-1 resistant and sensitive tumors.

Pembrolizumab showed no detectable efficacy in the reconstituted autologous immune-cancer cell tumors from the three analyzed patient samples. This may be due to the resistant profile of the tumor samples. Alternatively, autologous PBMCs may not be sufficiently effective, which may be attributed to insufficient levels of memory cells and T cells that are specific to the tumor. Regardless of the underlying cause, the successful demonstration of our approach provides a foundation for future investigations and screening of therapeutic immunomodulators.

## Advantages and limitations of the humanized AVI model

Over the years, a number of mouse models has been developed for studies of immune-cancer cell communication, from syngeneic and genetically engineered, to sophisticated models humanized with PBMCs or with CD34+ hematopoietic stem cells (HSCs) that can be used for patient-derived xenografts (de La Rochere et al, 2018). The technology of the humanized AVI model presents similarities with that of the hu-PBMC-mouse model. Both allow the study of fast T cell responses. Nevertheless, in mouse, host, immune and tumor crossed allogenic reactions narrow the window of investigations to a few weeks, which can be challenging since, in most of cases, grafted patient cells have low intake and growth rates in these models. In the hu-AVI models, allogenic reactions are likely to also occur. However, it does not limit the investigations, due to inherent characteristics of the model supporting high tumor intake and rapid growth, as well as fast-track analysis of therapy efficacy.

Another advantage is that our co-transplantation procedure enables for precise control of the tumor/immune cell ratio within the tumor for each individual. This feature facilitates the management of inter-individual variability in immune cell infiltration and the execution of quantitative and statistical analyses. Moreover, unlike in the mouse, the humanized AVI-model supports the survival and the representativeness of a relevant repertoire of immune populations and therefore the reconstruction of a varied humanized system that more closely resembles the tumor stroma. This advantage enables the exploration of tumor cell communications and immune populations beyond T lymphocytes, which are almost the only persistent population of PBMCs in mice (de La Rochere et al, 2018). The hu-CD34+ mouse allows the reconstitution of the full repertoire of immune cell types. However, not all reach full maturity and several exhibit functional impairments, with some suggested to result from species-difference of education on thymus. Despite these drawbacks, this model holds the advantage of allowing long-term investigations. Indeed, the main limitations of the humanized AVI model, which are also present in the hu-PBMC mouse model, are the acute nature of the humanization procedure and the short investigational time window, which does not capture long-term processes. It would be interesting to prolong the experimental process in the AVI model in order to evaluate if this could capture additional important features of the tumor microenvironment accounting for the response of cancer cells to immune-based therapies. Further detailed characterization of the immune-cancer communication will help to establish comparisons with existing models.

The humanized AVI-model's notable flexibility is an undeniable asset. Hu-PBMCs can be replaced by immune populations that have been selected, genetically modified or subjected to experimental manipulations beforehand, as well as by immune cells autologous to tumor cells. The flexibility of these models is also reflected in the range of therapeutic treatment options, as demonstrated by our ability to conveniently test immunotherapy either individually or in combination with chemotherapies. Unlike in mouse, implementing the hu-AVI model with patient-derived xenografts does not introduce additional complexity. It retains the general advantages of the paradigm, from the miniaturized nature of the procedure, which is particularly suitable for small clinical samples, to the ability to target tumor formation in tissues selected according to specific cancer characteristics. Other unique features of the humanized AVI models lie in their speed of implementation, the favorable and reproducible conditions for tumor growth and immune cell survival, and the power of 3D imaging at scales ranging from the cellular to the whole organism. They are a more cost-effective solution than existing in vivo models. They also fully comply with the ethical rules governing animal experimentation.

In this regard, these models are particularly well-suited for preclinical drug discovery and lead validation studies. Their capacity to reveal differential responses among patient tumors makes them highly suitable for identifying predictive biomarkers. Furthermore, they will facilitate the exploration of numerous biological questions, including the communication between cells or the comparison of circulating and tumor-infiltrating immune cells. These questions could be addressed using models with PBMCs from healthy donors and with those with autologous immune-cancer cell samples.

# Methods

**Reagents and tools table**

| Reagent/Resource | Reference or Source | Identifier or Catalog Number |
|---|---|---|
| **Experimental models** | | |
| Embryonated eggs | Couvoir de E.AR.L les Bruyères and couvoir Elevage Avicole du Grand Buisson. | N/A |
| MDA-MB-436 (*H. Sapiens*) | ATCC | HTB-130 |
| MDA-MB-231 (*H. Sapiens*) | ATCC | HTB-26 |
| HCT 116 (*H. Sapiens*) | Cytion | 300195 |
| Peripheral blood from healthy donor (*H. Sapiens*) | Etablissement Français du Sang | N/A |
| TUM634 metastasis melanoma sample | NeuroBioTec | N/A |
| CRC-03 sample | NeuroBioTec | N/A |
| BRE-01 Breast cancer sample | Biological Resource Center of the Lyon Sud Hospital (Hospices Civils de Lyon) | N/A |
| CRC-01, CRC-02, CRC-04, CRC-05, CRC-06 Colorectal cancers and BRE-02 breast cancer sample with autologous PBMC | AP-HP, CRB Ferdinand Cabanne of the CHU Dijon and CHU Montpellier. | N/A |

| Reagent/Resource | Reference or Source | Identifier or Catalog Number |
|---|---|---|
| CRC-07 with autologous PBMC CRC-08 with autologous PBMC | Institut de Recherche Saint-Louis, INSERM U1342, Paris, France | N/A |
| **Antibodies** | | |
| Anti-human CD 274 APC | BioLegend | 329708 |
| Anti-human HLA-A2 APC | BioLegend | 343308 |
| Anti-human CD45 BV450 | BD Biosciences | 560367 |
| Anti-human CD3 BUV 395 | BD Biosciences | 563546 |
| Anti-human CD8 PerCP Cy 5.5 | BD Biosciences | 560662 |
| Anti-human CD4 Alexia Fluor 700 | BD Biosciences | 566318 |
| Anti-human CD16 APC Fire 750 | BioLegend | 302059 |
| Anti-human CD56 PE | BD Biosciences | 556647 |
| Anti-human CD19 PE-Cy7 | BD Biosciences | 560728 |
| Anti-human CD14 APC | BD Biosciences | 561708 |
| Anti-human CD279 BV650 | BioLegend | 329949 |
| Anti-human Tim-3 APC | BioLegend | 345011 |
| Anti-human Tim-3 BV650 | BioLegend | 345027 |
| Anti-human CD69 PE-CF594 | BD Biosciences | 562645 |
| Anti-human CD25 BUV737 | BD Biosciences | 612806 |
| Anti-human CD152 APC | BD Biosciences | 560938 |
| Anti-human IgG APC | BioLegend | 400119 |
| Anti-human IgG BV650 | BioLegend | 400163 |
| Anti-human IgG PE-CF594 | BD Biosciences | 562538 |
| Anti-human IgG BUV737 | BD Biosciences | 753710 |
| mouse IgG | Invitrogen | 02-6502 |
| Fixable Viable Stain V510 | BD Biosciences | 564406 |
| Rat Anti-human CD45 | Invitrogen | MA5-17687 |
| Rabbit anti-human cleaved Caspase 3 | Ozyme | 9661 L |
| Anti-rat IgG Alexa Fluor 647 | Interchim | FP-SC6110 |
| Anti-rabbit IgG FluoProbes 547H | Interchim | FP-SB6110 |
| Anti-HLA-A,B,C (pan- HMC I) | Biolegend | 311402 |
| Hoechst | Invitrogen | H21486 |
| **Chemicals, Enzymes and other reagents** | | |
| DMEM (Dulbecco's modified Eagle medium) | Gibco | 61965059 |
| RPMI1640 (Roswell Park Memorial Institute medium Dulbecco's) | Gibco | 61870044 |
| CaCl2 | Merck Sigma | C3306-100g |
| Type IV Collagenase IV | Merck Sigma | C5138-100mg |
| DNase I | Merck Sigma | DN25-10mg |
| Trypsin | Merck Sigma | T5266-500mg |

| Reagent/Resource | Reference or Source | Identifier or Catalog Number |
|---|---|---|
| Ficoll | Cytiva | 17-1440-03 |
| Percoll | Cytiva | 17-0891-02 |
| FBS | Dutscher | S1400-500A |
| HBSS (Hank's Balanced Salt Solution) | Gibco | 14170138 |
| CellTraceTM CFSE solution | Life Technologies | C34554 |
| CellTracker™ Orange CMTMR Dye | Invitrogen | C2927 |
| Ethyl-Cinnamate | Merck Sigma | 112372 |
| Brilliant Stain Buffer | BD Biosciences | 563794 |
| Paraformaldehyde (PFA) | Cliniscience | 15714-S |
| Pembrolizumab | MSD | 100 mg/4 ml |
| Gemcitabine | Accord | 2000 mg/20 ml |
| Carboplatin | Accord | 600 mg/60 ml |
| Docetaxel | Accord | 20 mg/1 ml |
| PBS | Merck Sigma | D8537-6X500ML |
| Triton | Merck Sigma | T9284-500mL |
| DMSO | Merck Sigma | D8418-250ml |
| BSA | Merck Sigma | A3059 |
| Gelatin | VWR | 24360.233 |
| Sucrose | Merck Sigma | S0389-1kg |
| **Software** | | |
| GraphPad Prism 10 | https://www.graphpad.com | N/A |
| FlowJo | https://www.flowjo.com/ | N/A |
| Imaris | https://imaris.oxinst.com/ | N/A |
| Image J | https://imagej.nih.gov/ij/index.html | N/A |
| LASX image analysis software | Leica | N/A |
| **Other** | | |
| UltraMicroscope BLAZE | Miltenyi | N/A |
| Flow cytometer Canto II | BD Biosciences | N/A |
| Flow cytometer LSR II | BD Biosciences | N/A |
| Olympus confocal microscope, FV1000, X81 | Olympus | N/A |
| Zeiss Axiozoom | Zeiss | N/A |
| MZ10F stereomicroscope | Leica | N/A |
| Pneumatic PicoPump PV820 | World Precision Instruments | N/A |
| Balance Quintix 35-1S | Sartorius | N/A |

## Chick embryos

Embryonated eggs were obtained from local suppliers (Couvoir de E.AR.L les Bruyères, Dangers, France and couvoir Elevage Avicole du Grand Buisson, Chabanière, France). Laying hen's sanitary status was regularly checked by the supplier according to French laws. Eggs were housed in an 18 °C incubator until use.

For the experiments, eggs were incubated at 38.5 °C in a humidified incubator. Engrafted embryos were randomized in each experimental group and were harvested at embryonic day 4 (4 days post-fertilization). The number of embryos used in each experiment is indicated in the corresponding figure.

## Cell lines

The MDA-MB-436 (ATCC-HTB-130) and MDA-MB-231 (ATCC-HTB-26) TNBC cell lines were obtained from American Type Culture Collection and HCT 116 from Cytion (116:300195) and were not re-authenticated. Cell lines were regularly tested for absence of mycoplasma contamination over the duration of experiments. MDA-MB-436 was cultivated in Roswell Park Memorial Institute medium Dulbecco's (RPMI1640, Gibco); MDA-MB-231 and HCT 116 were cultivated in Dulbecco's modified Eagle medium (DMEM, Gibco) both supplemented with 10% Fetal Bovine Serum (FBS). HCT 116 cells are incubated with human cytokines produced by PBMCs reported to stabilize PD-L1 surface expression. PD-L1 expression on cell lines was checked by cytometry using anti-human CD 274 APC (29E2A3 clone, BioLegend). HLA-A2 haplotype was determined according to Adams et al, 2005 and Thomas et al, 2020.

## Patient tumor samples and hu-PBMCs

All experiments involving human samples conformed to the principles set out in the WMA Declaration of Helsinki and the Department of Health and Human Services Belmont Report. The donors' and patients' ages, sexes and ethnicities were not disclosed. Patient sample TUM634 metastasis melanoma and CRC-03 were obtained from NeuroBioTec (CRB HCL, Lyon France, Biobank BB-0033-00046) and is part of a collection declared at the French Department of Research (DC-2008-72). BRE-01 breast cancer sample was obtained from Biological Resource Center of the Lyon Sud Hospital (Hospices Civils de Lyon) following surgery. CRC-07 and CRC-08 samples with their autologous PBMCs were obtained from Institut de Recherche Saint-Louis (France). Other CRC patients and BRE-02 with autologous PBMC samples were obtained from AP-HP, CRB Ferdinand Cabanne of the CHU Dijon and CHU Montpellier. Patients were informed and signed consent to reuse biological samples for research purposes. Fresh or frozen patient samples were dissociated in Hank's Balanced Salt Solution (HBSS) with 156 units/ml of type IV collagenase, 5 mM $CaCl_2$ and 50 units/ml of DNase I for 20 min at 37 °C and then incubated with 5 mg/ml trypsin for 2 min at 37 °C under gentle mixing. Non-dissociated tissue was removed by filtration trough 40 μm nylon cell strainer (BD Falcon).

Human peripheral blood mononuclear cells (PBMCs) were obtained from healthy donors through the Établissement Français du Sang (EFS, France) under an agreement with ONCOFACTORY. Blood donations were collected after informed consent in accordance with the EFS ethical guidelines and the French Public Health Code. Human peripheral blood mononuclear cells (hu-PBMCs) fraction was isolated from peripheral blood by density gradient separation (Ficoll, Cytiva). Cells were cryopreserved in RPMI 20% FBS 10% DMSO at a concentration of $20 \times 10^6$ cells until later use. HLA-A2 haplotype of immune cells was defined by cytometry using anti-human HLA-A2 APC (BB7.2 clone, BioLegend).

## Ethical statement

Our experiments on avian embryos were performed between embryonic days 2 and 4, i.e., before the final third of gestation, in compliance with EU Directive 2010/63/EU (formerly 86/609/EEC). Specific ethical authorization was not required. The study fully adheres to the 3Rs principles. Experiments were limited to those not covered by existing data and were carried out according to carefully pre-planned protocols. All procedures followed the approved guidelines, favoring early developmental stages when pain perception is ineffective. Whole-organism studies in avian embryos were necessary in order to model cancer in an environment that mimics the human condition, as well as the mode of administration of therapies.

## Xenografts in the avian embryo

Embryonated eggs were incubated at 38.5 °C in a humidified incubator until E2 stage (HH14). Cell lines or patient samples were labeled with 1 or 7 μM CellTrace™ CFSE solution (Life Technologies). Hu-PBMC samples could be submitted to density gradient according to manufacturer's protocol (Cytiva) to isolate living cells and stain them with Orange Cell Tracker at 5 μM (CellTracker™ Orange CMTMR Dye, Invitrogen). Before co-engraftment, hu-PBMCs and tumoral cells were mixed at different ratio depending on the experiments. Cells were grafted into stage E2 chick embryos, within the developing ventral somite for CRC, breast cancer cells and hu-PBMCs or at the top of the dorsal neural tube for melanoma cells, with a glass capillary connected to a pneumatic PicoPump (PV820, World Precision Instruments) under a fluorescence stereomicroscope. Targeted tissue areas for the graft were visualized under the stereomicroscope.

For cell lines, approximately 2500 living cells were grafted per embryo, 200 to 500 for patient samples. For patient samples, the full cellular content obtained after dissociation was engrafted including stromal and/or immune cells.

For intra-venous injection, hu-PBMCs were labeled with Orange Cell Tracker CMTMR Dye at 5 μM (Invitrogen) or not labeled. Hu-PBMCs were diluted in PBS at $3.3 \times 10^5$ cells/μL or $1.65 \times 10^5$ cells/μL and $2 \times 10^5$ cells were injected at HH16 stage in the marginal sinus of the embryo.

The blockade of MHC I at the surface of T cells was performed during an in vitro pre-engraftment step. Hu-PBMCs and cancer cells were mixed at an effector:target 4:10 ratio, and submitted to 30 min incubation with an anti-HLA-A, -B, -C antibody (50 μg/mL) in RPMI medium.

## Anticancer therapy

Keytruda® (pembrolizumab, stock solution: 25 mg/mL), Gemcitabine (100 mg/mL), Carboplatin (10 mg/mL) et Docetaxel (20 mg/

mL) were obtained from the Centre Leon Berard, Lyon, France. A solution of 0.9% NaCl was used as excipient for treatment preparation and vehicle control. The determination of maximum tolerated dose was achieved by testing intravenously injected decreasing doses. For pembrolizumab, tested doses ranged from pure solution (890 mg/kg) to dilution by 5-fold (178 mg/kg, 35.6 mg/kg). For docetaxel, tested doses ranged from 35 mg/kg to dilution by 5-fold (7, 1.4 mg/kg and 0.28 mg/kg). For treatment, pembrolizumab was diluted at 1:5 (final solution: 5 mg/ml), Gemcitabine at 1:5263 (19 µg/ml), Carboplatin at 1:3.8 (2.6 mg/ml), and Docetaxel at 1:513 (39 µg/ml). Grafted embryos were randomized for experimental group design.

## Evaluation of therapy and hu-PBMC injection toxicity in avian embryos

Avian embryos were harvested at E4 (HH25) stage, weighted (Sartorius Quintix 35-1S) and measured along the rostro-caudal axis using Leica LASX image analysis software. The Body Surface Area (BSA) was calculated using Dubois & Dubois formula:

$$BSA\,(m^2) = 0.20247 \times height(m)0.725 \times weight(kg)0.425.$$

For each embryo, we analyzed six criteria to assess if the embryonic development properly proceeded: the survival (heart beating), craniofacial morphology (presence of each cerebral compartment and eyes), presence of four limb buds, cardiac morphology, and anatomy of embryonic annexes such as the allantois.

## Tissue clearing and whole-mount SPIM imaging

PFA-fixed E4 embryos (HH25) were cleared using an adapted Ethyl-Cinnamate protocol (Klingberg et al, 2017). Briefly, tissues were dehydrated in ethanol successive baths finally cleared in Ethyl Cinnamate (Sigma, 112372). Cleared samples were imaged using the UltraMicroscope BLAZE (Miltenyi). 3D-images were built using Imaris™ software. Volumetric analysis was blindly performed using Imaris™ "Surface" module adjusted on CFSE or Orange Cell Tracker fluorescence. The distribution patterns were quantified using the "Surface" module adjusted on the CFSE fluorescence, and "Spot" module adjusted on the Orange Cell Tracker fluorescence. The distance was measured using the "shortest distance to surfaces" tool, representing the distance between the spot center position and the closest surface. The volume overlap between Orange Cell Tracker positive population and CFSE positive population was evaluated using "background subtraction" and "colocalization" tools. The hu-PBMC-tumoral cell contacts were analyzed using the "spot" module adjusted on CFSE and Orange Cell Tracker fluorescence. Each individual cell was represented as a spot. The distance was measured using the "shortest distance spot to spot" tool, as the distance between the center of two spots with different fluorescence. For the determination of average cell diameters, 20 CFSE+ and 20 Orange Cell Tracker+ cells were selected in 10 engrafted embryos and their diameter measured in 2D-image view.

## Tissue preparation for flow cytometry

Fluorescently-labeled tumors formed in avian embryos treated either with pembrolizumab or excipient, were micro-dissected 48 h after grafting using a stereomicroscope. In experiments with engrafted unlabeled hu-PBMCs, the whole embryo trunk was harvested. Dissected tissues were dissociated using an enzymatic cocktail comprising 156 units/ml of type IV collagenase and 40 units/ml of DNase I. Unlabeled hu-PBMCs were isolated by Percoll® separation (Cytiva).

For experiments including intravenously injected hu-PBMCs, fluorescently labeled tumors were micro-dissected and dissociated 48 h post injection under stereomicroscope. Equivalent control tissues were dissected from non-grafted embryos or equivalent counterpart in engrafted embryo. The peripheral blood of avian embryos was harvested with a glass capillary connected to the pneumatic PicoPump and hu-PBMCs were isolated by density gradient separation (Ficoll).

## Staining for flow cytometry acquisition

Cells were resuspended in phosphate-buffered saline (PBS) and pelleted at 600 r.c.f. for 5 min. Discrimination of viable cells with Fixable Viable Stain V510 (BD Biosciences) and saturation with mouse IgG whole molecule (Invitrogen) were firstly performed according to manufacturer's instructions. Cells were resuspended in PBS −2% FBS and incubated with antibodies for 20 min at 4 °C. If more than 3 brilliant violet fluorochromes were used in a panel, a Brilliant Stain Buffer (BD Biosciences) was used according to manufacturer's instruction. All antibodies were purchased from BD Biosciences unless otherwise noted. The specificity of all antibodies to human species has been checked. Flow cytometric analysis was performed using anti-human CD45 BV450 (HI30), anti-human CD3 BUV395 (UCHT1), anti-human CD8 PerCP Cy 5.5 (RPA-T8), anti-human CD4 Alexia Fluor 700 (SK3), anti-human CD16 APC Fire 750 (3G8, BioLegend), anti-human CD56 PE (MY31), anti-human CD19 PE-Cy7 (HIB19), anti-human CD14 APC (M5E2), anti-human CD279 BV650 (PD-1, EH12.2H7, BioLegend), anti-human TIM-3 APC or BV650 (F38 2E2, BioLegend), anti-human CD69 PE-CF594 (FN50), anti-human CD25 BUV737 (2A3), anti-human CD152 APC (CTLA-4, BNI3). Cell suspensions could be fixed in 2% Paraformaldehyde (PFA), and analyzed on FACs Canto II (BD Biosciences) or LSR II (BD Biosciences). IgG isotype control antibody conjugates (BD Biosciences and BioLegend) were included to establish background. Analysis was performed using FlowJo (BD Biosciences) software. The viability of hu-PBMCs was monitored by gating the viable population within CD45+ population.

## Immunofluorescence on cryosections

Avian embryos were harvested and fixed in 4% PFA. After overnight incubation in PBS-15% sucrose, embryos were embedded in PBS- 7.5% gelatin- 15% sucrose to perform 15 to 20 µm transverse cryosections. Permeabilization and saturation of sections were performed in PBS-bovine serum albumin 3%-Triton 0.5%. Anti-human CD45 (dilution 1/250, YAML501.4, Invitrogen), anti-human cleaved Caspase 3 (dilution 1/100, Asp175, Ozyme) were used as primary antibody. Anti-rat IgG Alexa Fluor 647 (dilution 1/500, Interchim), antirabbit IgG FluoProbes 547H (dilution 1/500, Interchim) were used as secondary antibody. Nuclei were stained with Hoechst (H21486, Invitrogen). Slices were imaged with a confocal microscope (Olympus, FV1000, X81) using either a 10X

## The paper explained

### Problem

Immunotherapy—cancer treatments that reactivate the immune system—stands out as one of the most promising therapeutic strategies. Advancing these therapies depends on models that faithfully replicate tumor-immune interactions in patient-like environments. However, current models, though powerful, are often complex and time-consuming to implement, hindering progress. Innovating faster, more streamlined approaches could bridge this gap.

### Results

We previously developed a fast, miniaturized xenotransplantation model by engrafting human cancer cell lines and patient-derived tumors into selected tissues of avian embryos. This approach allows tumors to grow within a native tissue and organismal context. Here, we present two new paradigms for humanizing the avian embryo model. Using advanced imaging and molecular analyses, we show that this reconstituted tumor microenvironment enables the study of immune-cancer cell interactions and the evaluation of immunotherapy responses in both cancer cell lines and patient tumors.

### Impact

Humanized avian embryo models offer versatile solutions for diverse research needs, from deciphering immune-cancer cell interactions and unraveling the mechanisms of patient response heterogeneity to rapidly screening novel immuno-modulator candidates.

objective for whole slice imaging or a 60 or 40X objectives to focus on immunolabeling cells. Some experiments were analyzed using Zeiss Axiozoom from the Centre d'Imagerie Quantitative Lyon Est (CIQLE). Image analyses were conducted using ImageJ software.

## Quantification and statistical analysis

Statistical analyses were performed using Prism 10 (GraphPad). For parametric tests, both normality (D'Agostino test) and homoscedasticity of variances were checked before running two-sided statistical tests. Unpaired t-tests or Mann–Whitney tests were performed for unpaired data depending on the result of the normality test. When variances were significantly different, Welch's correction was applied. For paired data with minimum 6 values, the Wilcoxon test was performed. For contingency table analyses (comparison of percentage distributions), a Chi-square test was used. The exact statistical tests and *P*-values are indicated in the figure legends and on the figures, respectively. For some experiment, two replicas can be performed, and tumor volumes were pooled per condition. Volumes were normalized between replicas with a factor of correction based on excipient volume average. For all experiments, embryos that were damaged or incorrectly grafted due to technical manipulations were excluded from the analysis. *N* represented the number of independent experiments (replicates), *n* represented the number of embryos pooled in each condition.

## Data availability

Data for Fig. 2J, Fig. 2K, Fig. 4F and Fig. 4H were deposited in BioStudies. Figure 2J: S-BSST2726. Figure 2K: S-BSST2728. Figure 4F: S-BSST2729. Figure 4H: S-BSST2730.

The source data of this paper are collected in the following database record: biostudies:S-SCDT-10_1038-S44321-026-00398-5.

## Peer review information

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

## Acknowledgements

We thank the Centre Léon Bérard for providing us with pembrolizumab and chemotherapies. We thank Dr Pascal Clayette for his helpful suggestions and comments on the manuscript. We thank Sébastien Fauteux and Arthur Chevallier for their help to initiate the project. This work was supported by internal funding from Oncofactory-ERBC. It was conducted within the framework of the Labex DevWeCAN of Universite de Lyon, within the program 'Investissements d'Avenir' (ANR-11-IDEX-0007) operated by the French National Research Agency (ANR) (VC).

## Author contributions

**Marjorie Lacourrège**: Conceptualization; Data curation; Formal analysis; Investigation; Methodology; Writing—original draft; Project administration; Writing—review and editing. **Loraine Jarrosson**: Formal analysis; Investigation; Methodology; Writing—review and editing. **Clélia Costechareyre**: Formal analysis; Investigation; Methodology; Writing—review and editing. **Mathis Marcel**: Formal analysis; Investigation; Methodology. **Audrey Prunet**: Investigation; Methodology. **Mayrone Mongellaz**: Investigation; Methodology. **Camille Futelot**: Investigation; Methodology; Writing—review and editing. **Cyrus Rouhani**: Investigation; Methodology. **Huong Giang Nguyen**: Resources. **Thomas Aparicio**: Resources. **Lionel Le Bourhis**: Resources. **Céline Delloye-Bourgeois**: Resources; Supervision; Validation; Visualization; Writing—review and editing. **Romain Teinturier**: Conceptualization; Resources; Supervision; Validation; Visualization; Writing—review and editing. **Valérie Castellani**: Conceptualization; Supervision; Validation; Investigation; Visualization; Methodology; Writing—original draft; Writing—review and editing.

Source data underlying figure panels in this paper may have individual authorship assigned. Where available, figure panel/source data authorship is listed in the following database record: biostudies:S-SCDT-10_1038-S44321-026-00398-5.

## Disclosure and competing interests statement

VC and CD-B are co-founders of OncoFactory SAS (http://www.oncofactory.com). ML, LJ, CC, MM, MM, CR, CF, and RT are employees of Oncofactory-ERBC Lyon.

# Expanded View Figures

**Figure EV1.   The paradigms of avian humanization create a tumor microenvironment.**

(A) Representative FACS profiles of CD69 and CD25 activation markers or TIM-3 and PD-1 exhaustion marker expressions within the CD3+ CD4+ population of hu-PBMCs infiltrated in MDA-MB-231 tumor site or in equivalent control tissues. Data are presented for 37 pooled embryos intra-venously injected with hu-PBMCs from donor 5. (B) Histograms showing the quantification of tumor volumes in embryos grafted with CFSE+ MDA-MB-231 cell line, and intra-venously injected or not with hu-PBMCs from donor 4. $N = 1$ experiment ($n = 5$ embryos in control group and $n = 8$ embryos in I.V injected group). Dots represent volumes normalized to each embryo's BSA. Data are represented as mean ± SEM. Mann–Whitney test, exact *P*-values indicated on graph. ns: not significant. (C) Histogram showing the fraction of Caspase 3 positive cells within the avian somitic tissue, in presence and absence of hu-PBMCs-MDA-MB-231 cells (hu-PBMCs from donor 6). For each experimental group, $n = 7$ sections from 3 embryos. Mann–Whitney test, exact *P*-values indicated on the graph. ns: not significant. (D) Microphotographs of immunofluorescent labeling of cryosections of chick embryos at 48 h post co-grafting of CFSE+ MDA-MB-231 cells (green) and hu-PBMCs from donor 6. Immune cells were detected with anti-human CD45 antibody (red), and apoptotic cells with anti-cleaved Caspase 3 antibody (white). Nuclei of avian and human cells were stained with Hoechst (blue). The images show somitic tissues free or populated by grafted human cells. White arrow indicates avian Caspase 3+ cells. (E) Histogram showing the global distribution pattern of hu-PBMCs from donor 10 when grafted alone ($n = 9$ embryos) or in combination with MDA-MB-231 cells at a ratio 1:1 ($n = 11$ embryos), $N = 1$. Data are represented as mean ± SEM. Each dot represents the average distance of hu-PBMCs from the site of injection for individual embryos. Mann–Whitney test, exact *P*-value indicated on graph. **$P < 0.01$. (F, G) Histograms showing the differences of hu-PBMC subpopulation fractions for two donors, between conditions of pre-grafting, grafting alone or grafting in combination with MDA-MB-231 or HCT 116 cells. Embryos were harvested 48 h post grafting. $N = 3$ for pre- and post-graft groups. For the histogram in (F), $N = 3$ for co-grafted experimental group ($n = 8$ to 11 pooled embryos); For the histogram in (G), $N = 2$ for co-engrafted experimental group ($n = 7$ to 9). The histogram of post-graft is also presented in Fig. 1G. Data are represented as mean ± SEM. Chi-square test, *P*-values are indicated. ****$P < 0.0001$, ***$P < 0.001$ ($P = 0.0003$), *$P < 0.05$ ($P = 0.0373$). (H, I) Histograms showing the percentage of CD69 positive cells (H) and TIM-3 positive cells (I) within the CD3+ CD8+ and CD3+ CD4+ populations, for 3 donors in two experimental groups of hu-PBMCs grafted alone or co-grafted with MDA-MB-231 cells. Donors are color-coded (red: donor 1, orange: donor 6 and blue: donor 8). Each dot represents the result of pooled embryos, see details Fig. 3H,I. Mann–Whitney test, *P*-values are indicated on the graph. *$P < 0.05$, ns: not significant.

                                    

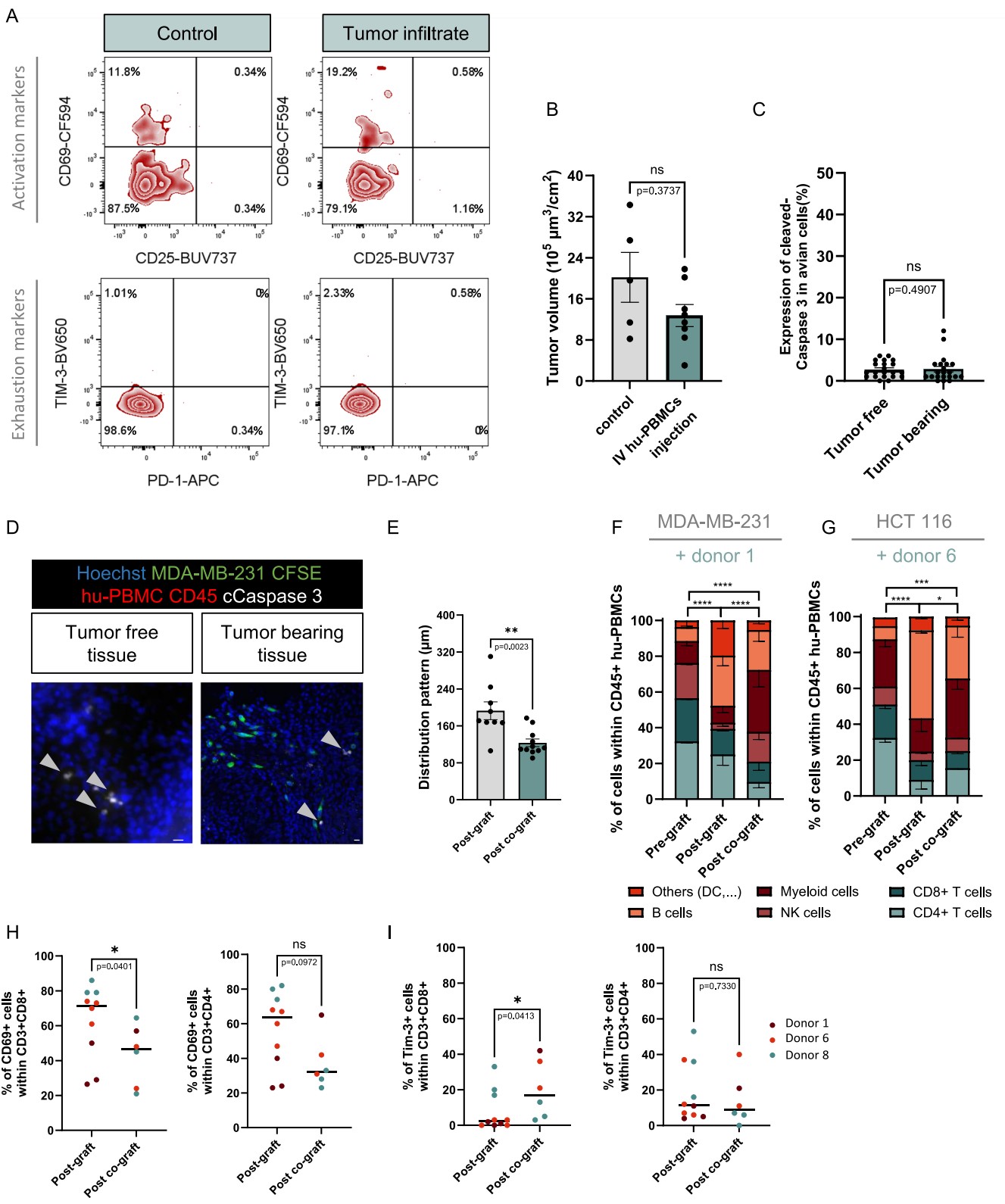

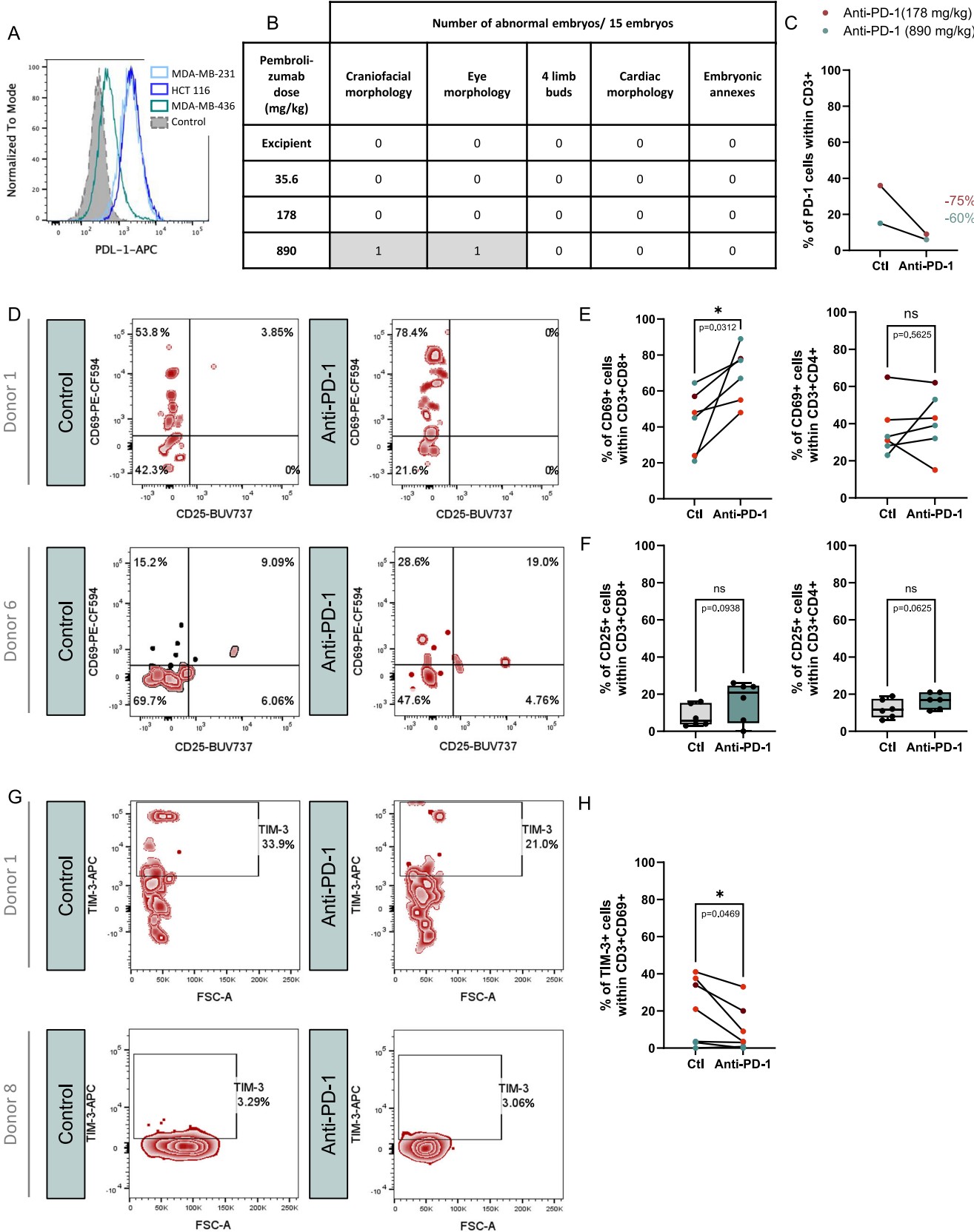

**Figure EV2.    The anti-PD-1 administration impacts human PBMCs.**

(A) Representative FACS profiles of PD-L1 expression in indicated cell lines (MDA-MB-231, MDA-MB-436, HCT 116). (B) Table presenting the morphological analysis of embryos treated with different pembrolizumab doses. (C) Histogram showing the fraction of PD-1 positive cells in CD3+ populations in embryos grafted with a mix of hu-PBMCs from donor 1 and MDA-MB-231 cells, treated either with anti-PD-1 (pembrolizumab) at 2 doses (178 in red or 890 mg/kg in blue) or control (NaCl 0.9%). Percentages represent the PD-1 expression diminution in pembrolizumab condition compared to control. $N = 1$, for 178 mg/kg dose ($n = 51$ pooled embryos in control group; $n = 54$ pooled embryos in pembrolizumab group) also shown in Fig. 4D; for 890 mg/kg ($n = 29$ and $n = 36$). No statistical test was performed. (D) Representative FACs profiles of CD69 and CD25 expression in the CD3+ CD8+ population of hu-PBMCs in embryos grafted with hu-PBMCs from donors 1 or 6, combined with MDA-MB-231 cells and treated either with control (NaCl 0.9%) or anti-PD-1 (pembrolizumab). Donor 1 ($n = 52$ pooled embryos per experimental group); donor 6 ($n = 20$ in control group; $n = 26$ in pembrolizumab group). (E) Histogram showing the fraction of CD69 positive cells in CD3+ CD8+ and CD3+ CD4+ populations in embryos grafted with a mix of hu-PBMCs and MDA-MB-231 cells and treated either with anti-PD-1 (pembrolizumab) or control (NaCl 0.9%). Each dot represents pooled embryos in experiments conducted with 3 donors (red: donors 1, orange: donor 6; blue: donor 8), see details Fig. 4E. Wilcoxon test, exact *P*-values indicated on the graph. *$P < 0.05$, ns: not significant. (F) Histograms showing the fraction of CD25 positive cells in the CD3+ CD8+ and CD3+ CD4+ populations in hu-PBMCs co-grafted with MDA-MB-231 cells in embryos treated either with anti-PD-1 (pembrolizumab) or with control (NaCl 0.9%). Each dot represents pooled embryos in experiments conducted with 3 donors (donors 1, 6 and 8). $N = 1$ for donor 1 ($n = 51$ pooled embryos in control group; $n = 54$ pooled embryos in pembrolizumab group); $N = 2$ donor 6 ($n = 17$ to 33; $n = 27$ to 36); $N = 3$ donor 8 ($n = 24$ to 27; $n = 19$ to 23). Experiments with fewer than 20 CD3+ CD8+ or 20 CD3+ CD4+ cells were excluded from analysis. Box-plots represent the median, interquartile range (25th–75th percentiles), with whiskers indicating the minimum and maximum values. Wilcoxon test, exact *P*-values indicated on the graph, ns: not significant. (G) Representative FACS profiles of TIM-3 expression in the CD3+ CD69+ population in embryos grafted with hu-PBMCs from donor 1 or 8, combined with MDA-MB-231 cells and treated either with control (NaCl 0.9%) or anti-PD-1 (pembrolizumab). Donor 1 ($n = 52$ pooled embryos per experimental group); donor 8 ($n = 24$ in control group; $n = 19$ in pembrolizumab group). (H) Histogram showing the fraction of TIM-3 positive cells in CD3+ CD69+ population in embryos grafted with a mix of hu-PBMCs and MDA-MB-231 cells and treated either with anti-PD-1 (pembrolizumab) or control (NaCl 0.9%). Each dot represents pooled embryos in experiments conducted with 3 donors (red: donors 1, orange: donor 6; blue: donor 8), see details Fig. 4G. Wilcoxon test, exact *P*-value indicated on the graph. *$P < 0.05$.

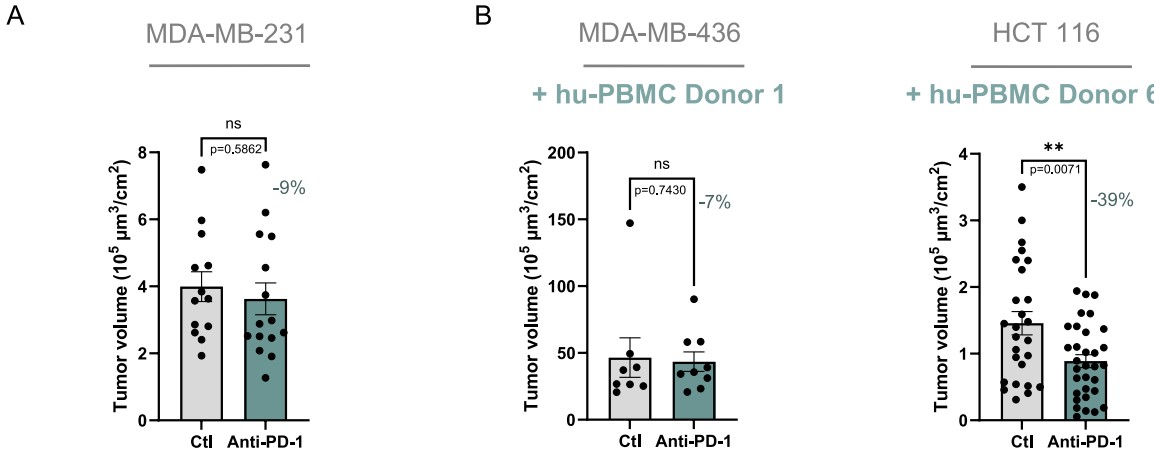

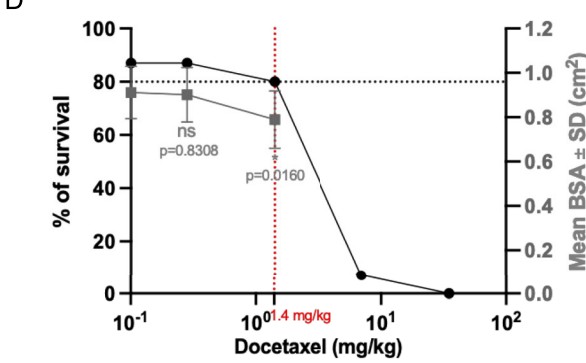

**Figure EV3. The anti-PD1 anti-tumoral effect is PBMC donor-dependent.**

(A) Histograms showing the quantification of the tumor volumes of MDA-MB-231, treated either with control (NaCl 0.9%) or anti-PD-1 (pembrolizumab). Dots represent volumes normalized to each embryo's BSA. $N = 1$ ($n = 13$ in control group and $n = 15$ in anti-PD-1 group). Data are represented as mean ± SEM. The percentage represents the average volume reduction. Unpaired T test, exact $P$-value indicated on graph, ns: not significant. (B) Histograms showing the quantification of the tumor volumes of MDA-MB-436 cells, and HCT 116 cells co-engrafted with hu-PBMCs from donor 1 or 6, respectively. Embryos were treated either with control (NaCl 0.9%) or anti-PD-1 (pembrolizumab). Dots represent volumes normalized to each embryo's BSA. $N = 1$ experiment per cell line, for MDA-MB-436 ($n = 8$ in control group and $n = 9$ in anti-PD-1 group); for HCT 116 ($n = 26$ and $n = 32$). Data are represented as mean ± SEM. The percentage represents the average volume reduction. Statistical analyses were conducted with Mann–Whitney for MDA-MB-436 and T test with Welch's correction for HCT 116. Exact $P$-values indicated on graphs. **$P < 0.01$, ns: not significant. (C) Table showing the characterization of HLA-A2 haplotypes of the different hu-PBMC donors and cancer cell lines. (D) Analysis of survival rate (left axis) and mean body surface area (BSA, right axis) of chick embryos injected with increasing doses of docetaxel (0.28, 1.4, 7, 35 mg/kg). Each dose and control (NaCl 0.9%) were injected to 15 embryos, $N = 1$. The dose 1.4 mg/kg was chosen as MTD. Data are represented as mean ± SD. Unpaired T-test compared to excipient, exact $P$-values indicated on the graph, ns: non-significative.

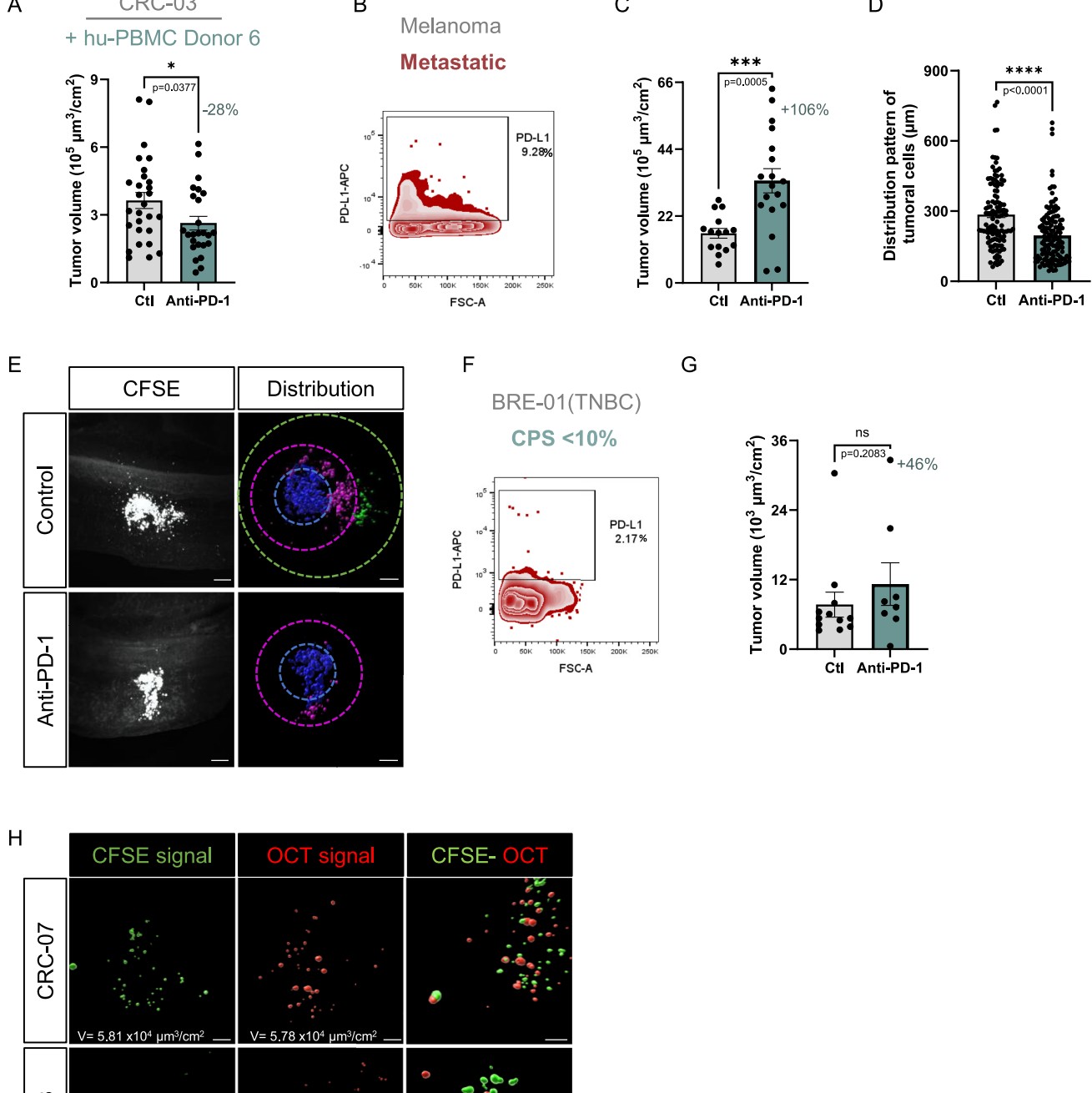

◄

**Figure EV4. The anti-PD-1 exerts an effect on several types of patient tumors.**

(A) Histograms showing the quantification of tumor volumes in embryos co-grafted with CRC-03 patient cells and hu-PBMCs from donor 6, treated with anti-PD-1 (pembrolizumab) or control (NaCl 0.9%). Dots represent volumes normalized to each embryo's BSA. $N = 1$ ($n = 28$ embryos in control group and $n = 25$ embryos in pembrolizumab group). Data are represented as mean ± SEM. The percentage represents the average volume reduction compared to control. Unpaired T test, exact *P*-value indicated on the graph. *$P < 0.05$. (B) Representative FACs profile of PD-L1 expression in a patient metastasis melanoma sample. (C) Histograms depicting the tumor volumes in embryos co-grafted with metastatic melanoma patient cells and hu-PBMCs from donor 1 and treated with anti-PD-1 (pembrolizumab) or control (NaCl 0.9%). Dots represent volumes normalized to each embryo's BSA. $N = 1$ ($n = 15$ embryos in control group and $n = 18$ in pembrolizumab group). Data are represented as mean ± SEM. The percentage represents the average volume reduction compared to control. Unpaired T test, exact *P*-value indicated on the graph. ***$P < 0.001$. (D) Histogram showing the average spreading distances calculated for the 10 cells furthest from the injection site of metastatic melanoma tumor per embryo. Dots represent individual embryos. $N = 1$ ($n = 15$ in control group and $n = 18$ in pembrolizumab group). Data are represented as mean ± SEM. Unpaired T test, exact *P*-value indicated on the graph. ****$P < 0.0001$. (E) Microphotographs of CFSE-labeled tumors imaged in light sheet microscopy and segmentation of spreading profiles using Imaris software. Images were taken from 2 representative E4 chick embryos co-grafted with metastatic melanoma patient sample and hu-PBMCs from donor 1, then treated with anti-PD-1 (pembrolizumab) or control (NaCl 0.9%). Spreading distances are segmented in 3 color coded circles: d < 150 μm in blue, 150 μm < d < 300 μm in pink, d > 300 μm in green. Scale bars: 150 μm. (F) Representative FACS profiles of PD-L1 expression in a triple-negative breast cancer (TNBC) patient sample (BRE-01) and its combined positive score (CPS). (G) Histogram showing the quantification of volume of tumors formed in embryos co-grafted with TNBC patient (BRE-01) cells and hu-PBMCs from donor 1 in embryos treated with anti-PD-1 (pembrolizumab) or control (NaCl 0.9%). Dots represent volumes normalized to each embryo's BSA. $N = 1$ ($n = 10$ embryos in control group and $n = 6$ in pembrolizumab group). Data are represented as mean ± SEM. The percentage represents the average volume reduction compared to control. Mann–Whitney test, exact *P*-value indicated on the graph. ns: not significant. (H) Representative light sheet microscopy photographs illustrating CFSE+ tumors from colorectal patient samples (CRC-07 and CRC-08) co-engrafted with their Orange Cell Tracker+ autologous hu-PMBCs, in E4 chick embryos. Panels show the normalized volume (V) for CFSE (in green) and for OCT (in red) fluorescent signals obtained with Imaris software. Scale bars: 50 μm.

