## [Peer Review File · EMBO Molecular Medicine]

Humanized avian embryo models replicate an immune tumor environment for rapid immunotherapy studies

Marjorie Lacourrège, Loraine Jarrosson, Clélia Costechareyre, Mathis Marcel, Audrey Prunet, Mayrone Mongellaz, Camille Futelot, Cyrus Rouhani, Giang Nguyen, Thomas Aparicio, Lionel Le Bourhis, Céline Delloye-Bourgeois, Romain Teinturier, Valérie Castellani

Corresponding authors: Valérie Castellani (valerie.castellani@univ-lyon1.fr), Céline Delloye-Bourgeois (Celine.DELLOYE@lyon.unicancer.fr), Marjorie Lacourrège (mlacourrege@erbc-group.com), Romain Teinturier (rteinturier@erbc-group.com)

Review Timeline:

Submission Date:	10th Feb 25
Editorial Decision:	6th Mar 25
Revision Received:	18th Nov 25
Editorial Decision:	5th Dec 25
Appeal:	16th Dec 25
Editorial Decision:	4th Feb 26
Revision Received:	19th Feb 26
Accepted:	19th Feb 26

Editor: Lise Roth

Transaction Report:

6th Mar 2025

Dear Dr. Castellani,

Thank you for the submission of your manuscript to EMBO Molecular Medicine. We have now heard back from the referees who reviewed your manuscript. As you will see below, the reviewers raise substantial concerns on your work, which unfortunately preclude its publication in EMM in its current form.

The reviewers acknowledge the interest of the work, however referees #2 and #3 also raise several major concerns, including - but not limited to- comparison of the avian model with other models, absence of allogeneic experiments, limited mechanistic understanding regarding immune system activation, limited sample sizes, etc.

Following further discussion with the referees and within the team, we would like to invite revisions of the manuscript that would focus on the following points:

- Comparison with other preclinical models (humanized mice or organoids)
- Repeat key experiments using PBMCs from same donor.

We realize that these experiments might necessitate considerable time and work, and we are ready to extend the revision time to 6-8 months. If you would like to discuss further the points raised by the referees, I am available to do so via email or video. Let me know if you are interested in this option.

Acceptance of the manuscript will entail a second round of review. EMBO Molecular Medicine encourages a single round of revision only and therefore, acceptance or rejection of the manuscript will depend on the completeness of your responses included in the next, final version of the manuscript. For this reason, and to save you from any frustrations in the end, I would strongly advise against returning an incomplete revision.

We require:

- 1) A .docx formatted version of the manuscript text (including legends for main figures, EV figures and tables). Please make sure that the changes are highlighted to be clearly visible.
- 2) Individual production quality figure files as .eps, .tif, .jpg (one file per figure). For guidance, download the 'Figure Guide PDF' (<https://www.embopress.org/page/journal/17574684/authorguide#figureformat>).
- 3) At EMBO Press we ask authors to provide source data for the main figures. Our source data coordinator will contact you to discuss which figure panels we would need source data for and will also provide you with helpful tips on how to upload and organize the files.
- 4) A .docx formatted letter INCLUDING the reviewers' reports and your detailed point-by-point responses to their comments. As part of the EMBO Press transparent editorial process, the point-by-point response is part of the Review Process File (RPF), which will be published alongside your paper.
- 5) A complete author checklist, which you can download from our author guidelines (<https://www.embopress.org/page/journal/17574684/authorguide#submissionofrevisions>). Please insert information in the checklist that is also reflected in the manuscript. The completed author checklist will also be part of the RPF.
- 6) All Materials and Methods need to be described in the main text using our 'Structured Methods' format. According to this format, the Methods section includes a Reagents and Tools Table (listing key reagents, experimental models, software and relevant equipment and including their sources and relevant identifiers) followed by a Methods and Protocols section describing the methods, ideally using a step-by-step protocol format. The aim is to facilitate adoption of the methodologies across labs. Please download and fill our Reagents and Tools Table template (.docx), which you can find in our author guidelines: <https://www.embopress.org/page/journal/14693178/authorguide#structuredmethods>.
When submitting your revised manuscript, please do not include the Reagents and Tools Table in the Methods section of the manuscript but upload it as a separate file choosing the file type "Reagent Table".
An example of a Method paper with Structured Methods can be found here: <https://www.embopress.org/doi/10.15252/msb.20178071>
- 7) Please note that all corresponding authors are required to supply an ORCID ID for their name upon submission of a revised

manuscript.

8) It is mandatory to include a 'Data Availability' section after the Materials and Methods. Before submitting your revision, primary datasets produced in this study need to be deposited in an appropriate public database, and the accession numbers and database listed under 'Data Availability'. Please remember to provide a reviewer password if the datasets are not yet public (see <https://www.embopress.org/page/journal/17574684/authorguide#dataavailability>).

9) For data quantification: please specify the name of the statistical test used to generate error bars and P values, the number (n) of independent experiments (specify technical or biological replicates) underlying each data point and the test used to calculate p-values in each figure legend. The figure legends should contain a basic description of n, P and the test applied. Graphs must include a description of the bars and the error bars (s.d., s.e.m.). Please provide exact p values.

10) Our journal encourages inclusion of *data citations in the reference list* to directly cite datasets that were re-used and obtained from public databases. Data citations in the article text are distinct from normal bibliographical citations and should directly link to the database records from which the data can be accessed. In the main text, data citations are formatted as follows: "Data ref: Smith et al, 2001" or "Data ref: NCBI Sequence Read Archive PRJNA342805, 2017". In the Reference list, data citations must be labeled with "[DATASET]". A data reference must provide the database name, accession number/identifiers and a resolvable link to the landing page from which the data can be accessed at the end of the reference. Further instructions are available at .

11) We replaced Supplementary Information with Expanded View (EV) Figures and Tables that are collapsible/expandable online. EV Figures should be cited as 'Figure EV1, Figure EV2' etc... in the text and their respective legends should be included in the main text after the legends of regular figures.

12) The paper explained: EMBO Molecular Medicine articles are accompanied by a summary of the articles to emphasize the major findings in the paper and their medical implications for the non-specialist reader. Please provide a draft summary of your article highlighting

13) Author contributions: CRedit has replaced the traditional author contributions section because it offers a systematic machine readable author contributions format that allows for more effective research assessment. Please remove the Authors Contributions from the manuscript and use the free text boxes beneath each contributing author's name in our system to add specific details on the author's contribution. More information is available in our guide to authors.

Please also suggest a visual abstract to illustrate your article as a PNG file 550 px wide x 300-600 px high. A cropped portion of this image will serve as thumbnail for the table of content on our webpage.

16) As part of the EMBO Publications transparent editorial process initiative (see our Editorial at

<http://embomolmed.embopress.org/content/2/9/329>), EMBO Molecular Medicine will publish online a Review Process File (RPF) to accompany accepted manuscripts.

In the event of acceptance, this file will be published in conjunction with your paper and will include the anonymous referee reports, your point-by-point response and all pertinent correspondence relating to the manuscript. Let us know whether you agree with the publication of the RPF and as here, if you want to remove or not any figures from it prior to publication. Please note that the Authors checklist will be published at the end of the RPF.

I look forward to receiving your revised manuscript.

Yours sincerely,

Lise Roth

***** Reviewer's comments *****

Referee #1 (Remarks for Author):

This is a very interesting and detailed paper that report the feasibility of a tumor-immune cell model to assess efficacy of immunotherapy.

The experiment are well described and convincing. Nevertheless some limitation could be better discussed.

1) This is a first step of the concept but as there is obviously a variability of effect according to the donor the experiment on tumor sample would more convincing if the authors use the circulating immune cell of the patients itself. Moreover the model could compare circulating immune cell and tumor infiltrating immune cell of the patients. These perspectives should be discussed more deeply but for this paper additional experiment are not mandatory to my opinion.

2) In the experiment with human tumor the result of immunotherapy in the patient could be given. It would be interesting to corroborate the clinical result of immunotherapy in each patient and in the avian model. Especially, is there a tumor response after pembrolizumab in the patients with melanoma.

Indeed, the next step to validate this model it would be mandatory to compare the result of patient treatment with immunotherapy and the result on avian model on a large number of patients. It would be particularly interesting to elucidate the cause of discrepancy between human treatment and avian model.

3) The anti-metastatic effect of pembrolizumab is highly speculative

Referee #2 (Remarks for Author):

The study presents an innovative approach to modeling human tumor-immune interactions using avian embryos. However, several critical issues must be addressed before it can be considered for publication in EMBO Molecular Medicine.

Major Concerns:

- While the use of avian embryos for humanized tumor-immune modeling is novel, the concept of humanizing animal models for immunotherapy studies is not new. Humanized mouse models have been widely used for similar purposes. The authors should clearly articulate how their model surpasses existing systems in terms of throughput, cost, and predictive power.
- The study lacks a direct comparison with humanized mouse models or immune organoids (DOI:10.1016/j.bioactmat.2024.10.010), making it difficult to assess the true novelty and superiority of the avian embryo system.
- The study relies heavily on allogeneic PBMCs, which may not fully replicate autologous immune-tumor interactions. Although the authors mention the potential for using autologous cells, this was not demonstrated experimentally, limiting the translational relevance.
- The avian embryo tumor microenvironment may differ significantly from that of humans. The absence of functional avian immune cells during the experimental window may limit the model's ability to fully recapitulate human immune-tumor dynamics.
- While the study shows tumor-PBMC interactions and T-cell activation, the molecular mechanisms underlying these processes remain unexplored. Key questions include:
 - o What are the primary chemokines or adhesion molecules involved in PBMC recruitment?
 - o What evidence supports antigen presentation in this model? Do the authors also observe dendritic cells?
 - o How do tumor cells suppress T-cell activation? Are PD-L1/PD-1 interactions the only mechanism, or are other immune checkpoints involved?
 - o Do infiltrating PBMCs exhibit cytotoxic activity against tumor cells? Are there changes in cytokine profiles or immune exhaustion markers over time?
- Sample sizes for some experiments (e.g., patient-derived xenografts) are small, limiting statistical power and generalizability.
- The short experimental timeframe (48-72 hours) may not capture long-term immune-tumor dynamics, such as immune memory or resistance development.

- The study lacks a thorough dose-response evaluation of pembrolizumab.
- There is no control for potential avian embryonic environmental effects on PBMC behavior, and a comparison with non-humanized avian embryos is missing.
- The model's ability to predict patient-specific responses to anti-PD-1 therapy is constrained by the small sample size and lack of long-term follow-up. Additionally, its applicability to combination therapies, which are increasingly used in the clinic, remains unclear.
- An ethical statement regarding animal welfare and human PBMC use (isolated from four different healthy donors) is missing. This omission should be addressed.

Minor Concerns:

- Although the manuscript is well-written, certain aspects require further clarification, such as the rationale for using chick embryos to study the tumor microenvironment and predict immunotherapy efficacy.
- The authors overstate the ability of the avian model to fully recapitulate the immune tumor microenvironment without considering the role of other stromal components, such as fibroblasts. Their conclusions should be adjusted accordingly.
- Figures and data presentation are generally strong, particularly the use of light-sheet microscopy for 3D imaging, but some data points lack statistical validation.
- The authors should explicitly discuss the limitations of their model in the discussion section.

Referee #3 (Remarks for Author):

The study by Lacourrège et al. presents a humanized patient-derived xenograft (AVI-PDX) model in avian embryos intending to recreate a human tumor immune microenvironment. The model uses co-implantation of human tumor cells and peripheral blood mononuclear cells (PBMCs) or subsequent intravenous injection of PBMCs. Tumor immune infiltration, the efficacy of anti-PD-1 immunotherapy (pembrolizumab), and response heterogeneity in solid cancers such as melanoma, colorectal cancer (CRC), and triple-negative breast cancer (TNBC) are analyzed.

Humanizing avian models to evaluate immunotherapies in human tumors is certainly novel. However, the scientific justification for the choice of the avian model is not completely clear, and the absence of robust comparisons with routinely used preclinical models (such as humanized mice) compromises the relevance and validity of the work. Although the research emphasizes the model's speed, this feature is irrelevant if the physiological and clinical relevance of the findings is not proven. Speed without precision does not provide scientific value.

One of the main methodological weaknesses of the study is the lack of consistency in the use of PBMCs from the same donors throughout the different experiments. Immunotherapy is deeply influenced by the individual variability of the immune system, and this lack of uniformity introduces a significant source of confusion in the results. Without the use of PBMCs from the same donor in comparative experiments, it is not possible to discern whether the observed differences are attributable to the experimental model, the type of tumor, or interdonor variability. This methodological inconsistency compromises the reproducibility and interpretation of the results, which decreases the robustness of the conclusions drawn.

It would be highly recommended that the authors repeat the key experiments using PBMCs from the same donors for each type of analysis. This would allow:

1. Reducing the experimental variability attributed to the donor's immune system.
2. Ensure direct and valid comparisons between different experimental conditions (response to different tumor cell lines or immunotherapy treatments).
3. Explore in a controlled manner the relevance of HLA and other donor immunological characteristics in the antitumor response, which could provide valuable information for patient stratification.

The study lacks more robust negative and positive controls, such as experiments without human PBMCs or with non-allogeneic PBMCs.

The authors focus solely on the presence of human immune cells without delving into functional evidence that these cells contribute to an effective antitumor immune response. The study lacks evidence demonstrating functional activation of the human immune system, such as cytotoxicity assays or cytokine secretion profiles.

In its current state, this manuscript does not meet the standards required for publication. The conceptual deficiencies, methodological flaws, and inadequate analysis are severe enough to warrant rejection of the work. Unless a thorough review is performed, with experimental redesign, robust functional validations, sample expansion, and solid statistical analysis, the study lacks scientific and clinical merit.

Point-to-point answer

We are grateful for the opportunity to revise and resubmit our manuscript and thank the editor and reviewers for the time and effort devoted to its evaluation. We have carefully considered all comments and revised the manuscript to address the concerns raised and to clarify several points that required further explanation.

We thank all the reviewers for their constructive and thoughtful feedback. Their detailed observations have helped us clarify key aspects of our work, refine our interpretations, and improve the overall quality of the manuscript. We have addressed each point as carefully as possible and believe that the revised version has been substantially improved as a result.

Prior to the point-to-point, a synopsis of the experiments, analyses and modifications that have been conducted in the revision is appended below.

LIST OF EXPERIMENTS PERFORMED DURING THE REVISION

- **Experiments to characterize hu-PBMCs from donor 8:** grafting of hu-PBMCs and analysis of embryo survival, hu-PBMCs survival and comparison of the immune cell populations between pre-grafted and post-grafted samples by flow cytometry.

Data included in Fig 1C, 1D, 1F, 1G.

- **Experiments to analyze activation and exhaustion markers in the AVI infiltration model:**

IV injection of hu-PBMCs from donor 5 in batches of grafted embryo with MDA-MB-231 cell line, flow cytometry to quantify activation and exhaustion markers of hu-PBMCs infiltrated in tumor and control tumor-free tissue.

Data included in Fig 2J, 2K, EV1A.

- **Experiments to homogenize the hu-PBMC donors in all experiments:**

1) Grafting of hu-PBMCs from donor 8 alone, co-grafting with either MDA-MB-231 or HCT 116 cell lines, analysis of immune cell subpopulations evolution in pre-grafted and post-grafted samples. These experiments with donor 8 replaced initial experiments achieved with different donors.

Data included in Fig 3G.

2) Embryos co-grafted with MDA-MB-436 or with HCT 116, and hu-PBMCs from donor 8 and treated with pembrolizumab or control, for analysis of tumor volumes.

Data including in figure 5D.

3) Co-grafting of CRC-03 with hu-PBMCs from donor 8, treated with pembrolizumab or control, for assessment of pembrolizumab effect on tumor volume. Immunolabeling and imaging with light sheet/confocal microscopy of the tumor microenvironment.

Data including in Fig 6A, 6B, 6C, 6E.

- **Experiment to evaluate the impact of hu-PBMCs on the tumor in the co-grafting model:** Grafting of MDA-MB-231 with and without hu-PBMCs, and comparative analysis of tumor volumes.

Data included in Fig 3J.

- **Experiment to evaluate the impact of hu-PBMCs on the tumor in the infiltration model:** Grafting of MDA-MB-231 and intravenous injection or not of hu-PBMCs, comparative analysis of tumor volumes.

Data included in Fig EV1B.

- **Experiments to study anti-PD-1 efficacy at several doses:** Treatment of embryos co-grafted with MDA-MB-231 et hu-PBMCs from donor 8, and evaluation of 3 doses of pembrolizumab by analysis of tumor volume.

Data included in Fig 5A and in Fig EV2C.

- **Experiment to study the effect of anti-PD-1 in non-humanized AVI models:** Grafting of MDA-MB-231 cells in absence of immune cells into batches of avian embryos treated with either pembrolizumab or control and analysis of tumor volumes.

Data included in Fig EV3 A.

- **Experiment to demonstrate that anti-PD-1-induced tumor volume reduction is mediated by T cells:** Treatment with either pembrolizumab or control of batches of embryos co-grafted with MDA-MB-231 and hu-PBMC from donor 8. The mix of cells was either grafted directly or pre-incubated with anti-MHC I antibodies. Comparative analysis of tumor volumes.

Data included in Fig 5C.

- **Experiments to study combination therapies including anti-PD-1, by testing two combinations: anti-PD-1 + gemcitabine-carboplatin, and anti-PD-1 + docetaxel-carboplatin:** Treatment of embryos co-engrafted with MDA-MB-231 et hu-PBMCs from donor 8, with either control, pembrolizumab, chemotherapies (gemcitabine-carboplatin or docetaxel-carboplatin) or tritherapy (pembrolizumab- gemcitabine-carboplatin or pembrolizumab-docetaxel-carboplatin) for analysis tumor volume reduction.

Data included in Fig 5H-I.

Experiments to set AVI models with patient tumors and autologous immune cells: Co-grafting of 3 patient samples (2 CRC and 1 TNBC) with their autologous hu-PBMC. Immunolabeling and imaging with light sheet/confocal microscopy of the tumor microenvironment.

Data included in Fig 6G, 6H and EV4H.

- **Experiments to assess anti-PD-1 efficacy in autologous immune-cancer AVI models:** Grafting of the samples with their autologous hu-PBMCs in batches of embryos, treatment with excipient or pembrolizumab, analysis of tumor volumes.

Data included in Fig 6I.

NEW ANALYSES OF THE DATA

Initial figure#	Revised figure#	New analyses
1F	1F	Statistical analyses performed
1G	1G	Statistical analyses performed
2D	2C	Statistical analyses performed
2C	2D	Statistical analyses performed
--	2H	Comparison of the immune cell populations from donor 5, between pre-grafted and infiltrated immune cells in tumors of embryos grafted with MDA-MB-231 cell line.
2G	2I	Statistical analyses performed.
--	6F	Representation and statistical analysis of tumor volume reduction based on MSI/MSS status of the CRC patient sample.

C) TEXT MODIFICATION IN INTRODUCTION AND DISCUSSION

- Introduction of general features of the avian models
- Discussion of new data
- Discussion of the tumor microenvironment provided by avian models
- Discussion of anti-PD-1 mode of action
- Discussion of the relevance of AVI-PDX models
- Discussion of the advantages and limitations of the AVI models, compared to existing mouse models
- Introduction of an ethical statement

Referee #1

This is a very interesting and detailed paper that report the feasibility of a tumor-immune cell model to assess efficacy of immunotherapy. The experiment are well described and convincing. Nevertheless, some limitation could be better discussed.

1) This is a first step of the concept but as there is obviously a variability of effect according to the donor the experiment on tumor sample would more convincing if the authors use the circulating immune cell of the patients itself. Moreover, the model could compare circulating immune cell and tumor infiltrating immune cell of the patients. These perspectives should be discussed more deeply but for this paper additional experiment are not mandatory to my opinion.

We fully understand the relevance of humanized models incorporating autologous immune cells. We now present the experimental evidence that the model can be engineered using patient tumors and their autologous PBMCs. This proof of concept was established using two CRC and one TNBC cases. We imaged CD45+ immune cells and CFSE labeled cancer cells in transverse sections of grafted embryos. We also pre-labelled the immune fraction and the cancer cell fraction in two colors prior to grafting for imaging in whole embryos. We could document the presence of both cell types. We also conducted anti-PD-1 efficacy studies. In all three cases, the tumors showed no response to anti-PD-1 administration. As there were no markers predicting sensitivity to anti-PD-1 that we could use, these samples were selected at random. Therefore, this result may either reflect resistance of cancer cells or inefficiency of autologous T cells. We have included these data in the results page 17, Figure 6I) to demonstrate the technical feasibility of these models and we introduce in the discussion page 23 how they pave the way for further investigation of the response mechanisms and the search for predictive biomarkers. It was not manageable to conduct a study comparing circulating and infiltrated immune cells. We believe this goes beyond the scope of this first work but we indicate this as a perspective of use of the AVI models at the end of the discussion.

2) In the experiment with human tumor the result of immunotherapy in the patient could be given. It would be interesting to corroborate the clinical result of immunotherapy in each patient and in the avian model. Especially, is there a tumor response after pembrolizumab in the patients with melanoma.

Indeed, the next step to validate this model it would be mandatory to compare the result of patient treatment with immunotherapy and the result on avian model on a large number of patients. It would be particularly interesting to elucidate the cause of discrepancy between human treatment and avian model.

We agree that a clinical study comparing patient responses with those of their corresponding replicas in the avian model would be of significant value. At this stage, however, our focus has been on establishing the model using samples obtained from hospital biological resource centers. The majority of these samples were collected at diagnosis, prior to any treatment. To ensure compliance with personal data protection regulations, all samples are anonymized, and we only have access to basic tumor characteristics such as grade and molecular markers. One post-treatment sample was included in our study, and in this case, we know it represents a relapse following chemotherapy. We highlighted this limitation and presented it as an important future direction in the discussion.

3) The anti-metastatic effect of pembrolizumab is highly speculative

We fully agree that our data should not be over-interpreted. Our objective was to illustrate that the response to anti-PD-1 administration can vary depending on the sample. We have taken extra precautions in our interpretation, adding to the discussion this sentence: "Whether for this patient tumor, the observed reduced migration of melanoma cells would result in decreased metastasis and to which extend this anti-PD-1 mediated effect could be generalized remains to be determined."

Referee #2

The study presents an innovative approach to modeling human tumor-immune interactions using avian embryos. However, several critical issues must be addressed before it can be considered for publication in EMBO Molecular Medicine.

Major Concerns:

- While the use of avian embryos for humanized tumor-immune modeling is novel, the concept of humanizing animal models for immunotherapy studies is not new. Humanized mouse models have been widely used for similar purposes. The authors should clearly articulate how their model surpasses existing systems in terms of throughput, cost, and predictive power.

We now provide an extended discussion and comparison of these aspects between the AVI models and mouse models to more clearly highlight the pros and cons of each, page 22.

- The study lacks a direct comparison with humanized mouse models or immune organoids (DOI:10.1016/j.bioactmat.2024.10.010), making it difficult to assess the true novelty and superiority of the avian embryo system.

We fully agree that methodologies alternative to animal models hold important advantages, especially those of providing models to study human biology. To compare the AVI model and organoids as those reported in the cited publication, we would need to engineer immune mini-organs with cancer cells. This represents a complex process to set up. In absence of experimental evidence for comparison of AVI models with organoids, it was difficult to discuss this question in the manuscript.

Regarding the comparison with mouse models, we chose to set up our AVI models with two cell lines, HCT 116 and MDA-MB-231 that were reported in mouse models to respond to anti-PD1 (Capasso et al, 2019; Wang et al, 2018; Shang et al, 2022).

This information is stated in page 8. In line with the previous point, we extended the discussion, more clearly assessing the strengths and weaknesses of the different models, in order to facilitate the understanding of our approach.

- The study relies heavily on allogeneic PBMCs, which may not fully replicate autologous immune-tumor interactions. Although the authors mention the potential for using autologous cells, this was not demonstrated experimentally, limiting the translational relevance.

We fully agree that testing autologous immune-cancer sample pairs is of high interest. We could obtain from clinical collaborators 2 CRC and 1 TNBC, each with their autologous PBMCs, that were grafted in avian embryos. We imaged immune cells and CFSE labeled cancer cells in transverse sections of grafted embryos. We also pre-labelled the immune fraction and the cancer cell fraction in two colors prior to grafting for imaging in whole embryos. These experiments validated the possibility of engineering the AVI-model with cancer-PBMCs sample pairs.

We then conducted efficacy studies with pembrolizumab by measuring tumor volumes. None of the cases were found to be responsive to the treatment. As there were no markers predicting sensitivity to anti-PD-1 that we could use, these samples were selected randomly. Therefore, this result may either reflect resistance of cancer cells or inefficiency of autologous T cells. We have included these data in the results page 17 (Figure 6I), to show the technical feasibility of these models and we mention page 23 how they could be used for translational studies.

- The avian embryo tumor microenvironment may differ significantly from that of humans. The absence of functional avian immune cells during the experimental window may limit the model's ability to fully recapitulate human immune-tumor dynamics.

We concur with the reviewer's comment that our initial manuscript did not provide adequate information regarding the avian embryo microenvironment. On the one hand, the absence of a

functional adaptive immune system is a key factor facilitating tumor engraftment and growth. This nevertheless offers a distinct advantage by allowing transplanted human immune cells to engraft within the tissues and exert their functions. On the other hand, the innate immune environment in the chick is existing from very early stages. The embryo is already populated by waves of primitive hematopoietic progenitors that give rise to macrophages. These cells contribute to tissue maturation. Nevertheless, it is important to note that the avian microenvironment does not fully replicate the complexities of the human immune system. We discuss the avian embryo microenvironment in a specific paragraph page 20.

- While the study shows tumor-PBMC interactions and T-cell activation, the molecular mechanisms underlying these processes remain unexplored.

Key questions include:

- o What are the primary chemokines or adhesion molecules involved in PBMC recruitment?

We fully understand the relevance of these questions. The primary objective of this study was to demonstrate that human PBMCs can be effectively recruited to the tumor site. We did not, at this stage, investigate the underlying mechanisms driving the tropism of immune cells toward tumor cells, which are likely to vary depending on the specific characteristics of the cancer cells involved. Addressing this point would require a dedicated study.

- o What evidence supports antigen presentation in this model?

To address this question, we performed experiments to study if an anti-MHC I antibody could block the anti-tumoral effect observed in condition of anti-PD-1 administration. Half of the mix of hu-PBMCs/cancer cells was directly grafted in avian embryos while the other half was pre-incubated with anti-MHC I antibody prior to grafting. Then, half of the embryos were treated with pembrolizumab, the remaining half with excipient. Interestingly, we could show that the anti-tumoral effect mediated by pembrolizumab was abolished by the anti-MHC I pre-treatment.

We also completed an experiment related to the infiltration model to further evidence the communication of T cells and cancer cells. We studied if T cells infiltrated in the tumor after hu-PBMC intravenous injection express activation and exhaustion markers. We found an increase of CD69 in T cells infiltrating the tumor, when compared to the T cell fraction that infiltrated the equivalent tissue of tumor-free embryos. These CD3⁺ cells showed very little expression of TIM-3 and PD-1 exhaustion markers. Thus, infiltrated T cells are activated in presence of cancer cells and are likely not yet inhibited at this time point.

- o Do the authors also observe dendritic cells?

We characterized post-grafted hu-PBMC subpopulations using antibodies that recognize T cells (CD3CD8 and CD3CD4), NK cells (CD56 and CD16), B cells (CD19) and monocytes (CD14). Approximately 90% of the cells were found to be within these subpopulations, with some variations observed among the donors. This is consistent with the reported proportions of hu-PBMCs from healthy donors. The cells remaining after the initial markers have been ruled out are likely to be dendritic cells. These may have been present in the human peripheral blood mononuclear cell (hu-PBMC) samples, or may derive from monocytes in tumor tissue. It is also possible that some of these cells are plasma B cells, which may have emerged in tumor tissue.

- o How do tumor cells suppress T-cell activation? Are PD-L1/PD-1 interactions the only mechanism, or are other immune checkpoints involved?

We studied the influence of cancer cells on T cells by detection of both PD-1 and TIM-3. We show an involvement of PD-1/PD-L1, which does not exclude a contribution of TIM-3-Galectin-9 interactions. For CTLA-4, we observed a significant expression level of this marker in micro-dissected tumors enriched with hu-PBMCs. Additional investigations would be needed to deeply address this point, which we could not conduct, as we had to make priorities in the revision process.

- o Do infiltrating PBMCs exhibit cytotoxic activity against tumor cells? Are there changes in cytokine profiles or immune exhaustion markers over time?

To answer this question, we performed an experiment consisting in grafting cancer cells in avian embryos and performing or not intravenous injection of hu-PBMCs. We quantified the tumor volumes and found no statistically significant impact of hu-PBMCs injection. However, the tumor volume

tended to be reduced, which suggests that infiltrated T cells may manifest moderate cytotoxic activity against the tumor.

In our initial work, we did not analyze the activation and exhaustion profile of immune cells in the infiltration model. We thus performed experiments to address this question. As control, we used equivalent tissue of embryos bearing no tumor. We found that CD69 levels increased in T cells infiltrated in the tumor, compared to the control, thus reflecting an early activation of these cells. Exhaustion markers (TIM-3 and PD-1) were in contrast comparable in both conditions, suggesting that T cells were not yet driven by cancer cells towards exhaustion. This may be due to the limited time window for immune-cancer communication, which can only occur after the infiltration process is complete. This finding is further corroborated by the outcomes of the co-grafting procedure. During this process, which permits communication between the two cell types from the start of the procedure, an increase in markers of exhaustion was observed. Analyzing cytokine profiles would be highly challenging, and potentially impossible, given the limited sample size.

- Sample sizes for some experiments (e.g., patient-derived xenografts) are small, limiting statistical power and generalizability.

In the present study, our primary objective was to provide evidence that our humanization approaches can be engineered with patient samples. In terms of generalizability, we built on our previous work, which demonstrated that a greater number of patient samples and cancer-types can be engrafted in avian embryos. Our previous published work reported 20 follicular lymphoma, 10 neuroblastoma, 12 TNBCs and 5 melanomas, and additional cases are yet to be published.

This work demonstrated that tumor intake and growth success is high in the avian embryo environment and that it is very well suited to small size samples. In the present study, for all experiments, each patient was considered as an individual case, with outcomes of various manipulations being compared between embryos engrafted with the same patient sample. Inter-patient comparisons were not made.

In the revised manuscript, we have attempted to avoid over-interpretation of the data, have paid close attention to clearly stating the goals of these experiments based on patient samples, and have discussed in the manuscript how these PDX models could be used in future studies.

- The short experimental timeframe (48-72 hours) may not capture long-term immune-tumor dynamics, such as immune memory or resistance development.

We fully agree with this comment. It is addressed in the paragraph discussing the advantages and limitations of the AVI models page 22 that we added to the revised manuscript.

- The study lacks a thorough dose-response evaluation of pembrolizumab.

In the initial study, to select the administration dose for efficacy tests, we performed a dose-escalation assay ranging from 1 mg/ml to the undiluted product (25 mg/ml). The 5mg/ml dose was selected based on the results of this assay. Thus, to further extend this study, we performed efficacy tests with two additional doses, framing the initially selected dose. We found that the lower dose had no impact on tumor volumes, and that the highest one (25 mg/ml) was not more effective than the 5 mg/ml dose in reducing them. These data are presented in Fig 5A. This is consistent with our observation from flow cytometry analysis that PD-L1 levels in T cells of the tumors were not different between 5 mg/ml and 25 mg/ml.

- There is no control for potential avian embryonic environmental effects on PBMC behavior, and a comparison with non-humanized avian embryos is missing.

To evaluate the impact of the avian embryonic environment, our unique option is to compare hu-PBMCs collected from avian embryos to the initial pre-grafted PBMC sample. These comparisons were made in many of the experiments.

Regarding the second point (non-humanized condition), we performed two sets of experiments:

- 1) Comparing the tumor volumes in conditions of grafting of cancer cells either alone (non-humanized) or combined with hu-PMCS (humanized). We achieved this experiment using MDA-MB-231 cell line and 2 donors already validated in the study. In both cases, we found no statistically significant impact of the hu-PBMCs on the tumor volumes. The data are presented on page 10 and in Fig 3J.

2) Analyzing the tumor volumes in avian embryos engrafted with cancer cells only (non-humanized condition) and treated with anti-PD-1 (pembrolizumab). The experiments were performed with MDA-MB-231 cell line. We found in this context that the pembrolizumab had no impact on the tumor volumes, which shows that its effect requires the presence of hu-PBMCs. The data have been included on page 12 and in Fig EV3 A.

- The model's ability to predict patient-specific responses to anti-PD-1 therapy is constrained by the small sample size and lack of long-term follow-up. Additionally, its applicability to combination therapies, which are increasingly used in the clinic, remains unclear.

With regard to the first point, we acknowledge that our work does not address the question of the predictability of patient responses. Such a study would require comparing patient responses with those of their avian counterparts, which is beyond the scope of this initial study. Nevertheless, we believe that our current findings demonstrate the feasibility of such studies, particularly with regard to colorectal cancer, and lay the groundwork for searching for new predictive markers of patient responses.

Second, we fully concur that demonstrating the model's applicability to combination therapies is a key step and is fully aligned with the goals of the present work. We addressed this point with the MDA-MB-231 AVI model and donor 8, a combination that we found effective to reveal pembrolizumab-mediated anti-tumoral volume. In the clinic, anti-PD-1 has been combined with several chemotherapies, including docetaxel/carboplatin and gemcitabine/carboplatin, to treat metastatic TNBC patients (Cortes et al, 2020; Sharma et al, 2024). The MDA-MB-231 cell line was reported sensitive to docetaxel/carboplatin, but poor responder to gemcitabine (Chen et al, 2014; Di et al, 2018). We thus set-up two tri-therapeutic treatments, combining pembrolizumab with either docetaxel/carboplatin or with gemcitabine/carboplatin and compared their effect to those of anti-PD-1 monotherapy and of bi-chemotherapy. We found that tumor volumes were unaffected by gemcitabine/carboplatin treatment, likely due to the poor response of MDA-MB-231 cells to gemcitabine. This suggests that combining it with carboplatin does not overcome this resistance. As observed in our previous experiments, pembrolizumab induced a significant reduction in tumor volume. This efficacy was preserved in co-administration with gemcitabine/carboplatin, with no additional benefit from the combination therapy. For the second combination therapy, we found that both the individual administration of docetaxel/carboplatin and pembrolizumab resulted in a significant decrease in tumor volumes compared to the control group. Interestingly, in this case, co-administration tended to be even more efficient than the individual treatments. These data are presented on page 14 as well as in Figure 5H and 5I.

- An ethical statement regarding animal welfare and human PBMC use (isolated from four different healthy donors) is missing. This omission should be addressed.

We apologize for this omission. We have added the following statement in the revised manuscript on page 26. "Our experiments on avian embryos were performed between embryonic days 2 and 4, i.e. before the final third of gestation, in compliance with EU Directive 2010/63/EU (formerly 86/609/EEC). Specific ethical authorization was not required. The study fully adheres to the 3Rs principles. Experiments were limited to those not covered by existing data and were carried out according to carefully pre-planned protocols. All procedures followed the approved guidelines, favoring early developmental stages when pain perception is ineffective. Whole-organism studies in avian embryos were necessary in order to model cancer in an environment that mimics the human condition, as well as the mode of administration of therapies."

We also included a statement related to the use of hu-PBMCs.

Minor Concerns:

- Although the manuscript is well-written, certain aspects require further clarification, such as the rationale for using chick embryos to study the tumor microenvironment and predict immunotherapy efficacy.

- The authors overstate the ability of the avian model to fully recapitulate the immune tumor microenvironment without considering the role of other stromal components, such as fibroblasts. Their conclusions should be adjusted accordingly.

With regard to these two points, we acknowledge that our initial manuscript did not provide sufficient information on the avian microenvironment and its differences and similarities with the human context. We have now added a brief summary of the properties of the AVI models, based on our previous work, to the introduction on page 3. We have also included a full paragraph dedicated to the microenvironment on page 20, stating that differences between developing and mature tissues, as well as inter-species differences, can impact cancer cell responses. The discussion section also describes the advantages and limitations of the model. Overall, we hope that these additions will improve understanding of the relevance of the AVI models.

- Figures and data presentation are generally strong, particularly the use of light-sheet microscopy for 3D imaging, but some data points lack statistical validation.

We re-examined all the figures. Statistics were provided for measures of tumor volumes from light sheet microscopy. However, we found that this was not the case for one experiment presented in Fig. 3E, which reports the proportion of immune cells in contact with cancer cells in 10 individual embryos. We now provide statistics showing that inter-individual variability is not statistically significant.

- The authors should explicitly discuss the limitations of their model in the discussion section.

As mentioned in an earlier response, we have added a new paragraph to the discussion to address this issue.

Referee #3

The study by Lacourrège et al. presents a humanized patient-derived xenograft (AVI-PDX) model in avian embryos intending to recreate a human tumor immune microenvironment. The model uses co-implantation of human tumor cells and peripheral blood mononuclear cells (PBMCs) or subsequent intravenous injection of PBMCs. Tumor immune infiltration, the efficacy of anti-PD-1 immunotherapy (pembrolizumab), and response heterogeneity in solid cancers such as melanoma, colorectal cancer (CRC), and triple-negative breast cancer (TNBC) are analyzed. Humanizing avian models to evaluate immunotherapies in human tumors is certainly novel.

However, the scientific justification for the choice of the avian model is not completely clear, and the absence of robust comparisons with routinely used preclinical models (such as humanized mice) compromises the relevance and validity of the work. Although the research emphasizes the model's speed, this feature is irrelevant if the physiological and clinical relevance of the findings is not proven. Speed without precision does not provide scientific value.

One of the main methodological weaknesses of the study is the lack of consistency in the use of PBMCs from the same donors throughout the different experiments. Immunotherapy is deeply influenced by the individual variability of the immune system, and this lack of uniformity introduces a significant source of confusion in the results. Without the use of PBMCs from the same donor in comparative experiments, it is not possible to discern whether the observed differences are attributable to the experimental model, the type of tumor, or inter-donor variability. This methodological inconsistency compromises the reproducibility and interpretation of the results, which decreases the robustness of the conclusions drawn.

It would be highly recommended that the authors repeat the key experiments using PBMCs from the same donors for each type of analysis.

This would allow:

1. Reducing the experimental variability attributed to the donor's immune system.
2. Ensure direct and valid comparisons between different experimental conditions (response to different tumor cell lines or immunotherapy treatments).
3. Explore in a controlled manner the relevance of HLA and other donor immunological characteristics in the antitumor response, which could provide valuable information for patient stratification.

We fully acknowledge the concerns raised and recognize that the way our data were initially presented may have contributed to several of the general issues identified. In addition to the need

for improved clarity, we also acknowledge that some key experiments were missing. These are essential to enhance dataset consistency and enable more direct comparisons. We have carefully reviewed the entire set of experiments to identify donor-related inconsistencies and have performed the following additional experiments:

In figure 3: panels G and H comparing pre-grafted, post-grafted (hu-PBMCs alone) and post-co-grafted (hu-PBMCs with MDA-MB-231 or HCT 116) to monitor the PBMC subpopulations, the 2 cell lines were engrafted with a different donor. We performed a novel experiment in which the 2 cell lines were engrafted with the same donor, donor 8. This donor was already used in the study and prove to be competent for anti-PD-1 efficacy. Survival of embryos, of CD45+ and the characterization of immune cell populations with this donor have been added in Fig 1 for more consistency between data.

In Figure 5: the donor for the two TNBC cell lines was the same, but different to that used for the HCT-116 cell line. We therefore conducted further experiments and are now presenting the anti-PD1 efficacy on the three studied cell lines using the same donor (donor 8).

In Figure 6: The graphs show the results for a pool of six CRC patient tumors. We used donor 8 for all of them except sample 3. We therefore re-examined the response of this sample in experiments conducted using donor 8.

With these new experiments, we now fully homogenize the use of donors throughout the study, also including the experiments dedicated to investigating the inter-donor variability.

The study lacks more robust negative and positive controls, such as experiments without human PBMCs or with non-allogeneic PBMCs.

We acknowledge that some important controls were missing. We conducted further experiments to test the efficacy of anti-PD-1 in a non-humanized condition lacking hu-PBMCs. We found that, in this context, tumor volumes were unaffected by the treatment (Fig EV3A). We also investigated whether co-grafting hu-PBMCs with tumor cells affects tumor volume compared to grafting tumor cells only and found no statistical difference between the two conditions (Fig 3J).

In the context of non-allogeneic PBMCs, we created models using autologous PBMCs and cancer cells for three patients. Using confocal and light sheet microscopy, we demonstrate that autologous cells can survive and integrate with tumor cells in avian tissues (6G, 6H and EV4H). We tested the outcome of intravenous injection of pembrolizumab. We documented no response in any of the cases. This may not necessarily be due to the technical process, as the tumors may not respond to anti-PD-1, and/or the immune cells may not be sufficiently competent. These experiments are presented in Fig. 6I. We were careful not to overinterpret the data, focusing instead on how they illustrate the technical feasibility of implementing autologous immune cells in humanized AVI models.

The authors focus solely on the presence of human immune cells without delving into functional evidence that these cells contribute to an effective antitumor immune response. The study lacks evidence demonstrating functional activation of the human immune system, such as cytotoxicity assays or cytokine secretion profiles.

We have previously attempted to assess T-cell cytotoxicity using immunodetection of the CD107 granulocyte marker in flow cytometry. We know that this marker is very transient and must be measured during the degranulation process, which is highly challenging. Unsurprisingly, we failed to obtain a result.

Miniaturization is a limiting factor in the dosage of cytokines in blood samples or tumor lysates using ELISA. To provide further evidence, we decided to investigate whether interfering with antigen presentation by T lymphocytes using an anti-MHC I antibody could abolish the reduction in tumor volumes induced by pembrolizumab. We pre-treated the mix of hu-PBMCs and tumor cells with anti-MHC I and engrafted it in avian embryos. A control mix that was not treated was engrafted in parallel. We analyzed the tumor volumes in the embryos treated with excipient or pembrolizumab and found that pre-treatment with anti-MHC I resulted in a loss of the pembrolizumab effect. The data are presented in Fig. 5C and discussed on page 13.

In its current state, this manuscript does not meet the standards required for publication. The conceptual deficiencies, methodological flaws, and inadequate analysis are severe enough to warrant rejection of the work. Unless a thorough review is performed, with experimental redesign, robust functional validations, sample expansion, and solid statistical analysis, the study lacks scientific and clinical merit.

We believe that our extensive revision has significantly improved our study, and that the additional controls and new experiments provide greater strength to the data and their interpretation.

4th Dec 2025

Decision on your manuscript EMM-2025-21404-V2

Dear Dr. Castellani, Dear Valerie,

Thank you for the submission of your revised manuscript to EMBO Molecular Medicine, which has been reviewed by Referees #2 and #3.

As you will see from the reports below, Referee #2 is satisfied overall with the revisions, but regrets the lack of comparison with existing humanized animal models. Referee #3 appreciates the efforts made to address the initial concerns but nevertheless mentions remaining methodological and conceptual limitations.

As EMBO Press only allows one round of experimental revisions and addressing the remaining concerns would require considerable time and effort, unfortunately I see no option but to return the manuscript to you at this stage with the decision that we cannot offer to publish it.

While we cannot pursue this manuscript further, we encourage you to transfer your study to our not-for-profit open-access sister journal, Life Science Alliance (LSA). We shared your manuscript and the accompanying reviews with LSA Executive Editor, Tim Fessenden, who is interested in these findings. He is pleased to offer publication of this manuscript at LSA pending verification that the current limitations and potential future work, described in the manuscript, reflects the comments of the reviewers in this regard. We encourage you to contact Dr. Fessenden at t.fessenden@life-science-alliance.org to discuss this work and the revisions requested. You may use the link below to immediately transfer your manuscript to LSA. You do not need to revise the manuscript before transferring it to LSA. Once you transfer, you will receive an invitation to revise and resubmit. Again please contact Dr. Fessenden if you have any questions about the LSA journal, the transfer process, or the revisions requested.

I am very sorry to disappoint you on this occasion and I hope you will view the possibility of a transfer favorably. If this is the case, please use the link below to transfer the manuscript directly.

With kind regards,

Lise

Lise Roth
Senior Editor
EMBO Molecular Medicine

***** Reviewer's comments *****

Referee #2 (Comments on Novelty/Model System for Author):

The ethical issues have been resolved. The previously missing ethics statement has now been provided.

Referee #2 (Remarks for Author):

The authors have satisfactorily addressed most of my questions, though a few points remain unresolved due to technical limitations. In particular, the comparison with existing humanized animal models could not be provided, which is understandable but leaves an important contextual gap. This limitation may influence how the study is interpreted and could affect the broader applicability of this avian model for tumor biology research.

Referee #3 (Comments on Novelty/Model System for Author):

The manuscript demonstrates technically competent experimental work, but several methodological and conceptual limitations reduce its overall impact. Technical quality is rated as Medium, as the authors present detailed experimental procedures and new data; however, challenges remain regarding donor variability, lack of functional validation, and incomplete statistical rigor. The novelty is also Medium, since the humanized avian platform is interesting, but its conceptual advancement over existing humanized mouse or organoid models is limited and insufficiently demonstrated. The medical impact is rated Low, due to

modest translational potential; reliance on allogeneic PBMCs, brief evaluation periods, and the lack of mechanistic or predictive clinical correlations restrict clinical relevance. Lastly, the adequacy of the model system is considered Adequate, as the avian embryo provides logistical speed and imaging benefits, but significant physiological and immunological differences from humans diminish its ability to accurately model immune-tumor interactions in a meaningful clinical context.

Referee #3 (Remarks for Author):

Thank you for the opportunity to review this revised version. The authors have made significant efforts to address the reviewers' concerns, resulting in noticeable improvements in clarity, structure, and scope. Several new experiments, especially the standardization of PBMC donors, the addition of autologous immune-tumor pairs, and the inclusion of MHC-I blocking data, strengthen the technical foundation of the study.

However, despite these enhancements, important conceptual and methodological limitations still exist that prevent the manuscript from reaching the robustness expected for publication in a journal of this scope.

Most notably, the main claims that the humanized avian embryo system accurately models meaningful tumor-immune interactions and can predict immunotherapy outcomes are not yet backed by strong functional evidence. While the model shows co-localization and some activation of human immune cells, it lacks detailed mechanistic insights, direct cytotoxicity assays, cytokine measurements, or exploration of antigen presentation pathways that would establish biological relevance comparable to established preclinical systems.

Furthermore, although the authors included discussions comparing the avian model to humanized mouse models and organoid systems, the rationale for selecting the avian embryo platform over standard models remains largely conceptual rather than demonstrated through direct comparisons. The absence of functional benchmarks or validation evidence limits the translational potential of the work.

Given these points, and despite considerable revision efforts, I believe the manuscript, in its current form, does not meet the scientific standards required for acceptance. I acknowledge the novelty and ambition of this approach and encourage further development of the platform. Enhancing the functional immunological characterization and providing mechanistic evidence of immune engagement and therapeutic response will be crucial to increase the impact of this promising model in future work.

As a service to authors, EMBO provides authors with the possibility to transfer a manuscript that one journal cannot offer to publish to another EMBO publication. The full manuscript and if applicable, reviewers reports are automatically sent to the receiving journal to allow for fast handling and a prompt decision on your manuscript. For more details of this service, and to transfer your manuscript to another EMBO title please click on Link Not Available

Point-to-point answer to Referee 3

Referee #3 (Comments on Novelty/Model System for Author):

The manuscript demonstrates technically competent experimental work, but several methodological and conceptual limitations reduce its overall impact. Technical quality is rated as Medium, as the authors present detailed experimental procedures and new data; however, challenges remain regarding donor variability, lack of functional validation, and incomplete statistical rigor.

The novelty is also Medium, since the humanized avian platform is interesting, but its conceptual advancement over existing humanized mouse or organoid models is limited and insufficiently demonstrated.

We are surprised by this opinion, given the following statement made in the comment to the authors: 'I acknowledge the novelty and ambition of this approach, and encourage further development of the platform'.

We would like to respectfully point out that the humanized avian model offers unique advantages: high flexibility in combining human immune and cancer cells, and very rapid implementation, substantially shortening experimental timelines compared to humanized mouse models. In addition, the avian platform presents important ethical and logistical advantages, reducing animal use and experimental burden relative to murine systems. This aligns with current regulatory and funding initiatives aimed at minimizing and, where possible, replacing animal experimentation in biomedical research. The referee makes reference to organoids, but to the best of our knowledge, there is no standard paradigm for humanized organoids that would apply to 'all cancers'. Besides, efficiency to implement immune-competent tumor organoids depends on immune cell quality, ability to growth ex-vivo, co-culture design, and technical standardization, making it currently more suited to research and preclinical validation than routine clinical use.

We also recognize the reviewer's concern regarding donor-to-donor variability. Importantly, this challenge is not unique to the humanized avian platform; similar variability has been widely reported in humanized murine models, particularly those based on peripheral blood mononuclear cell (PBMC) injection, where donor heterogeneity influences immune reconstitution and experimental outcomes. By explicitly discussing this in the revised manuscript and framing it as a general limitation of humanized systems rather than a model-specific weakness.

The medical impact is rated Low, due to modest translational potential; reliance on allogeneic PBMCs, brief evaluation periods, and the lack of mechanistic or predictive clinical correlations restrict clinical relevance.

Please, see our comment below.

Lastly, the adequacy of the model system is considered Adequate, as the avian embryo provides logistical speed and imaging benefits, but significant physiological and immunological differences from humans diminish its ability to accurately model immune-tumor interactions in a meaningful clinical context.

The core principle of our platform is the establishment of a human immune cell-driven microenvironment. The avian embryo offers a unique experimental advantage in this regard, as it naturally lacks a functional adaptive immune system at the developmental stages used. This absence prevents host immune interference, like GvHD in humanized mouse model, and

enables the direct engraftment and interaction of human immune and tumor cells in vivo, without the need for genetic immunosuppression or irradiation.

Furthermore, we observe the full spectrum of immune populations present in human PBMCs over a short experimental timeframe, including T cells, B cells, NK cells, and myeloid populations. This short-term setting allows preservation of the initial cellular diversity of patient-derived PBMCs at the tumor site. In contrast, widely used murine hu-PBMC models are characterized by a rapid and dominant expansion of human T cells, while other immune populations are poorly maintained or rapidly lost. As a consequence, the resulting immune context becomes strongly T cell-biased and, over time, diverges from the cellular composition of human tumor immune microenvironments, which are typically heterogeneous and involve multiple immune lineages.

By maintaining a broader representation of PBMC-derived immune populations at early time points, our platform enables the study of early, localized immune-tumor interactions in a context that is more reflective of the multicellular immune environment observed in human tumors. While we acknowledge that this does not fully recapitulate systemic human immunity, it supports the relevance of the model for short-term mechanistic and exploratory studies.

Referee #3 (Remarks for Author):

Thank you for the opportunity to review this revised version. The authors have made significant efforts to address the reviewers' concerns, resulting in noticeable improvements in clarity, structure, and scope. Several new experiments, especially the standardization of PBMC donors, the addition of autologous immune-tumor pairs, and the inclusion of MHC-I blocking data, strengthen the technical foundation of the study.

However, despite these enhancements, important conceptual and methodological limitations still exist that prevent the manuscript from reaching the robustness expected for publication in a journal of this scope.

Most notably, the main claims that the humanized avian embryo system accurately models meaningful tumor-immune interactions and can predict immunotherapy outcomes are not yet backed by strong functional evidence.

We respectfully disagree with this opinion. Firstly, we present several pieces of experimental evidence in support of the existence of cancer-immune dialogues. Secondly, we do not claim that the avian model generally predicts immunotherapy outcomes in patients. Rather, we state that it reproduces the response expected from the clinic, based on currently used biomarkers of patient stratification. In the discussion, we make it clear that these markers are only partial, and we suggest that the avian model could facilitate the discovery of new biomarkers of anti-PD1 sensitivity and resistance.

Secondly, we selected cancer cell lines that had been reported in previous scientific publications to respond, or not, to anti-PD1 treatment. We demonstrate that the avian model reproduces these responses, which to our point of view, establishes its preclinical relevance to assess the efficacy of mAb targeting immune checkpoints.

While the model shows co-localization and some activation of human immune cells, it lacks detailed mechanistic insights, direct cytotoxicity assays, cytokine measurements, or exploration of antigen presentation pathways that would establish biological relevance comparable to established preclinical systems.

In summary, our main findings show that blocking MHC-1 with an antibody abrogates the effect of anti-PD1 administration. We also demonstrate that anti-PD1 is ineffective in the absence of the humanization process in avian embryos, which clearly establishes that this component is required to mediate the effect. In addition to the microscopy studies, we provide evidence from several flow cytometry investigations showing that T cells are inactivated by cancer cells and reactivated by anti-PD1 administration. Beyond tumor volume reduction, we observed a significant increase in cleaved caspase-3 in cancer cells, providing functional evidence that human T cells exert cytotoxic activity in this system. We acknowledge that additional analyses, such as cytokine profiling or direct cytotoxicity assays would further strengthen the mechanistic understanding of immune–tumor interactions. Nevertheless, these results provide the first functional evidence of cancer–immune dialogue in the humanized avian model and establish a solid foundation for future, more detailed investigations.

Furthermore, although the authors included discussions comparing the avian model to humanized mouse models and organoid systems, the rationale for selecting the avian embryo platform over standard models remains largely conceptual rather than demonstrated through direct comparisons. The absence of functional benchmarks or validation evidence limits the translational potential of the work.

Our philosophy is that all experimental models have inherent advantages and limitations, and their selection should be guided by the specific scientific questions being addressed. In this context, we highlight the distinguishing features of the humanized avian model, including the high success rate of patient cell engraftment with low cell numbers, the flexibility to combine different immune and cancer cell types, the possibility for rapid screening, and favorable ethical considerations. These characteristics address certain limitations of conventional mouse models, which often require large cell numbers, long experimental timelines, and complex breeding of immunodeficient strains. Conversely, mouse models offer advantages, such as long-term follow-up and fully developed mammalian physiology, which are not shared by the avian system.

Above all, the high rate of clinical attrition in oncology and immuno-oncology underscores the limitations of conventional preclinical models in accurately predicting human responses, contributing to the large number of failures in clinical trials. It is increasingly recognized that a combination of complementary models, through a “toolbox” approach, rather than reliance on a single “most relevant” model is needed to improve translational success. In this context, there is a growing consensus toward more human-centric models, which better recapitulate human biology and improve predictability. The humanized avian model represents a step in this direction, providing an additional platform to help bridge the gap between preclinical studies and patient outcomes.

Given these points, and despite considerable revision efforts, I believe the manuscript, in its current form, does not meet the scientific standards required for acceptance. I acknowledge the novelty and ambition of this approach and encourage further development of the platform. Enhancing the functional immunological characterization and providing mechanistic evidence of immune engagement and therapeutic response will be crucial to increase the impact of this promising model in future work.

We thank the referee for this comment, which summarizes the next steps perfectly.

4th Feb 2026

Dear Dr. Castellani, Dear Valerie,

Thank you for your email asking us to reconsider our decision, and please accept my apologies for the delay in getting back to you, which is due to reduced office activity over the holiday period and the high volume of new submissions received during that time.

Your manuscript was previously rejected post-revisions, as one referee found that methodological and conceptual limitations remained. You appealed this decision and provided a point-by-point rebuttal letter to the referee's concerns. We have considered your arguments and additionally sought the advice of an external expert who has read your manuscript, the referees' reports and your response to the latest reviews. The advisor has now got back to us and highlighted the platform's potential and novelty, as set out below:

"I have carefully read the correspondence between the authors and Rev #3 and the manuscript, and my thoughts are that although the model certainly has limitations, it is a novel approach to model the immune-tumor interface with human cells. It is not a model to replace existing humanized mice or human organoids [...] but rather to complement them. I think this is a rather out of the box idea and I like that as opposed to in vitro human organoids, it has functional vascularization. I understand the reviewer's concerns, but I think this is a first proof-of-concept and more work can be done to use this setting in the future to predict responses or test drugs."

Having discussed your manuscript and the referees' reports once more, and taking the advisor's feedback into account, we have decided to invite you to make minor revisions to the manuscript, in which the limitations highlighted by referee #3 should be discussed, and the proof-of concept nature of the work clearly stated.

Please also address the following editorial matters:

1/ Manuscript text:

- Please remove the red font and indicate in track changes mode any new modification.
- "Material and Methods" should be renamed "Methods".
 - o Cells: please indicate whether the cells were authenticated.
 - o Human samples: If collected and within the bounds of privacy constraints report on age, sex and gender or ethnicity for all study participants. Include a statement confirming that the experiments conformed to the principles set out in the WMA Declaration of Helsinki and the Department of Health and Human Services Belmont Report.
- Please enter funding information in the submission system and list the funders in the acknowledgements section.
- References: please reformat to have 10 author names listed before et al, please remove DOIs for published articles.
- Please address the queries from our data editors in the figure legends:
 1. Please note that the box plots need to be defined in terms of minima, maxima, centre, bounds of box and whiskers, and percentile in the legends of figures 3H, I; 4E, G.
 2. Please note that the error bars are not defined in the legend of figure 3B.
 3. Please do not include statistics and error bars for n=1 or n=2.

2/ Source Data:

- Please provide the links to the deposited .fcs files in the Data Availability section.

3/ Please provide The paper explained:

EMBO Molecular Medicine articles are accompanied by a summary of the articles to emphasize the major findings in the paper and their medical implications for the non-specialist reader. Please provide a draft summary of your article highlighting

4/ Please provide a synopsis:

Synopses are displayed on the journal webpage and are freely accessible to all readers. They include a short stand first (maximum of 300 characters, including space) as well as 2-5 one-sentences bullet points that summarizes the paper. They should be designed to be complementary to the abstract - i.e. not repeat the same text. We encourage inclusion of key acronyms and quantitative information (maximum of 30 words / bullet point). Please use the passive voice.

Please also suggest a visual abstract to illustrate your article as a PNG file 550 px wide x 300-600 px high.

5/ As part of the EMBO Publications transparent editorial process initiative (see our Editorial at <http://embomolmed.embopress.org/content/2/9/329>), EMBO Molecular Medicine will publish online a Review Process File (RPF) to accompany accepted manuscripts.

This file will be published in conjunction with your paper and will include the anonymous referee reports, your point-by-point response and all pertinent correspondence relating to the manuscript. Let us know whether you agree with the publication of the RPF and as here, if you want to remove or not any figures from it prior to publication.

I look forward to receiving your revised manuscript.

With kind regards,

Lise

1. Modify the proof-of concept nature of the work clearly stated.

We tried to take into account the concerns of the referee 3, the best we could.

These are the comments that this referee made on our revised work :

« While the model shows co-localization and some activation of human immune cells, it lacks detailed mechanistic insights, direct cytotoxicity assays, cytokine measurements, or exploration of antigen presentation pathways that would establish biological relevance comparable to established preclinical systems. »

« Enhancing the functional immunological characterization and providing mechanistic evidence of immune engagement and therapeutic response will be crucial to increase the impact of this promising model in future work ».

We have extended the discussion.

First, we modify the introducing paragraph of the discussion, replacing “we show” by “we provide the first proof-of-concept” p18.

“By combining experimental manipulations in the avian embryo, light sheet microscopy and molecular marker analyses, we provide **the first proof-of-concept that** this humanized AVI model enables assessing the efficacy of immunotherapeutic treatments.....

Second in the paragraph entitled “The humanized avian models reveal efficacy of anti-PD-1 administration” p22, we added: “By administering an anti-MHC-I antibody, we demonstrated that this anti-tumoral effect is mediated by T cell recognition of the tumor, providing initial mechanistic insights. **However, further research is needed in the form**

of direct cytotoxicity assays, cytokine measurements and exploration of antigen presentation pathways to better characterize immune engagement and the mechanisms underlying the therapeutic response. Due to the miniaturized nature of avian models, the dosage of chemokines in the tumor microenvironment is likely to require specific technical developments.”

Third, p24 in the paragraph “advantages and limitations of the humanized avian model” we added “ **Further detailed characterization of the immune-cancer communication will help to establish comparisons with existing models.**”

We hope that these modifications make the current stage of the technological development of the humanized AVI model clearer.

1/ Manuscript text:

- Please remove the red font and indicate in track changes mode any new modification.

Done

- "Material and Methods" should be renamed "Methods".

Done

o Cells: please indicate whether the cells were authenticated.

We indicated the provider source details and mention that cells were not re-authenticated, p26.

o Human samples: If collected and within the bounds of privacy constraints report on age, sex and gender or ethnicity for all study participants. Include a statement confirming that the experiments conformed to the principles set out in the WMA Declaration of Helsinki and the Department of Health and Human Services Belmont Report.

The sentence has been included, p26.

- References: please reformat to have 10 author names listed before et al, please remove DOIs for published articles.

Done

2. Please note that the error bars are not defined in the legend of figure 3B.

The definition has been added, p40.

3. Please do not include statistics and error bars for n=1 or n=2.

This comment applies to Figure 2, graph H, histogram “pre-graft”. The error bars of this histogram have been removed and the legend corrected accordingly.

The legend of Figure 2 C-D has also been clarified.

2/ Source Data:

- Please provide the links to the deposited .fcs files in the Data Availability section.

The links have been introduced in the “data availability” p32.

3/ Please provide The paper explained:

EMBO Molecular Medicine articles are accompanied by a summary of the articles to emphasize the major findings in the paper and their medical implications for the non-specialist reader.

Please provide a draft summary of your article highlighting

- the medical issue you are addressing,

- the results obtained and

- their clinical impact.

See point 4.

4/ Please provide a synopsis:

Synopses are displayed on the journal webpage and are freely accessible to all readers. They include a short stand first (maximum of 300 characters, including space) as well as 2-5 one-sentences bullet points that summarizes the paper. They should be designed to be complementary to the abstract - i.e. not repeat the same text. We encourage inclusion of key acronyms and quantitative information (maximum of 30 words / bullet point). Please use the passive voice.

A file has been uploaded that contains the draft summary and the synopsis.

Please also suggest a visual abstract to illustrate your article as a PNG file 550 px wide x 300-600 px high.

A visual abstract has been uploaded.

5/ As part of the EMBO Publications transparent editorial process initiative (see our Editorial at <http://embomolmed.embopress.org/content/2/9/329>), EMBO Molecular Medicine will publish online a Review Process File (RPF) to accompany accepted manuscripts. This file will be published in conjunction with your paper and will include the anonymous referee reports, your point-by-point response and all pertinent correspondence relating to the manuscript.

Let us know whether you agree with the publication of the RPF and as here, if you want to remove or not any figures from it prior to publication. Please note that the Authors checklist will be published at the end of the RPF.

We agree on the publication of the RPF.

19th Feb 2026

Dear Dr. Castellani, Dear Valerie,

Thank you for submitting your revised files. I am pleased to inform you that your manuscript is accepted for publication and is now being sent to our publisher to be included in the next available issue of EMBO Molecular Medicine.

You may qualify for financial assistance for your publication charges - either via a Springer Nature fully open access agreement or an EMBO initiative. Check your eligibility: <https://link.springer.com/journal/44321/how-to-publish-with-us>

With kind regards,

Lise

>>> Please note that it is EMBO Molecular Medicine policy for the transcript of the editorial process (containing referee reports and your response letter) to be published as an online supplement to each paper. If you do NOT want this, you will need to inform the Editorial Office via email immediately. More information is available here: <https://link.springer.com/partners/embo-press/editorial-policies#Peer%20review>